# Mode-Shell correspondence,
# a unifying phase space theory in topological physics –
# Part I: Chiral number of zero-modes

Lucien Jezequel[1] and Pierre Delplace[2]

[1,2]Univ Lyon, ENS de Lyon, CNRS, Laboratoire de Physique, F-69342 Lyon, France

April 16, 2024

### Abstract

We propose a theory, that we call the *mode-shell correspondence*, which relates the topological zero-modes localised in phase space to a *shell* invariant defined on the surface forming a shell enclosing these zero-modes. We show that the mode-shell formalism provides a general framework unifying important results of topological physics, such as the bulk-edge correspondence, higher-order topological insulators, but also the Atiyah-Singer and the Callias index theories. In this paper, we discuss the already rich phenomenology of chiral symmetric Hamiltonians where the topological quantity is the chiral number of zero-dimensional zero-energy modes. We explain how, in a lot of cases, the shell-invariant has a semi-classical limit expressed as a generalised winding number on the shell, which makes it accessible to analytical computations.

# 1 Introduction

The bulk-edge (or bulk-boundary) correspondence is a fundamental concept in topological physics. Very elegantly, it relates the number of robust gapless boundary modes of a physical system with a topological invariant defined from the bulk wavefunctions in reciprocal space of energy gapped materials. For that reason, such boundary states are said to be topological, or topologically protected. The bulk-edge correspondence was first explicitly introduced in the context of the quantum Hall effect [1] in order to clarify two interpretations of the quantised transverse conductance [2,3]: a bulk interpretation, where the transverse conductivity of an infinite quantum Hall system was theoretically shown to be proportional to a topological index of the Bloch bands [4], and an edge interpretation where uni-directional edge states were expected to exist as bended Landau levels due to edge confinement [5] and were shown to carry electric charges without dissipation along the boundaries in multi-probe (experimental) geometries [6,7].

Remarkably, the bulk-edge correspondence turned out to be the key concept that allowed the rise of topological physics beyond quantum matter. As initiated with photonic crystals [8–10], it was realized that the validity of the bulk-edge correspondence was much less demanding than the quantisation of a response function of the physical system, such as a conductivity. It followed that many experimental platforms, quantum and classical, emerged with the aim of engineering, probing and manipulating robust boundary modes, for instance, for robust wave guiding [11] or quantum computing [12,13]. Another success of the bulk-edge correspondence is its validity in any dimension. For instance, the observation, through ARPES measurements, of topologically protected surface

states behaving as two-dimensional ($2D$) massless Dirac fermions, was a convincing experimental proof of the existence of $3D$ topological insulators [14], while no equivalent of a quantised bulk conductivity was available as an alternative signature. Actually, this success of the bulk-boundary correspondence was twofold, because those $3D$ topological insulators also belonged to a different symmetry class than that of the quantum Hall effect it was originally conceived. The correspondence still holds in other symmetry classes and in arbitrary dimension [15–18], but the bulk topological invariant changes and accordingly, the nature of the boundary modes it describes: massless Dirac fermions as $2D$ surface states of $3D$ topological insulators [14,19], helical Kramers pairs as $1D$ edge states of the $2D$ quantum spin Hall effect [20–22], Majorana quasi-particles as $0D$ boundary modes of $1D$ topological superconducting wires [13], are among the most famous examples.

Since then, the bulk-edge correspondence has been challenged several times, and had to adapt to incorporate new phenomenologies that did not fit the standard paradigm. One important development of the last years was the discovery of higher order topological insulators, that display a richer hierarchy of boundary modes that are not predicted by the usual bulk-boundary correspondence. In $3D$, such materials can host hinge states or corner states rather than surface states [23–25]. Another recent fruitful direction is the study of topological modes in continuous media, mostly motivated by classical wave physics, such as geo- and astrophysical fluids [26–29], active fluids [30], plasmas [27] but also photonics [31,32]. Of interest was the apparent failure of the bulk-edge correspondence in the absence of a lattice – and thus of a compact reciprocal space – that stimulated several extension works [33–40]. One way to address the problem is to focus on domain-walls made of smooth varying potentials rather than hard wall boundaries. This fruitful approach allows the description of interface modes, at the domain wall, whose topological origin is encoded in phase space rather than in reciprocal space, leading to what we could call a "phase space - interface/domain wall" correspondence. Similarly to the topological characterization of Weyl nodes in $3D$, such a phase space approach also allows for counting topological modes of $2D$ systems in a specific valley, as encountered in various valley-Hall effects or in fluids, by assigning them directly an integer-valued topological invariant [35,41–43], without needing to resort to valley Chern numbers (that are purely defined in $k$-space) that typically involve half-integer numbers [44–47]. Interestingly, the combined use of real and reciprocal space is also used to characterize the topology of defect modes in insulators and superconductors, which constitutes another important generalization of the bulk-boundary correspondence [48]. A last stimulating development of the bulk-edge correspondence we would like to mention concerns non-Hermitian systems. This field of research can somehow be traced back to the finding of topological edge states in periodically driven (Floquet) systems [49–52], quantum walks [53,54], and scattering networks [55,56], whose full description is ruled by unitary operators, such as evolution operators or scattering matrices, rather than usual Hermitian Hamiltonians [50,57]. In that context, the standard bulk-edge correspondence was also found to fail, but suitable generalizations, with new topological invariants that account for the periodic dynamics, were then found out in various symmetry classes [50,51,58,59]. Since more recently, non-Hermitian topology rather more implicitly designates classical or quantum systems whose dynamics displays topological properties that are dictated by non-unitary evolutions, whether because of different sources of gain or loss [60–83].

These numerous upheavals and challenges have continually refined the bulk-edge correspondence's contours in order to preserve this powerful concept. This evolution goes together with the difficulty to provide a general formalism encompassing such a rich phenomenology, while being also both based on sound mathematics and of practical convenience for most physicists. Many proofs of bulk-edge correspondences exist in the literature [2,3,38–40,84–89]. Some of them are based on

basic reasonings and elementary math accessible to undergraduates students, but tackle only very specific examples. Other much more advanced proofs gain in generality but their abstraction prevents the development of a physical intuition. Of course, the seek for rigor and generality stimulated many mathematical works, besides physics studies; nothing more legitimate from a field sometimes designated as *topological physics*, and that borrows so explicitly from mathematics. Actually, back to the seventies, a series of mathematical works revealed a somehow similar phenomenology. Of particular importance is the Callias index theory [90–92] that implies, in our language, that certain operators defined on continuous infinite (unbounded) spaces, like $\mathbb{R}^n$, exhibit topological modes, similarly to aforementioned interface states in continuous systems. This theory is itself an extension of the seminal Atiyah-Singer index theory [93–96] that applies for continuous but compact/bounded spaces, such as a circle or a torus. Obviously, and in contrast with the bulk-edge correspondence, in such situations, the topological modes cannot be edge states owing to the absence of boundary in the system.

The purpose of this article is to shape a theory that generalizes the bulk-edge correspondence in all the possible directions mentioned above. Such a theory must apply to any dimension, account for discrete and continuous spaces, be independent of translation invariance, describe both boundary and interface states, as well as higher order hinge or corner states. It should also capture topological modes which are not purely localized in space, but instead live in a delimited region of reciprocal space, such as a valley. We call this theory the *mode-shell* correspondence. Similarly to index theories, the mode-shell correspondence states the equality between two integer-valued indices $\mathcal{I}_{\text{modes}} = \mathcal{I}_{\text{shell}}$. The first index gives the number of *modes* (e.g. edge states) in a certain energy region, while the second one is an invariant defined an a *shell* surrounding this *mode* in phase space $(x, k) \in \mathbb{R}^{2n}$ where the Hamiltonian (or more generically the wave operator) is assumed to be gapped. This general and basic equality is the first brick of the theory. If the two indices can in principle always be computed numerically, it is illusory to expect them to be computable analytically in general. One may however hope to have a simpler formulation in systems with some structure. This brings us to the second step of the theory, which consists in a semiclassical approximation [97] of the shell invariant $\mathcal{I}_{\text{shell}}$. When such an approximation is possible, $\mathcal{I}_{\text{shell}}$ can be expressed as an integral over the shell. In that limit, one recovers well-known expressions of so-called bulk topological invariants, such as winding numbers and Chern numbers, but also of less standard invariants which are nonetheless physically relevant.

In order to keep the presentation intelligible and the article length reasonable, we shall focus in this paper on Hermitian wave operators with chiral symmetry. More specifically, we shall even only focus on *zero-dimensional zero-energy modes*, simply dubbed zero-modes in the following. Such modes are usually associated to the edge states of $1D$ systems in the AIII symmetry class of the tenfold way classification of topological insulators and superconductors [15]. The *zero-dimensional* denomination means that those modes are localized in "every" direction (with a finite short extension). Here, our aim is not to provide another derivation of the tenfold way, but rather to extract the many topological aspects of this single and apparently simple case, through the mode-shell correspondence. Indeed, while those zero-modes are captured by a single mode index $\mathcal{I}_{\text{modes}}$, the semiclassical expression of $\mathcal{I}_{\text{shell}}$ it is equal to changes, depending on the configuration at hand (lattice problem, domain wall, higher-order topological insulator...). This way, the mode-shell theory generates the "bulk" invariants in a systematic way. In this paper, we provide an explicit derivation of the mode-shell correspondence in the chiral case for zero-modes, and illustrate it with several

explicit models.

The outline of the paper is as follows. In section [2], we present a non technical overview of the mode-shell correspondence. In particular, we introduce the mode invariant $\mathcal{I}_{\mathrm{mode}}$ for chiral symmetric systems, and show how it is related to the shell invariant $\mathcal{I}_{\mathrm{shell}}$. We introduce the notion of the symbol Hamiltonian $H(x, k)$ that is a phase space representation of the operator Hamiltonian $\hat{H}(x, \partial_x)$ through a Wigner-Weyl transform. We discuss the semi-classical approximation that simplifies the shell invariant into a general winding number for arbitrary dimensional systems. Section [3] is dedicated specifically to $1D$ systems. The mode-shell correspondence is then derived and illustrated on models for $1D$ lattices, continuous bounded and continuous unbounded geometries. From there, we show that the mode-shell correspondence includes the bulk-edge and "phase space -interface/domain wall" correspondences, where zero-energy modes are localized in position $x$-space at a boundary or an interface, but it also describes a dual situation where the topological modes are localized in wavenumber $k$-space, and even an hybrid situation with a confinement in $x - k$ phase space. Section [4] is devoted to higher dimensional chiral symmetric systems hosting zero-modes whose topological origin can eventually be reduced to that of the chiral zero-dimensional zero-energy modes described in section [3]. Those cases include (but are not restricted to) weak and higher order chiral topological insulators.

Other higher dimensional chiral symmetric topological systems are expected from the tenfold way classification [15], such as strong chiral topological insulators in $3D$. Those are not discussed in the present paper, but will be treated in the Part II of this work, where the mode-shell correspondence will be applied to address higher dimensional topologically protected modes, such as $1D$ spectral flows of quantum Hall systems, $2D$ Dirac and $3D$ Weyl fermions.

## 2 Overview of the mode-shell correspondence for chiral symmetric systems

The aim of this section is to introduce, in a non technical way, the mode-shell correspondence by focusing on the zero-dimensional zero-energy modes of chiral symmetric systems.

### 2.1 Chiral symmetry and chiral index

In this section, we introduce an index, denoted by $\mathcal{I}_{\mathrm{modes}}$, that counts the number of chiral zero-energy modes. This index can be used when the Hamiltonian $\hat{H}$ has a chiral symmetry, that is, when there exists a unitary operator $\hat{C}$ satisfying the anti-commutation relation $\hat{H}\hat{C} + \hat{C}\hat{H} = 0$. This symmetry typically appears when the system is bi-partied in two groups of degrees of freedom $A$ and $B$, such that the Hamiltonian only couples $A$ and $B$. These two groups can, for example, be two groups of atoms that interact in a lattice through a nearest neighbor interaction (see Figure [1]).

Chiral symmetry is given by a diagonal operator in the $A - B$ block basis, with coefficients $+1$ on A and $-1$ on B, that is

$$\hat{C} = \begin{pmatrix} \mathbb{1}_A & 0 \\ 0 & -\mathbb{1}_B \end{pmatrix} \tag{1}$$

where $\mathbb{1}$ denotes the identity operator. We shall call such a basis the *chiral basis* in the following. The *chirality* of a mode $|\psi\rangle$ then refers to the eigenvalues of the chiral operator; it is $+1$ for the modes $|\psi\rangle$ satisfying $\hat{C}|\psi\rangle = |\psi\rangle$ and $-1$ for those satisfying $\hat{C}|\psi\rangle = -|\psi\rangle$. The chirality is a

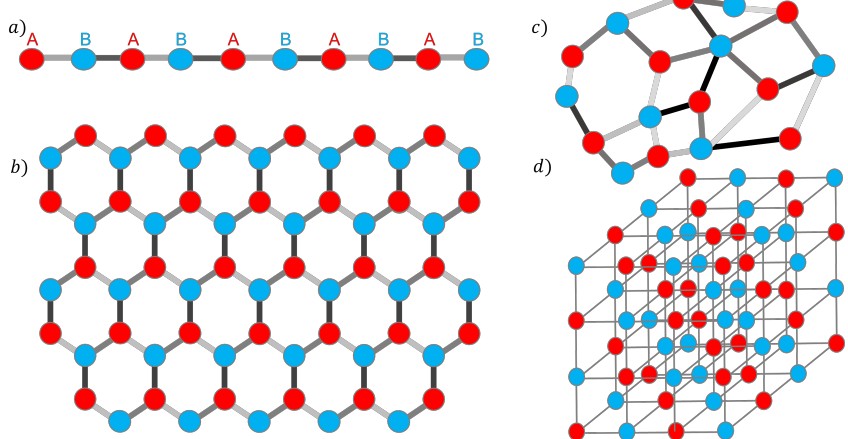

Figure 1: Examples of chiral lattices in a) $1D$, b) and c) $2D$ and d) $3D$. The example c) illustrates a disordered (amorphous) lattice which is still chiral symmetric. The red/blue dots represent the sites of opposite chirality and the grey links represent the different couplings between those sites. All these lattices can host topological modes for well-chosen Hamiltonians.

signature of the polarisation of the modes on the A or B degrees of freedom. In the chiral basis, the Hamiltonian is off-diagonal

$$\hat{H} = \begin{pmatrix} 0 & \hat{h}^{\dagger} \\ \hat{h} & 0 \end{pmatrix} \tag{2}$$

where the operators $\hat{h}$ and $\hat{h}^{\dagger}$ encode the couplings between $A$ and $B$ degrees of freedom. It follows from (1) and (2) that the identity $\hat{C}\hat{H} + \hat{H}\hat{C} = 0$ is automatically satisfied. A direct consequence of chiral symmetry is that every eigenstate $|\psi\rangle$ of $\hat{H}$ with a non-zero eigen-energy $E$ comes with a chiral symmetric partner $\hat{C}|\psi\rangle$ of opposite energy $-E$.

A special attention will be paid on zero-energy modes of chiral symmetric systems (usually simply dubbed *zero-modes*). The key point is that those zero-modes are topologically protected when they are exponentially localized in *regions* outside of which the Hamiltonian is gapped. Those regions can for instance correspond to edges, interfaces or defects in real space, and the zero-modes then correspond to various kinds of boundary states. But we will see that those regions may also more generally designate a part of phase space (position and wavenumber space). Here we are concerned with a *chiral index* that counts algebraically the number of localized zero-energy modes in those regions, with a sign given by their chirality. In other words, the chiral index counts the total chirality of the zero-modes and can thus be formally introduced as

$$\mathcal{I}_{\mathrm{modes}} = \#\,\text{zero-modes of chirality} +1 - \#\,\text{zero-modes of chirality} -1 \ . \tag{3}$$

An alternative (although equivalent) definition of the chiral index can be found by using the off-diagonal structure (2) of $\hat{H}$ in the chiral basis, so that the zero-modes $|\psi\rangle = (|\psi\rangle_A, |\psi\rangle_B)^t$ must satisfy

$$\hat{H}|\psi\rangle = 0 = \begin{pmatrix} \hat{h}^{\dagger}|\psi\rangle_B \\ \hat{h}|\psi\rangle_A \end{pmatrix} . \tag{4}$$

It follows that the zero-modes $|\psi\rangle$ of positive chirality are in bijection with the $|\psi\rangle_A$ in the kernel of $\hat{h}$. The zero modes of negative chirality are as well in bijection with the $|\psi\rangle_B$ in the kernel of $\hat{h}^\dagger$. So one can rewrite the index in the commonly used form

$$\mathcal{I}_{\text{modes}} = \dim \ker(\hat{h}) - \dim \ker(\hat{h}^\dagger) \equiv \text{Ind}(\hat{h}). \tag{5}$$

where $\text{Ind}(\hat{h})$ is known as the analytical index of the operator $\hat{h}$.

It is worth stressing here that chiral symmetry is not restricted to lattices, and can also be encountered when dealing with classical waves in continuous media. Actually, it turns out that chiral symmetry of the wave operator for fluid models follows from time-reversal symmetry of the original set of primitive differential equations. This is because primitive equations – basically momentum conservation and mass conservation – are first order differential equations in time. Indeed, they yield a relation between fields that are odd with respect to time inversion $t \rightarrow -t$, such as velocity fields $v(t) \rightarrow -v(-t)$, and fields that are even with respect to time reversal, such as pressure fields $p(t) \rightarrow p(-t)$. When time-reversal symmetry is satisfied, the left and right members of those equations must have the same parity under time reversal. But because of the first order differentiation of those fields with respect to time, that also changes sign under time-reversal $\partial_t \rightarrow -\partial_t$, the time derivative of the odd fields is only given by even fields, and *vice versa*. This automatically creates a bipartition between the physical fields depending on their parity in time, which eventually translates into a chiral symmetry of the wave operator in phase space. Such a structure appears in geo- and astrophysical fluid dynamics in the absence of Coriolis force [28, 41] and in plasmas models [27, 29] provided the magnetic field is off. [1]

In the rest of the paper, we will develop a theory that applies for both discrete and continuous media, quantum or classical, and we will keep the notation $\hat{H}$ when referring to classical wave operators, that will be assumed to be Hermitian, and even abusively call them "Hamiltonians" for the sake of standardizing the notations.

## 2.2 Role and necessity of a smooth energy filter $f(E)$

Actually, the definitions (3) and (5) of the chiral index only work in idealized infinite systems but are difficult to manipulate or to approximate in finite size systems. Indeed, in finite size systems, the zero-modes of the different regions are always coupled with each other through exponentially small but non-zero overlapping. This coupling, in general, shifts the energy of the modes such that, in perfect rigour, one never reaches perfect zero-energy modes. To overcome this limitation, we introduce a formulation of the chiral index that is continuous in the coefficients of $\hat{H}$, making it easier to manipulate in practical computations and simulations.

To do so, we first assume the system to be gapped far away from the zero-mode, and we denote by $\Delta > 0$ the half-amplitude of the gap $[-\Delta, \Delta]$. Then we define the operator $\hat{H}_F = f(\hat{H})$ where we choose $f$ to be an odd function taking the value $-1$ for negative gapped energies $E < -\Delta$ and $+1$

---

[1]Let us stress that in classical waves systems, time-reversal symmetry is just an orthogonal symmetric matrix which is 1 on even degrees of freedom and $-1$ on the odds ones. This is different from quantum mechanics where the Schrödinger equation carries a complex structure and where time-reversal symmetry is encoded as a complex conjugation or in general as an anti-unitary operator. This may cause confusion when one wants to apply to classical waves the ten-fold classification [15,16] which was constructed with the quantum version of the time-reversal symmetry in mind.

for positive gapped energies $E > \Delta$ with a smooth transition in the gapless region in between (see Figure 2). This means that $\hat{H}_F$ is the operator with the same eigenmodes as $\hat{H}$ but with rescaled energies $E \to f(E)$. This operation flattens the gapped bands and hence $\hat{H}_F$ can be seen as a flatten Hamiltonian. Then the chiral index can be formally defined as

$$\mathcal{I}_{\text{modes}} = \text{Tr}\left(\hat{C}(1 - \hat{H}_F^2)\right) . \tag{6}$$

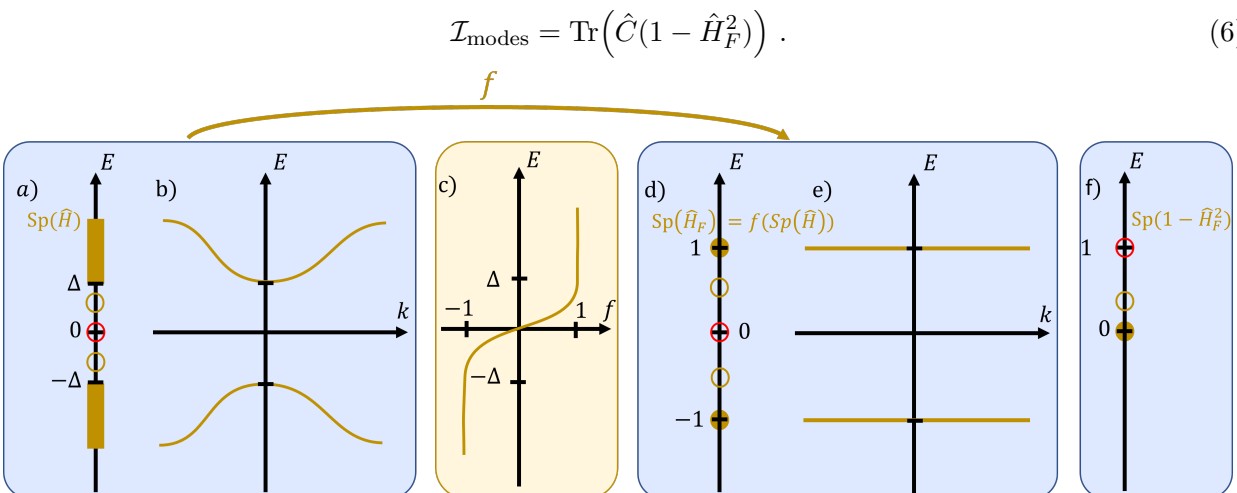

Figure 2: a) Projected energy spectrum denoted $\text{Sp}(\hat{H})$ of a typical topological Hamiltonian $\hat{H}$. United stripes denote the gapped bulk bands, circles denote the isolated eigenvalue of modes localised at the edge and red circles denote the topological zero modes. b) Dispersion relation in the bulk where edge modes cannot be seen. c) Sketch of a possible smooth flattening function. d-e) Spectrum of the operator $\hat{H}_F = f(\hat{H})$ where the bulk bands are flattened. f) Spectrum of $1 - \hat{H}_F^2$, where only a finite number of non-zero excitations remains: the bulk excitations vanish and the original zero-modes are dominant.

To see why (6) is indeed a meaningful definition of the chiral index, we express it in a common diagonal basis $\{|\psi_\lambda\rangle\}$ of $\hat{C}$ and $\hat{H}^2$ (which is always possible since $[\hat{H}^2, \hat{C}] = 0$ as $\{\hat{H}, \hat{C}\} = 0$) and we get

$$\mathcal{I}_{\text{modes}} = \sum_\lambda C_\lambda(1 - f(E_\lambda)^2) \langle\psi_\lambda|\psi_\lambda\rangle = \sum_\lambda C_\lambda(1 - f(E_\lambda)^2) \tag{7}$$

with $C_\lambda$ and $E_\lambda^2$ the eigenvalues $\hat{C}$ and $\hat{H}^2$. The term $1 - f(E_\lambda)^2$ is identically zero for all the modes that do not lie in the gap $[-\Delta, \Delta]$. We are thus left with the zero modes we would like to keep (full circles in figure 2), and *a priori* other gapless but non-zero modes (hollow circles in figure 2). As a matter of fact, the latest come by pairs of opposite chirality, due to chiral symmetry as if $|\psi\rangle$ is an eigenmode of both $\hat{H}^2$ and $\hat{C}$ with eigenvalues $E_\lambda^2$ and $C_\lambda$ then $H|\psi\rangle$ is also an eigenmode with eigenvalues $E_\lambda^2$ and $-C_\lambda$ except when $H|\psi\rangle = 0$. They therefore cancel out two by two in the sum thanks to the introduction of the chiral operator $\hat{C}$ in the definition of $\mathcal{I}_{\text{modes}}$. The only contributions that remain are those of the zero-energy modes $\hat{H}|\psi_\lambda\rangle = 0$ that do not allow a valid way to construct a symmetric partner of opposite chirality. So we end up with $\mathcal{I}_{\text{modes}} = \sum_{\lambda, E_\lambda = 0} C_\lambda$ which is exactly the chirality of the zero-modes.

The two equivalent expressions (3) and (6) of the chiral index $\mathcal{I}_{\text{modes}}$ show that the number of zero-modes of the Hamiltonian $\hat{H}$ is a topological quantity: (3) shows that $\mathcal{I}_{\text{modes}}$ is an integer number while (6) shows that it depends continuously of the Hamiltonian. $\mathcal{I}_{\text{modes}}$ is therefore an integer that is stable under smooth variations of the coefficients of $\hat{H}$, hence its topological nature.

However, as they are written, the different expressions of $\mathcal{I}_{\mathrm{modes}}$ count the *total* number of zero-modes of $\hat{H}$. This is an issue when dealing with finite size systems, or with numerical simulations, that involve more than one gapless region (e.g. two edges, multiple corners ...). In those cases, one is more interested in the chirality of the modes localised in specific sub-regions of phase space (just counting the zero-modes near an edge/corner/...) than the total chirality of the zero-modes of the entire system, which is also often trivial. One therefore needs a cut-off in phase space to obtain this *local* topological information, a process we now aim at describing.

## 2.3 Role and necessity of a phase space filter $\hat{\theta}_\Gamma$

In order to capture the chiral zero-modes in specific regions of phase space, one needs to add, to the definition of $\mathcal{I}_{\mathrm{modes}}$, a function $\hat{\theta}_\Gamma$ that selects the a zero-mode in phase space. (sketched in red in figure 3). Such a *cut-off operator* is close to identity over a gapless target region that encloses the zero-mode, over a typical distance $\Gamma$ (in green in figure 3), and then drops to zero away from it, where the Hamiltonian is gapped. We shall later refer to the domain where $\hat{\theta}_\Gamma$ drops as the *shell*. In this way, the selected zero-modes are localised within the shell, while the other zero-modes remain outside (in blue in figure 3). A *local* version of the the chiral index thus reads

$$\mathcal{I}_{\mathrm{modes}} = \mathrm{Tr}\Big(\hat{C}(1 - \hat{H}_F^2)\hat{\theta}_\Gamma\Big) \tag{8}$$

and which, by construction, counts the chirality of the zero-modes in a selected region of phase space. More formally, this phase space representation of zero-modes is typically made possible thanks to a Wigner transform, that we introduce in section 2.5. The red and blue gapless regions in figure 3 are thus sketches of the amplitude of the Wigner function of the zero-modes.

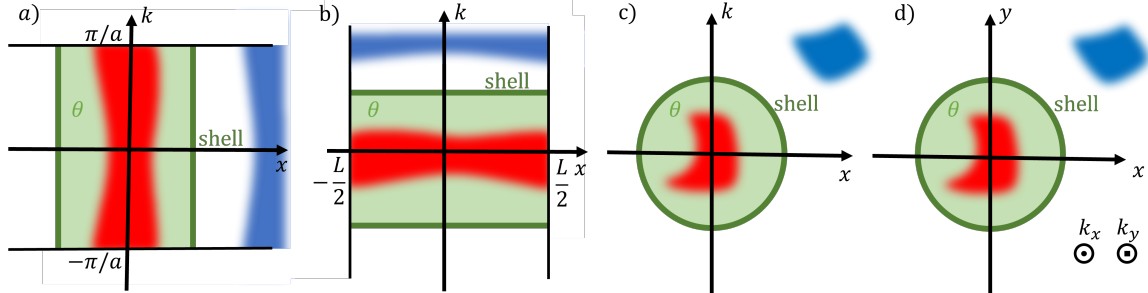

Figure 3: Sketches of the Wigner representation of zero-modes (in red/blue depending of their positive/negative chirality) embedded in different phase spaces with examples where the zero-modes are localised a) in position but not in wavenumber b) in wavenumber but not in position c) in position and in wavenumber d) in position but in a $2D$ space. The cut-off function $\theta$ selects a region that encloses one zero-mode by taking the value $\theta \sim 1$ (green), while dismissing the other gapless regions ($\theta \sim 0$ in white). The transition region where the cut-off goes from one to zero is called the shell (dark green line).

Importantly, the quantisation of the index does not strongly depend on the shape of the shell, nor on how the cut-off operator is explicitly defined, as long as it is close to identity in the target gapless region (where the Wigner representation of the zero-mode is located) within the shell and close to zero in the other gapless regions, outside the shell. As we will see below, the target region, defined in phase space, is in correspondence with the localisation of the zero-modes, and many

situations can be covered by the same local chiral index (8), which makes it quite general and powerful.

For instance, if we are interested in finding a zero-mode localized in real space, at an edge of a $1D$ chain, positioned around $x \sim 0$, the cut-off operator should target a region in phase space which is therefore only constrained in the $x$ direction and not in wavenumber $k$. Although a specific expression of the cut-off function is not necessary, the reader can think of its profile as that a Gaussian $\hat{\theta}_\Gamma = e^{-x^2/\Gamma^2}$ or a Fermi-like function $\hat{\theta}_\Gamma = (1 + e^{x^2 - \Gamma^2})^{-1}$. In those expressions, the cut-off parameter $\Gamma$ determines how large is the selected region. Such a real-space cut-off is illustrated in figure 3 a) and used in the mode-shell correspondence in section 3.1 to derive the bulk-boundary correspondence.

Our formalism allows to tackle the dual situation of the previous case on the same footing, where the zero-modes are now localized in wavenumber, for instance in the slow varying modes region of a continuous Hamiltonian. Possible cut-off operators then read $\hat{\theta}_\Gamma = e^{\Delta/\Gamma^2} \approx e^{-k^2/\Gamma^2}$ or $\hat{\theta}_\Gamma = (1 + e^{-\Delta - \Gamma^2})^{-1} \approx (1 + e^{k^2 - \Gamma^2})^{-1}$ where $\Delta$ is the Laplacian operator, and the associated target region in phase space is represented in figure 3 b). This formalism is then similar to the so-called heat kernel approach used in the context of the Atiyah-Singer index theorem. A model displaying such zero-modes is addressed in section 3.2.

More generally, the zero-modes can also be localised in a mixed way in position/wavenumber and cut-off operators can then be chosen as $\hat{\theta}_\Gamma = e^{(-x^2 + \partial_x^2)/\Gamma^2} \approx e^{-(x^2 + k^2)/\Gamma^2}$ or $\hat{\theta}_\Gamma = (1 + e^{x^2 - \partial_x^2 - \Gamma^2})^{-1} \approx (1 + e^{x^2 + k^2 - \Gamma^2})^{-1}$ in that case[2]. The shell enclosing the target region in phase space is then typically a circle (figure 3 c) and a corresponding example is shown in section 3.3.

Finally, this approach can be generalized to higher dimensions, to address zero-modes in higher-order topological insulators with chiral symmetry. A simple example is that of corner states of a two-dimensional system. In that case, cut-off operators can be chosen as $\hat{\theta}_\Gamma = e^{-(x^2 + y^2)/\Gamma^2}$ or $\hat{\theta}_\Gamma = (1 + e^{x^2 + y^2 - \Gamma^2})^{-1}$, and the target region in phase space is shown in figure 3 d). This higher dimensional case, is discussed among others in section 4.

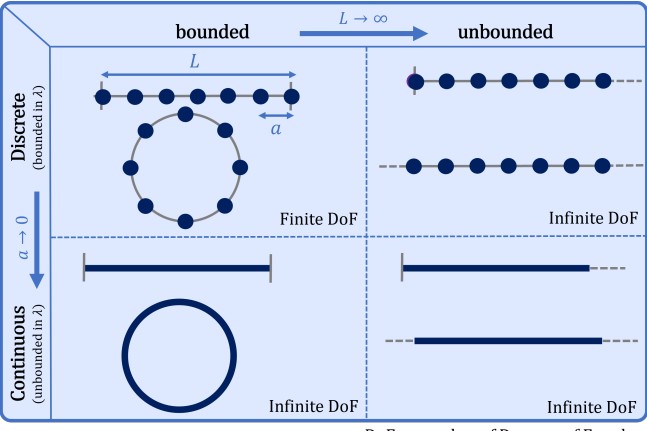

Figure 4: Table summarising the different categories of systems according to two infinite limits: a large length limit, where the length $L$ of the system is considered as infinitely large, and a small length limit where the characteristic distance $a$ between two sites becomes infinitely small and where therefore the set of possible wavenumbers becomes unbounded.

---

[2]We choose to work with adimensioned models, hence the adimensioned expression in $x$ and $k$.

In finite systems, the necessary introduction of a cut-off operator alters the quantisation of the chiral index, which is no longer exactly an integer. However, in large systems, when the gapless regions we want to select are far away from each other in phase space and $\Gamma$ is large, the correction to an integer value decays exponentially fast with the sizes of the system [98] and it is reasonable to still talk about quantised index with a satisfying approximation. Moreover in the limit case of infinite systems, the cut-off parameter $\Gamma$ can be put to infinity, so that $\hat{\theta}_\Gamma$ is replaced by the identity and we recover the previous exact index (6).

In all those cases, the notion of "large system" should be understood as large compared to the typical coupling distances of the Hamiltonian $\hat{H}$ in phase space. So, if the cut-off operator acts in position space, we need $\hat{H}$ to be short-range in position space, and the system's size must be large compared to the typical coupling distance in position. The unbounded limit $L \to \infty$ in figure 4 satisfies this condition. If the cut-off operator acts in wavenumber space, we need $\hat{H}$ to be short-range in wavenumber space and the lattice wavenumber $k_0 = 2\pi/a$ to be large compared to typical coupling distance in wavenumber. The continuous limit $a \to 0$ in figure 4 satisfies this condition. Also, another reason why we choose $f$ to be a smooth function in energy is because it is a required property to extend the short-range behaviour in phase space of $\hat{H}$ to $\hat{H}_F$ (see Appendix C).

## 2.4 Mode-shell correspondence

The chiral index we have introduced requires the use of the cut-off operator that embeds a gapless target region in phase space where zero-modes live. The boundary of this embedding, namely the shell, plays a crucial role in the theory that we now want to emphasize. This is due to the fact that, up to a rearrangement of its terms the index $\mathcal{I}_{\text{modes}}$ can be shown to be equal to an invariant $\mathcal{I}_{\text{shell}}$, that essentially depends on the properties of $\hat{H}$ *on* the shell, the region where the cut-off drops from the identity to zero. This index reads

$$\mathcal{I}_{\text{shell}} = \text{Tr}\left(\hat{C}\hat{\theta}_\Gamma\right) + \frac{1}{2}\,\text{Tr}\left(\hat{C}\hat{H}_F[\hat{\theta}_\Gamma, \hat{H}_F]\right) . \tag{9}$$

The first term $\text{Tr}\left(\hat{C}\hat{\theta}_\Gamma\right)$ does not depend on $\hat{H}$. It is the polarisation in the number of degrees of freedoms of positive/negative chirality, weighted by $\theta_\Gamma(x)$. For example, in a lattice, this term is just the polarisation in the number of sites of positive/negative chirality (again weighted by $\theta_\Gamma(x)$). In this paper we will mostly deal with situations where the density of states of positive/negative chirality is *balanced* and where this term therefore vanishes. If such density of states does not compensate, then this term is necessary to recover the mode-shell correspondence [99–101].

The second term, $\frac{1}{2}\,\text{Tr}\left(\hat{C}\hat{H}_F[\hat{\theta}_\Gamma, \hat{H}_F]\right)$, does depend on $\hat{H}$. However, the trace contains the commutator $[\hat{H}_F, \hat{\theta}_\Gamma]$ which vanishes both *inside* the shell, where $\hat{\theta}_\Gamma \approx \mathbb{1}$ (any operators commutes with the identity), and *away* from the shell since there $\hat{\theta}_\Gamma \approx 0$. Therefore, the non-negligible contributions of the trace only come from the shell which is the region where $\hat{\theta}_\Gamma$ goes from the identity to zero. This property explains the appellation of the index $\mathcal{I}_{\text{shell}}$.

The fact that $\mathcal{I}_{\text{modes}}$ can be re-expressed into $\mathcal{I}_{\text{shell}}$ is proved in a few lines of algebra. In fact it suffices to use the anti-commutation relation with the chirality operator $\hat{C}\hat{H}_F = \hat{C}f(\hat{H}) = f(-\hat{H})\hat{C} = -\hat{H}_F\hat{C}$ (remember that $f$ is an odd function) as well as the cyclicity of the trace to

rearrange the terms in the following order[3], and we get

$$
\begin{aligned}
\mathcal{I}_{\text{modes}} &= \text{Tr}\Big(\hat{C}\hat{\theta}_\Gamma(1 - \hat{H}_F^2)\Big) \\
&= \text{Tr}\Big(\hat{C}\hat{\theta}_\Gamma\Big) - \text{Tr}\Big(\hat{C}\hat{\theta}_\Gamma\hat{H}_F^2\Big) \\
&= \text{Tr}\Big(\hat{C}\hat{\theta}_\Gamma\Big) - \frac{1}{2}\left(\text{Tr}\Big(\hat{H}_F^2\hat{C}\hat{\theta}_\Gamma\Big) + \text{Tr}\Big(\hat{H}_F\hat{C}\hat{\theta}_\Gamma\hat{H}_F\Big)\right) \\
&= \text{Tr}\Big(\hat{C}\hat{\theta}_\Gamma\Big) + \frac{1}{2}\text{Tr}\Big(\hat{C}\hat{H}_F[\hat{\theta}_\Gamma, \hat{H}_F]\Big)
\end{aligned}
\tag{10}
$$

which shows the equality

$$
\mathcal{I}_{\text{modes}} = \mathcal{I}_{\text{shell}}
\tag{11}
$$

that we call the *mode-shell correspondence*, as it relates the number of chiral zero-modes to a property on the shell surrounding those modes in phase space. Because of this equality, we will use the notation $\mathcal{I}$ to denote both indices.

In general, the index $\mathcal{I}$ can be computed numerically and is prone to describe the topology of in-homogeneous or disordered systems since its definition does not rely on any periodicity assumption. However the shell formulation of the invariant is particularly suitable to semi-classical approxima-tions [97] in a lot of systems which simplifies its computation and provides another topological meaning to the index.

## 2.5 Winding numbers as semi-classical limits of the chiral invariant in phase space

The index formulation we developed is made at the operator level whereas semi-classical approxima-tions are usually performed in phase space $(x, k)$ ($\hbar = 1$ in the quantum situations). The connection between, on one hand, operators such as the cut-off operator $\hat{\theta}_\Gamma$ or the Hamiltonian $\hat{H}$, and, on the other hand, functions in phase space, is made possible by Wigner-Weyl calculus. In particular, we will use the Wigner transform of the Hamiltonian operator, defined as (see Appendix B)

$$
H(x, k) = \int_\mathbb{R} dx' \left\langle x + \frac{x'}{2}\middle| \hat{H} \middle| x - \frac{x'}{2}\right\rangle e^{-ikx'}
\tag{12}
$$

with $k \in \mathbb{R}$ when the Hamiltonian $\hat{H} = H(x, \partial_x)$ is a differential operator that describes a continuous model, and as

$$
H(n, k) = \sum_{n'} \left\langle n'\middle| \hat{H} \middle| n\right\rangle e^{-ik(n'-n)}
\tag{13}
$$

with periodic parameter $k \in [0, 2\pi]$ to address the discrete case, where the lattice sites (or unit cells) are labelled by an integer $n$. Those expressions generalize straightforwardly to higher dimensions. In both cases, we will refer to $H = H(x, k)$ as the *symbol* of $\hat{H}$. It is a reduced operator acting only on the internal degrees of freedom of the systems, but parametrized in phase space. Similarly, zero-modes can be represented in phase space by a Wigner transform of their density matrix, leading schematically to the red and blue spots in figure 3. The mapping of the Hamiltonian $\hat{H}$ into a symbol Hamiltonian $H(x, k)$ allows us to express the chiral index as a generalized winding number, given by an integral over the $2D - 1$-dimensional shell in phase space

$$
\mathcal{I} \underset{\text{S-C lim}}{=} \frac{-2(D)!}{(2D)!(2i\pi)^D} \int_{\text{shell}} \text{Tr}^{\text{int}}(U^\dagger dU)^{2D-1} \equiv \mathcal{W}_{2D-1}
\tag{14}
$$

---

[3]This derivation can be performed in infinite systems since the cut-off operator makes the trace finite.

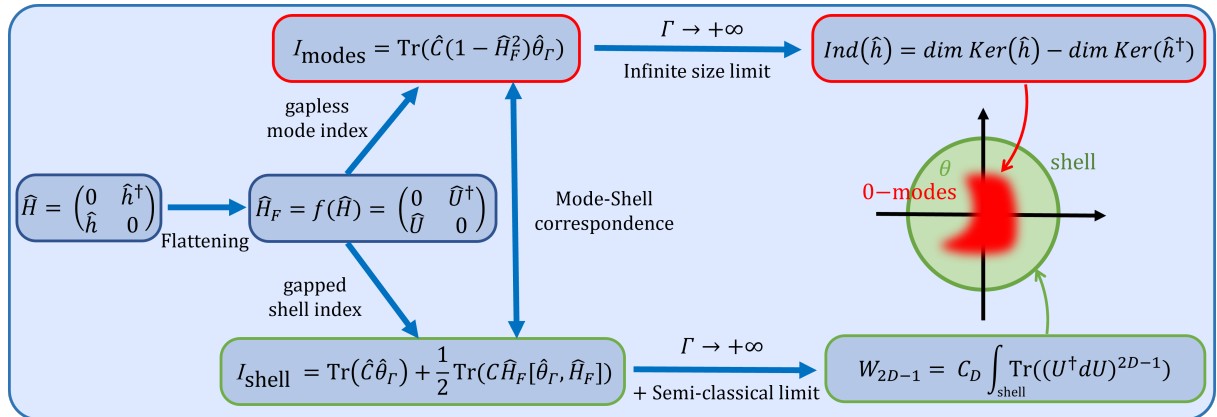

Figure 5: Summary diagram of the mode-shell correspondence. We use a smoothly flatten version $\hat{H}_F$ of the Hamiltonian $\hat{H}$ to define two indices: $I_{\text{modes}}$ counting the chiral number of zero-modes localised in a target gapless region in phase space, a gapless property of $\hat{H}$, and $I_{\text{shell}}$ measuring gapped properties on the boundary enclosing the gapless region (namely the shell) and which reduces, in a semi-classical limit, to a (higher) winding number. Both indices are equal due to the mode-shell equality (11). The prefactor $C_D$ of $W_{2D-1}$ is given in (14).

where $D$ is the dimension of the system, the trace $\text{Tr}^{\text{int}}$ only acts on the internal degrees of freedom and $U(x,k)$ is a unitary operator that constitutes the off-diagonal component of the symbol of the flatten Hamiltonian $H_F = \begin{pmatrix} 0 & U^\dagger \\ U & 0 \end{pmatrix}$ that acts on the internal degrees of freedom of the system. Since, on the shell, the Hamiltonian has no gapless mode, $H_F$ has energies $E_F = \pm 1$ and can thus be written as $H_F(x,k) = H(x,k)/\sqrt{H(x,k)^2}$. The notation $dU$ denotes the 1-differential form in phase space $dU = \sum_i (\partial_{x_i} U) dx_i + (\partial_{k_i} U) dk_i$. $(U^\dagger dU)^{2D-1}$ is then a $2D-1$-form which is an antisymetrised sum of all possible orders of derivatives $\partial_{z_j} U$ in the product $(U^\dagger dU)^{2D-1}$ where $z_j$ is a phase space coordinate (either position or wavenumber). For a given set of phase space coordinates $(z_1, \ldots, z_{2D-1})$, the related component of the form reads

$$\sum_{j_1,\cdots,j_{2D-1}=1}^{2D-1} \epsilon_{j_1,\cdots,j_{2D-1}} \prod_{m=1}^{2D-1} \left( U^\dagger \partial_{z_{j_m}} U \right) \tag{15}$$

where $\epsilon_{j_1,\cdots,j_{2D-1}}$ is the antisymetrised Levi-Civita tensor with convention $+1$ for the basis $\prod_i dk_i \wedge dx_i$. For example, when $D = 2$, the phase space has 4 dimensions. The form $\text{Tr}\left( U^\dagger dU \right)^3$ is then a 3-form with 4 components in phase space, which reduces to one component when projecting on the shell. As an instance, the component in $dx \wedge dk_x \wedge dk_y$ reads

$$\text{Tr}\left( U^\dagger \partial_x U [U^\dagger \partial_{k_x} U, U^\dagger \partial_{k_y}] + [U^\dagger \partial_{k_x} U, U^\dagger \partial_{k_y}] U^\dagger \partial_x U \right), \tag{16}$$

and similarly for the three others components. We provide an explicit demonstration of the formula (14) in appendix F.

The formula (14) can be seen as a generalization of the bulk-edge correspondence. When dealing with bounded one-dimensional ($1D$) lattices with open boundary conditions, $\mathcal{I}$ can be seen as an *edge index* that counts the chirality of the zero-modes at one boundary, while $\mathcal{W}_1$ is the usual *bulk* winding number expressed as an integral over the $1D$ Brillouin zone in $k$-space. However, the formula

(14) describes a much richer class of chiral systems that goes well beyond $1D$ lattices. Indeed, the system of interest can be of higher dimension, discrete or continuous, bounded or unbounded, and the zero-modes characterized by (14) can be localized in position (such as edge states), but also in wavenumber space.

The surface of integration, i.e. the shell, is a surface of dimension $2D-1$ that encloses the chiral zero-mode in phase space of dimension $2D$. The shell is therefore always a surface of odd dimension. This contrasts the celebrated classification of topological insulators where the chiral symmetric class (AIII) is known to allow topologically non-trivial phases in odd dimensions only [15]. The fact that our formula (14) predicts the existence of chiral zero-modes also in even dimension $D$ is because the shell lives in phase space, and is therefore not restricted to the $k$-space Brillouin zone.

The formula (14) also includes other previously existing results in topological physics that differ from the standard bulk-edge correspondence. It includes for example the formula derived by Atyiah and Singer in the 60s [102] for continuous operators when the position manifold is a torus [4] and where the shell is therefore the unit sphere in wavenumber space tensored with the manifold in position space $(x, k) \in \mathbb{T}^d \times \mathbb{S}^{d-1}$. Our formula also includes the formula proposed by Teo and Kane to classify topological point defects zero-modes [48]. In that case, the shell consists of the sphere enclosing the zero-modes in position space tensored with the Brillouin zone $(x, k) \in \mathbb{S}^{d-1} \times \mathbb{T}^d$. Finally it also includes the Callias index formula [34, 90] (also derived by Hornander [91] generalising a result by Fedosov [92]) which deals with defects localised in position space, as in the Teo and Kane's work, but for continuous operators, and where the shell is then the phase space sphere $(x, k) \in S^{2d-1}$ (localised in position and wavenumber). Our general formula (14) thus unifies all these results. The generality of the formula makes it more flexible and covers for examples the cases with both continuous and discrete dimensions, which would not fit into any of the previously cited theories.

Note that an equivalent expression of the winding number in (14), can be obtained by homotopy in terms of $h(x, k)$, the symbol of $\hat{h}$, as

$$\mathcal{W}_{2D-1} = \frac{-2(D)!}{(2D)!(2i\pi)^D} \int_{\text{shell}} \text{Tr}^{\text{int}}(h^{-1}dh)^{2D-1} \ . \tag{17}$$

This expression could be of practical interest since it bypasses the computation of $H_F$.

Finally, we should note that the formula (14) is obtained in a certain *semi-classical limit*, hence the subscript "S-C lim" (we shall just write *lim* in the rest of the paper). This limit is reached when the variations of the symbol in position $x$ or in wavenumber $k$ become small compared to the gap of the symbol. This hypothesis can be stated as follow (see appendix (B) for justification):

**Semi-classical hypothesis:** *For a given symbol $H(x, k)$, its characteristic variation distances in position $d_x$ and wavenumber $d_k$ spaces can be estimated through the formula*

$$1/d_{x/k} \sim \|\partial_{x/k} H(x, k)\|/\Delta(x, k) \tag{18}$$

*where $\Delta(x, k)$ is the gap of the symbol $H(x, k)$. The semi-classical limit is reached asymptotically near the shell when $\epsilon \equiv 1/(d_x d_k) \ll 1$.*

For example, in $1D$ lattices, the symbol of the Hamiltonian becomes completely independent of position in the bulk, so that $1/d_x \to 0$. In most of the examples treated here, we will have

---

[4]This restriction comes from simplifying hypotheses in the semi-classical expansion. Manifold with curvature lead to more complex expressions which would require a separate paper.

$\epsilon = 1/(d_x d_k) = O(1/\Gamma)$. In other words, (at least) one of the characteristic distances of variation becomes small for points $(x, k)$ in phase space which are close to the shell. Hence, the semi-classical approximation becomes exact in the asymptotic limit $\Gamma \to +\infty$. This semi-classical approximation makes the winding number $\mathcal{W}_{2D-1}$ in general simpler to calculate than the original chiral index $\mathcal{I}$, making the formula (14) of practical interest. All those results are recapped in figure 5.

## 3   Mode-shell correspondences in $1D$ spaces

### 3.1   The bulk-edge correspondence for $1D$ unbounded chiral lattices

**General results**

In this section, we discuss the particular case of Hamiltonians on $1D$ lattices with edges and show how the usual winding number is obtained as a semi-classical approximation of the shell index and therefore counts the number of chiral zero energy edge states: a result known as the *bulk-edge correspondence*, which is well established for $1D$ lattices, both physically and mathematically [99, 100, 103–108]. This derivation will serve as a pedagogical example to introduce a few key tools and concepts in more details. We shall also treat in parallel the case of *interface* zero-modes, in contrast with *edge* modes. We will therefore assume that the gapless target region is either an edge, or an interface, located at $x \sim 0$, so that the cut-off operator can be chosen as $\hat{\theta}_\Gamma = e^{-x^2/\Gamma^2}$. The chirality of zero-modes localised in that region is given by the shell index (9) with that specific cut-off operator. Let us now show how, under some assumptions, a semi-classical approximation of this index is made possible and yields a more familiar and simpler expression.

In the following, $n \in \mathcal{L}$ is the unit cell index of the lattice, it runs over $\mathcal{L} = \mathbb{N}$ if we deal with a lattice with an edge and over $\mathcal{L} = \mathbb{Z}$ in the case of an interface. We also introduce $\alpha$ to label the (finite) internal degrees of freedom (e.g. orbital, spin...). We assume the chiral operator $\hat{C}$ to be diagonal in the $(n, \alpha)$ basis and independent of the unit cell, and denote by $C_\alpha$ the chirality of the internal degrees of freedom. We then use the discrete Weyl transform (13) where $\langle n' | \hat{H} | n \rangle$ is the matrix containing the couplings between the internal degrees of freedom of the unit cells $n$ and $n'$. The symbol Hamiltonian $H(n, k)$ we obtain thus acts only on the internal degrees of freedoms, with parameters $(n, k) \in \mathcal{L} \times S^1$ living on the discrete phase space. In some sense, this discrete Wigner transform can be seen as a generalisation of the Bloch transform to non-periodic couplings on a grid.

We then make the following hypothesis: we assume that the Hamiltonian $\hat{H}$ is asymptotically periodic far from the boundary/interface. More precisely, in the case of an edge ($\mathcal{L} = \mathbb{N}$), we assume that the symbol Hamiltonian $H(n, k)$ converges asymptotically to a *bulk*, (i.e. position independent) Hamiltonian $H^+(k)$ when $n \to +\infty$. Similarly, in the case of an interface ($\mathcal{L} = \mathbb{Z}$), we ask that the symbol Hamiltonian converges toward two bulk Hamiltonians far to the left/right of the interface, that is $H(n, k) \to H^\pm(k)$ when $n \to \pm\infty$.[5]

Let us now estimate the term $\text{Tr}\,\hat{C}\hat{H}_F[\hat{\theta}_\Gamma, \hat{H}_F]$ of the chiral index, with $\hat{\theta}_\Gamma = e^{-x^2/\Gamma^2}$ in the limit $\Gamma \to +\infty$. For that purpose, we first rewrite the trace as an integral in phase space by using the

---

[5]In general, we only need the weaker assumption $\frac{1}{d_x d_k} \to 0$, as defined in (18), to obtain a valid semi-classical limit which is useful in some cases.

Moyal $\star$ product between symbols as

$$\operatorname{Tr} \hat{A}\hat{B} = \frac{1}{2\pi} \sum_n \int_0^{2\pi} dk \operatorname{Tr}^{\mathrm{int}}(A \star B)(n, k) \tag{19}$$

where $\operatorname{Tr}^{\mathrm{int}}$ is the trace on the internal degrees of freedom only (see appendix B). We obtain

$$\frac{1}{2} \operatorname{Tr} \hat{C}\hat{H}_F[\hat{\theta}_\Gamma, \hat{H}_F] = \frac{1}{4\pi} \sum_{n \in \mathcal{L}} \int_0^{2\pi} dk \ \operatorname{Tr}^{\mathrm{int}}(C \star H_F \star [\theta_\Gamma, H_F]_\star)(n, k) \tag{20}$$

where $[A, B]_\star = A \star B - B \star A$ is the Moyal commutator. Next we take the limit $\Gamma \to +\infty$. As discussed in the previous section, in that limit, $\theta_\Gamma \approx \mathbb{1}$ near the interface/boundary, $\mathcal{I}$ is the topological index describing the chiral number of the zero-modes localised at the interface/boundary. Moreover as $\Gamma \to +\infty$, $\theta_\Gamma(n)$ varies slower and slower with $n$, so that we probe a region which is further and further in the bulk where $H(n, k)$ has asymptotically no dependence in position, by hypothesis. The product of the symbols $H(n, k)$ with $\theta_\Gamma(n)$ is therefore prone to a semi-classical approximation, obtained in the limit $\Gamma \to +\infty$.

The leading term of such a semi-classical expansion is obtained by simply replacing all the Moyal products by standard product $A \star B \sim AB$, and the Moyal commutator by a Poisson bracket $[A, B]_\star \sim i\{A, B\}$ (see Appendix B), so that

$$\frac{1}{2} \operatorname{Tr}\left(\hat{C}\hat{H}_F[\hat{\theta}_\Gamma, \hat{H}_F]\right) = \frac{-1}{4i\pi} \sum_{n \in \mathcal{L}'} \int_0^{2\pi} dk \operatorname{Tr}^{\mathrm{int}}(\ CH_F(n, k)\delta_n\theta_\Gamma(n)\partial_k H_F(n, k)) + O(1/\Gamma) \tag{21}$$

where $\delta_n\theta_\Gamma(n, k) = \theta_\Gamma(n + 1, k) - \theta_\Gamma(n, k)$ is the discrete derivative. Note that we do not have the term $\partial_k\theta_\Gamma(n)\delta_n H_F(n, k)$ in the Poisson bracket because $\theta_\Gamma(n)$ has no dependence $k$. As we will see, this first term of the semi-classical expansion converges already to a finite constant when $\Gamma \to +\infty$. So, the next term of the semi-classical expansion, which must be of smaller order in $\Gamma$, vanishes when $\Gamma \to +\infty$ and there is no need to consider them. We will use the notation $\underset{\mathrm{lim}}{=}$ to mean that an equality is true up to the vanishing of higher order terms in the limit $\Gamma \to +\infty$. Then, since the variation of $\theta_\Gamma(n)$ mainly comes from the high $|n| \gg 1$ region, we can approximate $H_F(n, k)$ by its bulk limit.

Let us focus first on the interface case ($\mathcal{L} = \mathbb{Z}$). We substitute $H_F(n, k)$ by $H_F^+(k)$ for $n > 0$ and by $H_F^-(k)$ for $n < 0$ leading to

$$\frac{1}{2} \operatorname{Tr}(\hat{C}\hat{H}_F[\hat{\theta}_\Gamma, \hat{H}_F]) \underset{\mathrm{lim}}{=}$$
$$\frac{-1}{4i\pi} \int_0^{2\pi} dk \operatorname{Tr}^{\mathrm{int}}\left(C\left(H_F^+(k) \sum_{n>0} \delta_n\theta_\Gamma(n)\partial_k H_F^+(k) + H_F^-(k) \sum_{n<0} \delta_n\theta_\Gamma(n)\partial_k H_F^-(k)\right)\right). \tag{22}$$

The sum over $n$ is performed by using $\sum_{n>0} \delta_n\theta_\Gamma(n) = \theta_\Gamma(+\infty) - \theta_\Gamma(0) = -1$ and $\sum_{n<0} \delta_n\theta_\Gamma(n) = \theta_\Gamma(0) - \theta_\Gamma(-\infty) = 1$, and we obtain

$$\frac{1}{2} \operatorname{Tr}\left(\hat{C}\hat{H}_F[\hat{\theta}_\Gamma, \hat{H}_F]\right) \underset{\mathrm{lim}}{=} \frac{1}{4i\pi} \int_0^{2\pi} dk \operatorname{Tr}^{\mathrm{int}}(\ C\left(H_F^+(k)\partial_k H_F^+(k) - H_F^-(k)\partial_k H_F^-(k)\right)). \tag{23}$$

Since $H_F^2 = \mathbb{1}$ in the bulk, we can introduce the unitaries $U^\pm$ such that

$$H_F^\pm = \begin{pmatrix} 0 & (U^\pm)^\dagger \\ U^\pm & 0 \end{pmatrix} \tag{24}$$

and rewrite (23) as

$$\frac{1}{2}\operatorname{Tr}\left(\hat{C}\hat{H}_F[\hat{\theta}_\Gamma, \hat{H}_F]\right) \underset{\lim}{=} \frac{1}{2i\pi}\int_0^{2\pi}dk\,\operatorname{Tr}^{\mathrm{int}}\left((U^+)^\dagger(k)\partial_k U^+(k) - (U^-)^\dagger(k)\partial_k U^-(k)\right) \tag{25}$$

where we recognize the winding number $W \equiv \frac{1}{2i\pi}\int_0^{2\pi}dk\,\operatorname{Tr}U^\dagger\partial_k U \in \mathbb{Z}$ of the unitary map $k \in S^1 \to U(k) \in S^1$, which leads to

$$\frac{1}{2}\operatorname{Tr}\left(\hat{C}\hat{H}_F[\hat{\theta}_\Gamma, \hat{H}_F]\right) \underset{\lim}{=} W_\uparrow^+ - W_\uparrow^- \tag{26}$$

with $W_\uparrow^+$ and $W_\uparrow^-$ the winding numbers of $U^+$ and $U^-$ defined in the bulks far to the positive and negative sides of the interface respectively, and integrated over the $1D$ Brillouin zone. The vertical arrow $\uparrow$ specifies the direction of integration in $k$, from 0 to $2\pi$.

If the lattice has "balanced unit cells", that is when there is an equal number of degrees of freedom of positive and negative chirality $C_\alpha$ per unit cell $n$ ($\sum_\alpha C_\alpha = 0$), this imply that the terms $\operatorname{Tr}\left(\hat{C}\hat{\theta}_\Gamma\right) = \sum_n \sum_\alpha C_\alpha \theta_\Gamma(n) = 0$. Therefore, in that case, one recovers the expected bulk-interface correspondence for $1D$ chiral chains in the limit $\Gamma \to +\infty$

$$\mathcal{I} \underset{\lim}{=} W_\uparrow^+ - W_\uparrow^- \ . \tag{27}$$

which is a particular case of our general formula (14) that we derive through relatively simple consideration.

Otherwise, if the unit-cell structure is broken at the boundary, this equality must be corrected by the term $\operatorname{Tr}\hat{C}\hat{\theta}_\Gamma$ to account for the chirality of the lattice's sites [99–101]. The term $\operatorname{Tr}\hat{C}\hat{\theta}_\Gamma$ is also non-zero when the bulk unit cell is unbalanced in chirality $\sum_\alpha C_\alpha \neq 0$. However, this case is excluded from our theory because it leads to bulk zero-modes that violate the gap hypothesis (see appendix D).

The case of an edge, rather than an interface, is obtained similarly. The only difference being that the sum in $n$ runs now over $\mathcal{L} = \mathbb{N}$ (for a left edge) instead of $\mathbb{Z}$. As a consequence, the second term in the right hand side of the equation (22) is missing, and we end up with the bulk-edge correspondence

$$\mathcal{I} \underset{\lim}{=} W_\uparrow^+ \tag{28}$$

that relates the chirality of zero-energy edge modes, at a given edge, to a winding number in the bulk of the lattice.

We now illustrate this approach on the seminal example of the dimerized chain: the so-called Su–Schrieffer–Heeger model.

**Example: The Su-Schrieffer-Heeger (SSH) chain**

A seminal example of a $1D$ chiral symmetric lattice model exhibiting zero-energy edge modes, is that of a $1D$ dimerised chain, often referred to as the Su-Schrieffer-Heeger (SSH) model [109] (see figure 6) even though there is an overlap with other types of dimerised model, like the Shockley chain [110, 111]. In any case, the unit cell owns two internal degrees of freedom denoted A and B (being the even/odd sites $n$ in the SSH case), and the model consists of nearest neighbour staggered couplings of amplitude $t$ and $t'$ between A and B. Let us revisit this celebrated SSH/Shockley model

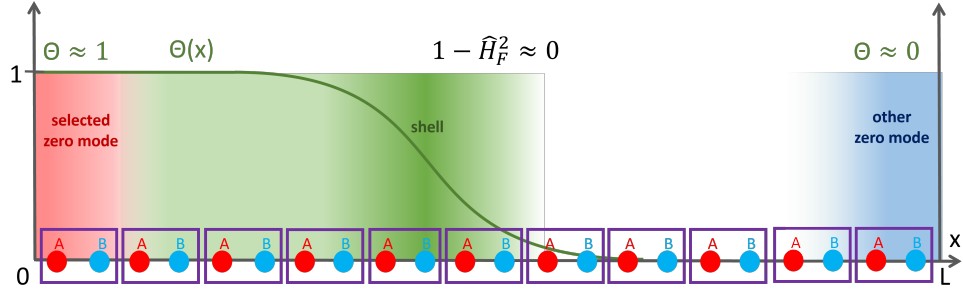

Figure 6: Representation of a SSH chain of $N = 10$ unit cells delimited in purple. The left red edge is the gapless region where we want to compute the chiral number of zero-modes. The right blue edge is the other gapless region of opposite chirality that is dismissed through the cut-off function $\theta(x)$ in green. The dark green zone is the shell where is evaluated the bulk-index $\frac{1}{2}\text{Tr}\big(\hat{C}\hat{H}_F[\hat{\theta}_\Gamma, \hat{H}_F]\big)$ since the coefficients of the trace quickly vanish away from it.

in the light of the mode-shell correspondence. In fact, this model is simple enough to be analytically solvable, and we will thus be able to derive the bulk-edge correspondence explicitly.

The corresponding Hamiltonian $\hat{H} = \hat{H}_{\text{SSH}}$ reads

$$\hat{H} = \sum_n t\,|B,n\rangle\langle A,n| + t'\,|B,n-1\rangle\langle A,n| + h.c. \tag{29}$$

except at the edges where the hopping term $t'$ leads to an empty site outside the lattice. In that case, it is put to zero (open boundary condition). Since this Hamiltonian only couples $A$ sites with $B$ sites, it is chiral symmetric and the chiral operator reads $\hat{C} = \sum_n |A,n\rangle\langle A,n| - |B,n\rangle\langle B,n|$. We can therefore define a chiral index $\mathcal{I}$. Then, far in the bulk, the Hamiltonian is invariant by translation and the Wigner-Weyl transform reduces to a discrete Fourier transform where

$$H(n,k) = \begin{pmatrix} 0 & t + t'e^{-ik} \\ t + t'e^{ik} & 0 \end{pmatrix} = H(k) \tag{30}$$

and whose energy spectrum $E$ is gapped for $t \neq t'$. Next, we want to compute the "flatten" version of the symbol, $H_F$. To do so, we use the fact that, at first order of the semi-classical expansion, the symbol of $\hat{H}_F = f(\hat{H})$ is simply given by applying directly the function $f$ to the symbol $H(n,k)$, that is $H_F = f(H(k))$. Moreover, we have chosen $f$ such that, for gapped states of energy $E$, we have $f(E) = E/\sqrt{E^2}$ so, in the bulk, $H_F(x,k) = H(x,k)/\sqrt{H(x,k)^2}$. Therefore, since $H^2(n,k) = |t + t'e^{ik}|^2\,\mathbb{1}$, we deduce that

$$H_F(k) = \frac{1}{|t + t'e^{ik}|} \begin{pmatrix} 0 & t + t'e^{ik} \\ t + t'e^{-ik} & 0 \end{pmatrix}. \tag{31}$$

This allows us to identify $U = (t + t'e^{ik})/|t + t'e^{-ik}|$ which is just a unit complex number here. A direct computation of the winding number $W_\uparrow = \frac{1}{2i\pi}\int_{k\in S^1} dk\,\text{Tr}^{\text{int}}(U^\dagger(k)\partial_k U(k))$ yields $W_\uparrow = +1$ for $|t'| > |t|$ and $W_\uparrow = 0$ for $|t'| < |t|$.

We now turn to the computation of zero-modes localized at a single edge. We thus assume the lattice to be semi-infinite, with no boundary to the right and a left boundary at $n = 0$.

The zero-modes of this model can be analytically found by searching them of the form $|\psi\rangle = \sum_{n\geq 0} \psi_{A,n} |A,n\rangle + \psi_{B,n} |B,n\rangle$ such that $\hat{H}|\psi\rangle = 0$. Combined with the boundary condition $\langle B,-1|\psi\rangle = 0$, we obtain the constraints

$$
\begin{aligned}
\langle B,n|\hat{H}|\psi\rangle = t\psi_{A,n} + t'\psi_{A,n+1} = 0 && n \geq 0 \\
\langle A,n|\hat{H}|\psi\rangle = t\psi_{B,n} + t'\psi_{B,n-1} = 0 && n > 0 \\
\langle A,0|\hat{H}|\psi\rangle = t\psi_{B,0} = 0 && n = 0 \ .
\end{aligned}
\tag{32}
$$

If we remove the pathological case $t' = 0$, this system implies $\forall n, \psi_{B,n} = 0$ and $\psi_{A,n} = \left(\frac{-t}{t'}\right)^n \psi_{A,0}$.

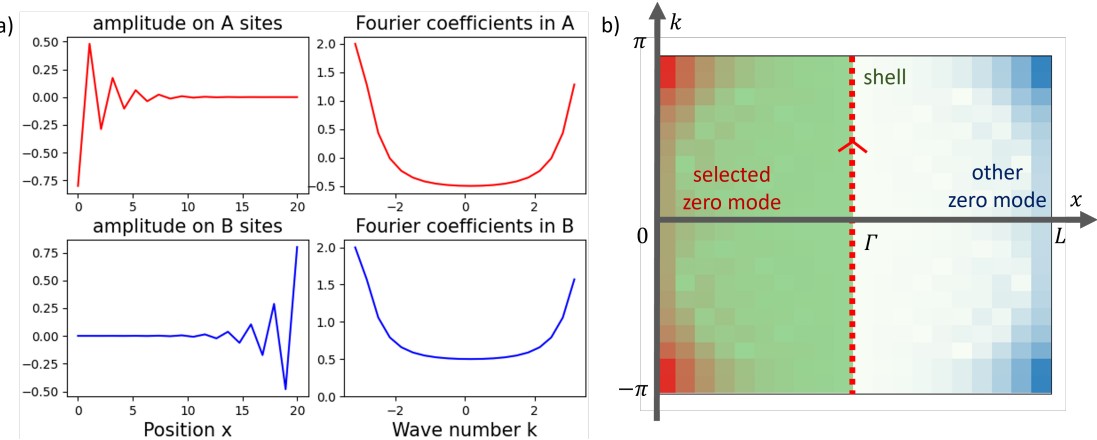

Figure 7: (left) Plot of the topological zero-modes of an SSH chain in real and Fourier space with $t' = 1$ and $t = 0.6$ for $N = 10$ dimers. (right) The absolute value of the Wigner-Weyl transform is plotted for the same edge mode in phase space. The region selected by the cut-off is shown in green, and its boundary (the shell), of length $\Gamma$ in real space, is highlighted by a dotted line along which the winding number is integrated.

To correspond to an edge mode, this solution must be normalized, which is only possible when $|t'| > |t|$. We deduce that one zero-energy edge mode of positive chirality (i-e: localised on the A sites only) exists for $|t'| > |t|$, leading to $\mathcal{I} = 1$, while no edge mode exists when $|t'| < |t|$, leading to $\mathcal{I} = 0$. As a result, in both cases we can check that $\mathcal{I} = W_\uparrow$ which is an illustration of the bulk-edge correspondence in a simple but non-trivial example.

**Validity of the semi-classical limit**

Since the SSH model is invariant by translation far in the bulk, we have $1/d_x \to 0$ on the shell when $\Gamma \to +\infty$. Besides, as $1/d_k$ remains bounded because $\hat{H}$ is short-range in position, it implies $1/(d_x d_k) \to 0$ and therefore the semi-classical limit becomes exact in the limit $\Gamma \to +\infty$.

## 3.2 A dual bulk–boundary correspondence in wavenumber space for bounded continuous systems

**General results**

In the previous section, we focused on $1D$ discrete lattices and discussed an example where the zero-modes are related to a winding number on a shell defined along the $k$ axis at large $x$, away from the zero-mode. In the large distance limit the lattice can be seen as infinite, and the different zero-modes, localized at opposite boundaries decouple, can be treated separately. Continuous systems are an other kind of systems with an infinite number of degrees of freedom. This infinity does not come from the the size of the system, but instead from the distance between two sites/degrees of freedom that becomes infinitesimal (see figure 4). At the Hilbert-space level, this limit can also be seen as the fast varying functions limit or, in other words, as the large wavenumber $k \to +\infty$ limit. In this section, we discuss how we can exploit such limits to create topologically protected zero-modes which are separated, not in position, but in wavenumber, and how the mode-shell correspondence captures this situation.

We are concerned with $1D$ continuous systems, where the physical quantities are encoded in vector-valued wave-function $|\psi(x)\rangle = (\psi_\alpha(x))_\alpha$ where $x$ is a continuous coordinate and $\alpha$ labels the internal degrees of freedom. These degrees of freedom can, for example, be the spin or pseudo-spin components of quantum (quasi-)particle, like in the Dirac equation, or be a combination of classical fields, like the velocity $v(x)$ and the pressure $p(x)$ in the acoustic wave equation. As in the previous section, we assume that the time evolution of the wave function is encoded by a Hamiltonian $\hat{H}$. Because we now deal with continuous system, $\hat{H}$ is in general be differential operator which depends on position $x$ and of some of its derivatives as $\hat{H} = \sum_n h_n(x)\partial_x^n$, where $h_n(x)$ are operators acting on the internal degrees of freedom.

Similarly to the discrete case, we use a Wigner transform (12) which associates, to an operator $\hat{H}$, a symbol $H(x,k)$ parameterised in phase space and acting on the internal degrees of freedom (see Appendix B) where now $k \in \mathbb{R}$ belongs to the whole real line which is not a bounded set (contrary to the lattice case where $k$ is reduced to the Brillouin zone $[0, 2\pi]$). Therefore, the major difference with the lattice case is that there is not only the limit $x \to \pm\infty$ (i-e: far away from an interface/edge) to be considered, but also the $k \to \pm\infty$ limit of fast varying solutions. Since the limit in real space is similar to that discussed previously, we would like to focus only on the momentum limit.

For that purpose, we consider systems where the position space is bounded. Also, we choose to consider the position space as a manifold with no edges. For example, the position space could be a circle (see figure 4), a torus, a sphere, etc..., and the differential operators in the Hamiltonian $\hat{H}$ act on continuous functions defined on those manifolds. Then, if the Hamiltonian is gapped in the large wavenumber limit (i.e. when acting on fast varying functions) then one can define the chiral index $\mathcal{I}$ (8) with $\hat{\theta}_\Gamma = \exp\{\Delta/\Gamma^2\}$, which is referred to as the heat kernel associated to the Laplacian $\Delta$ on the manifold. As we already saw, this index is equal to the chirality of zero-modes through the analytical index (5). This framework is actually that discussed in the celebrated Atiyah-Singer index theorem, as it is described in the mathematical community [93, 95, 96]. Here, we focus on the $1D$ case where the underlying manifold is the circle, and derive the semi-classical winding number associated to chiral zero-modes. Note that since our position space is a circle, and not just a real line, it implies some subtleties in the definition of the symbol, the formula (12) being only valid in the real line case. But, as long as $\hat{H}$ is short range compared to the topology of the manifold, we

can always use the definition (12) in a local chart around $x$ to extend it to the circle case[6].

In order to derive the semi-classical index, we proceed similarly to the discrete case: We first express the term $\mathrm{Tr}\left(\hat{C}\hat{H}_F[\hat{\theta}_\Gamma, \hat{H}_F]\right)$ of the shell index, in phase space through the trace identity $\mathrm{Tr}\,\hat{A} = \frac{1}{2\pi}\int_{\mathbb{S}} dx \int_{\mathbb{R}} dk\,\mathrm{Tr}^{\mathrm{int}}\,A(x,k)$ with an integration in position on the circle. This operation maps the commutator of operators into the Moyal commutator of their symbols. We then take the limit $\Gamma \to \infty$ and keep the lower order term in $1/\Gamma$, which amounts to approximate the Moyal commutator by a Poisson bracket. This Poisson bracket contains only the term $\partial_k\theta_\Gamma\partial_x H_F$ because here the cut-off function $\theta_\Gamma = e^{-k^2/\Gamma^2}$ depends only on wavenumber and not on position. This leads to the expression

$$\mathcal{I}_{\underset{\mathrm{lim}}{=}} \frac{1}{4i\pi}\int_0^{2\pi} dx \int_{-\infty}^{+\infty} dk\,\mathrm{Tr}^{\mathrm{int}}\,CH_F(x,k)\partial_k\theta_\Gamma(k)\partial_x H_F(x,k)\ . \tag{33}$$

Next, we perform the integration over $k$. This is not as simple as the integration over $x$ in the discrete case where we assumed a bulk (i.e. $x$ independent) limit of the symbol Hamiltonian, since here $H_F(x,k)$ may not be totally independent of $k$. We can however use the fact that the right hand side of (33) does not depend of the special shape of $\theta_\Gamma(k)$ (see appendix F). Therefore, we can smoothly deform the cut-off function $\theta_\Gamma = \exp(-k^2/\Gamma^2)$ into the sharper one $\tilde{\theta}_\Gamma = \mathbb{1}_{|k|\leq\Gamma}$ such that the derivative $\partial_k\theta_\Gamma$ can be replaced by a $\delta$-Dirac distribution, which transforms the surface integral in phase space into two line integrals over $x$ at $k = \pm\Gamma$ as

$$\mathcal{I}_{\underset{\mathrm{lim}}{=}} \frac{1}{4i\pi}\int_0^{2\pi} dx\,\mathrm{Tr}^{\mathrm{int}}(C(-H_F(x,\Gamma)\partial_x H_F(x,\Gamma) + H_F(x,-\Gamma)\partial_x H_F(x,-\Gamma))) \tag{34}$$

$$\underset{\mathrm{lim}}{=} \frac{1}{2\pi i}\int_0^{2\pi} dx\,\mathrm{Tr}^{\mathrm{int}}\left(-U^\dagger(x,\Gamma)\partial_x U(x,\Gamma) + U^\dagger(x,-\Gamma)\partial_x U(x,-\Gamma)\right)\ . \tag{35}$$

Finally, we obtain that the chiral index is again related to a difference of winding numbers, but where the integration runs now over position space for large positive/negative wavenumbers, as depicted by horizontal dashed lines in figure 8. We will thus indicate this "horizontal" line integral in phase space by horizontal arrows, so that we get

$$\mathcal{I}_{\underset{\mathrm{lim}}{=}} -W_\rightarrow^+ + W_\rightarrow^- \tag{36}$$

where $\pm$ refers to $k = \pm\Gamma$. This is a second application of the mode-shell correspondence in $1D$ where the relation to a difference of winding number $W_\rightarrow^\pm$ is a particular case of (14). It can be seen as dual to the lattice case previously discussed, and in particular, (36) can be compared to (27). In both cases, the shells correspond to lines in a single subspace, either $x$ or $k$, and they both enclose chiral-zero modes in phase space. In the present case, those modes are "located" in

---

[6]In particular one can use the geodesic chart to describe the neighborhood of $x$ as a subset of $\mathbb{R}$ (see [94] for a more formalised definition). There is however some problem for curved manifold, the semi-classical expansion is modified in those cases. Also our proof of the semi-classical invariants in the higher-dimension case relies on the existence of operators verifying $[a_i, b_j] = \delta_{i,j}\mathbb{1}$ which can only be found when the phase space is $\mathbb{R}^d \times \mathbb{R}^d$ or $\mathbb{R}^d \times \mathbb{T}^d$. Therefore our formula (57) will only works in the case where the position manifold is a n-torus (which has no intrinsic curvature). As the general expression of the symbol index in the Atiyah-Singer theorem involves the curvature of the manifold, it is not surprising that our formula is limited to the n-torus cases which are manifolds of zero curvature. We believe there is a way to derive the general Atiyah-Singer theorem using the fact that any manifold can in fact be embedded in $\mathbb{R}^m$ where our semi-classical formula could be applied. But the derivation of the formula would go beyond the scope of this paper.

the region of small wavenumber region, while the shell, in the semi-classical limit, is considered in the region of high wavenumber. The mode-shell correspondence thus better translates here to a correspondence between different region separated in wavenumber, rather than to a *bulk-edge* or *bulk-interface* correspondence in position. We now illustrate this correspondence with an example.

**Example: $1D$ Dirac equation with periodic potential and velocity**

To illustrate the previous result, we propose the following model of a Dirac Hamiltonian on a circle

$$\hat{H}(x, \partial_x) = \begin{pmatrix} 0 & V(x) + c(x)\partial_x \\ V(x) - \partial_x c(x) & 0 \end{pmatrix} \tag{37}$$

where the potential $V(x)$ and the local velocity $c(x)$ are bounded and $L$-periodic functions that can change sign. The symbol of this operator is obtained by replacing the product of the non-commuting operators $c(x)\partial_x$ by the Moyal product of their symbol $c(x) \star ik = c(x)ik - c'(x)/2$ with $c'(x) \equiv \partial_x c(x)$, that is

$$H(x, k) = \begin{pmatrix} 0 & V(x) + ikc(x) - c'(x)/2 \\ V(x) - ikc(x) - c'(x)/2 & 0 \end{pmatrix}. \tag{38}$$

To check if the the semi-classical expression of the shell invariant can be used in the limit $\Gamma = k \to \infty$, we evaluate $1/(d_x d_k) = \|\partial_x H(x, k)\| \|\partial_k H(x, k)\| / \Delta(x, k)^2$ that varies as $\frac{1}{k} |\frac{c'(x)}{c(x)}|$ in that limit. The semi-classical criteria $1/(d_x d_k) \ll 1$ is thus reached, unless $c(x)$ vanishes, a situation we consider here. In that case, the term $1/d_k$ accidentally vanishes and we should check the next order term of the semi-classical expansion, $\|\partial_x \partial_k H/\Delta\|_{c=0} = c'(x)/(V(x) - c'(x)/2)|_{c=0}$. This term is not necessary negligible in general, and in particular when the amplitude of the potential $V(x)$ is comparable with that of the gradient of the velocity $c'(x)$ (the gap may even close). One way to make this term small, is to consider a very large periodic system $L \to \infty$ such that the gradient of velocity can be re-expressed as $c'(x') = c'(x)/L$ after the rescaling $x = Lx'$, and becomes negligible compared to $V(x')$. The small parameter $\epsilon \equiv 1/L$ thus controls the semi-classical limit, when $V(x)$ and $c'(x)$ are comparable in amplitude. By setting $\Gamma = k \sim 1/\epsilon \to \infty$, one gets the semi-classical symbol

$$H(x, k) \simeq \begin{pmatrix} 0 & V(x) + ikc(x) \\ V(x) - ikc(x) & 0 \end{pmatrix} \tag{39}$$

(where we have substituted the notation $x' \to x$) which is indeed uniformly gapped in phase space and satisfies $1/(d_x d_k) \ll 1$ when $k \to \infty$. From now, we can set $V(x) = \cos(x)$ and $c(x) = \sin(x)$ in the operator (37), and get a semi-classical description in the large system limit $L \gg 1$ with (39), where the shell invariant can be expressed as the winding number $W_\to$. Note that the term $c'(x)$ may be kept in the symbol of other models, in particular if $c(x)$ is not allowed to vanish, as encountered for instance in [28, 112], and where the topological number must be extracted from a slightly more involved symbol Hamiltonian than cannot be obtained from the operator by simply substituting $x \to x$ and $\partial_x \to ik$.

To compute the winding numbers $W_\to$ for the model above, we introduce the shell $\{(x, k), x \in [0, 2\pi], k = \pm\Gamma\}$ which consists in two circles in position at fixed $k = \pm\Gamma$ in a cylindrical phase space. The off diagonal component of the Hamiltonian is just $h(x, k) = \cos(x) - i\epsilon k \sin(x)$ so one can compute the winding numbers and we find $W_\to^+ = -1$ for positive $k$ and $W_\to^- = 1$ for negative $k$

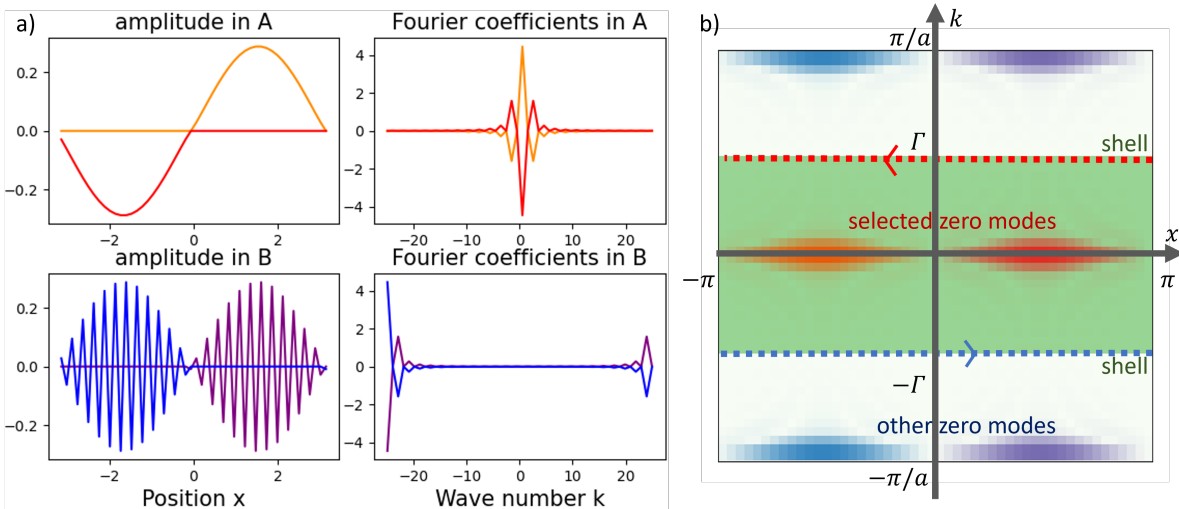

Figure 8: a) Zeros-modes of a discretised version of (37) where continuous derivatives are replaced by their discrete counterparts, with $\epsilon = 1/L = 0,05$ and $N = 50$ sites (containing each two degrees of freedoms A/B) and therefore an inter-site distance of $a = 2\pi/N$. The zero-modes are shown in real space (left) and in wavenumber space (right). Those modes are stationary with an exponentially small error $\|\hat{H}|\psi\rangle\| \sim 10^{-9}$. b) Absolute value of the Wigner-Weyl transform of the different zero-modes in phase space. The integration contours of the winding numbers correspond to the shell enclosing the low wavenumber zero-modes and are denoted by two dotted lines.

(for $\epsilon > 0$). Therefore, according to the mode-shell correspondence the total chirality of zero-modes of $\hat{H}$ should be $\mathcal{I} = -(-1) + 1 = 2$. One thus expects the operator $\hat{H}$ to have at least two zeros-modes of positive chirality in the relatively slowly varying region. This is indeed the result obtained for numerical simulations of the model (see Figure 8), where we indeed find 2 zero-modes of positive chirality (i.e. fully polarized on the A degrees of freedom) and located in the low-wavenumber $k \sim 0$ region. Due to the discretisation procedure when solving the model numerically, we moreover find two other zero-modes localised at high wavenumber. Those additional zero-modes have together a chirality of $-2$ (i.e. fully polarized on the B degrees of freedom) that balance the total chirality of zero-modes in this discretised version of the model.

The existence of such topological modes on a manifold of odd dimension (here the 1D-circle) can be surprising for someone familiar with the Atyah-Singer index theory, where it is known that elliptic operators on odd manifolds have a trivial topology. We explain why this fact is actually not incompatible with our result in appendix E.

**Remarks on the protection of zero-modes separated in wavenumber space**

It is clear that for the finite size topological insulators, such as the SSH chain, boundary modes localised at opposite edges can hybridise. The coupling between those modes can however be negligible whenever the lattice is sufficiently long, that is, more precisely, when the characteristic distance $d_x$ of the coupling elements $\hat{H}_{x,x'}$ in position space remains much smaller than the size $L$ of the lattice.

Similarly to the example discussed above, the discretisation of a continuous model typically

induces multiple gapless modes which are separated from each other in wavenumber. Using the duality between wavenumber and position space we can translate the previous criteria of weak hybridization into wavenumber space, by demanding that the couplings in wavenumber space $\hat{H}_{k,k'}$ are short-range and decays with a characteristic distance $d_k$ in wavenumber which is much smaller than the lattice wavenumber $k_0 = 2\pi/a$ of the lattice (where $a$ is the lattice spacing). Using the Wigner-Weyl transform, this is equivalent to demand that the symbol Hamiltonian $H(x, k)$ varies slowly in position space (see appendix A and B): its typical variations must evolve over a much larger distance than the inter-site spacing.

One should note that this condition for the non-hybridization of the zero-modes in wavenumber space is quite different from the position case, and may be difficult to reach in practice, depending on the physical context of interest. For example, in condensed matter systems, the introduction of an impurity or a vacancy in the lattice induces variations of the electronic potential over a characteristic distance equivalent to the size of the lattice and immediately hybridises edge states separated in wavenumber, and thus gap them [113]. Therefore, condensed matter applications would require a strict limitation of such impurities. In other physical systems, like in fluid mechanics or in acoustics, the smooth variation of the system's parameters in space is probably more naturally realised due to local homogenisation.

## 3.3 A mixed $x - k$ correspondence in phase space for unbounded continuous $1D$ systems

In the previous sections, we explained how the topological nature of chiral zero-modes is revealed by isolating them through large gapped regions which surround them either in position (case of unbounded $1D$ lattices) or in wavenumber (case of bounded continuous $1D$ systems). In the present section, we want to address the mixed case where the modes are surrounded by a gap region both in position and momentum directions.

For that purpose, let us consider unbounded $1D$ continuous systems. We will make use of the continuous Wigner transform (12) to map the Hamiltonian $\hat{H}$ to the symbol $H(x, k)$ acting on internal degrees of freedom, and parameterised in phase space $(x, k) \in \mathbb{R} \times \mathbb{R}$ (see Appendix B). We therefore have to deal with both limits $x \to \pm\infty$ (i.e. far away from an interface hosting zero-modes) and $k \to \pm\infty$ (i.e. fast varying solutions). We thus consider a mixed cut-off operator such as $\hat{\theta}_\Gamma = e^{-(x^2 - \partial_x^2)/\Gamma^2}$ of symbol $\theta_\Gamma(x, k) \approx e^{-(x^2 + k^2)/\Gamma^2}$ at first order of the semi-classical expansion. Now, the gap hypothesis means that we assume the symbol $H(x, k)$ to be gapped both when $|x| \to \infty$ and when $|k| \to +\infty$ (even for $x$ near the interface). For example, $H(x, k) = \begin{pmatrix} 0 & x+ik \\ x-ik & 0 \end{pmatrix}$ satisfies such requirement since its spectrum $\pm\sqrt{x^2 + k^2}$ converges uniformly toward infinity for both $x \to \pm\infty$ and $k \to \pm\infty$.

We can then derive the semi-classical expression of the chiral invariant by rewriting the term $\text{Tr}\left(\hat{C}\hat{H}_F[\hat{\theta}_\Gamma, \hat{H}_F]\right)$ similarly to the two previous sections (the term $\text{Tr}\,\hat{C}\hat{\theta}_\Gamma$ vanish to preserve the gap assumption if we have a balanced number of degrees of freedom), that is by turning the trace into an integral over phase space and then expanding to lowest order in $1/\Gamma$ by assuming that $H_F(x, k)$ varies slowly for large $|(x, k)|$, which leads to

$$\mathcal{I} \underset{\lim}{=} \frac{-1}{4i\pi} \int_\mathbb{R} dx \int_\mathbb{R} dk \, \text{Tr}^{\text{int}}(CH_F(\partial_x\theta_\Gamma\partial_k H_F - \partial_k\theta_\Gamma\partial_x H_F)) \ . \tag{40}$$

Note that all the terms of the Poisson bracket appear, in contrast with the winding numbers previously derived in sections 3.1 and 3.2. If we denote by $dA = \partial_x A\,dx + \partial_k A\,dk$ the differential

one-form of the symbol $A$, the expression (40) can be written in a more compact fashion as

$$\mathcal{I} \underset{\lim}{=} \frac{-1}{4i\pi} \int_{\mathbb{R}^2} \mathrm{Tr}^{\mathrm{int}}(CH_F d\theta_\Gamma \wedge dH_F) \tag{41}$$

where $\wedge$ is the usual anti-symmetric wedge product. Moreover since $H_F(x,k)$ is assumed to vary slowly, the integration of $d\theta_\Gamma$ can be done independently. The integration on the two-form is then reduced to the integration of a one-form on the circle of radius $\Gamma$, which is tangent to the gradient of $\theta$. This leads to the final result

$$\mathcal{I} \underset{\lim}{=} \frac{1}{4i\pi} \int_{S^1(\Gamma)} \mathrm{Tr}^{\mathrm{int}}(CH_F dH_F) \tag{42}$$

$$\underset{\lim}{=} \frac{1}{2i\pi} \int_{S^1(\Gamma)} \mathrm{Tr}^{\mathrm{int}}(U^\dagger dU) \equiv W_\circlearrowleft \tag{43}$$

which is again a winding number, but where the integration runs now over the circle $x^2 + k^2 = \Gamma^2$ in phase space instead of the Brillouin zone $k \in [0, 2\pi]$ (for discrete unbounded systems) or the position space $x \in [0, 2\pi]$ (for continuous circular systems). This is therefore a different semi-classical manifestation of the mode-shell correspondence, where the circle encloses the zero-mode in phase space.

**Example: The Jackiw-Rebbi model**

The simplest example of a continuous $1D$ Hamiltonian operator $\hat{H}$ involving both $x$ and $\partial_x$ which is topological is given by the celebrated Jackiw-Rebbi model

$$\hat{H} = \begin{pmatrix} 0 & x - \partial_x \\ x + \partial_x & 0 \end{pmatrix} . \tag{44}$$

This Hamiltonian can be thought of as a one dimensional Dirac Hamiltonian $\hat{H} = -i\partial_x \sigma_y$ with a linearly varying potential $V(x)\sigma_x$ that can be seen as a mass term.[7] Such a Hamiltonian can for instance be obtained in stratified and/or compressible fluids where the pressure and velocity are additionally coupled through an acoustic-buoyant frequency $S(x) = V(x)$ [28, 41] which changes sign in space. This coupling can have many origins, for example in fluids, where the sound velocity varies in space and reaches a minimum. We will also see later that this Hamiltonian can be obtained as a continuous version of an SSH model with slowly varying couplings.

The Hamiltonian (44) is easily diagonalizable by introducing the bosonic creation-annihilation operators $a = (x + \partial_x)/\sqrt{2}$ and $a^\dagger = (x - \partial_x)/\sqrt{2}$

$$\hat{H} = \sqrt{2} \begin{pmatrix} 0 & a^\dagger \\ a & 0 \end{pmatrix} . \tag{45}$$

One can then easily check that $\hat{H}$ has a unique zero-mode $(e^{-x^2/2}/\sqrt{2\pi}, 0)^t$ (see figure 9) with positive chirality in the convention $\hat{C} = \begin{pmatrix} 1 & 0 \\ 0 & -1 \end{pmatrix}$.

---

[7]Usually the potential is written as $V(x)\sigma_z$ but this is equivalent to our model up to a change of basis which exchanges $\sigma_z \leftrightarrow \sigma_x$.

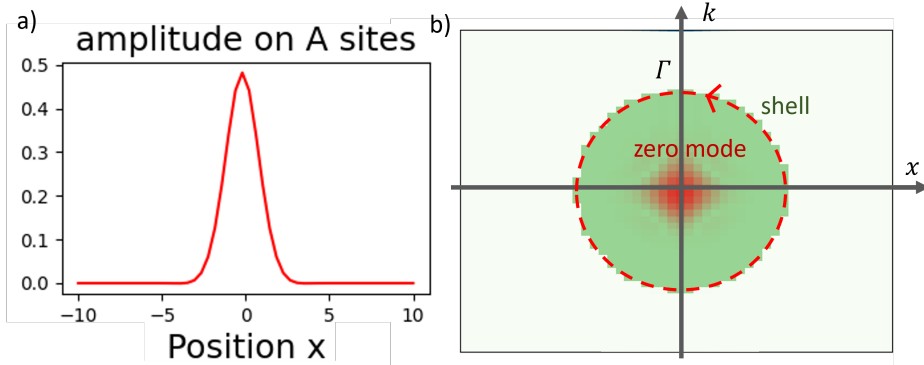

Figure 9: a) Amplitude of the zero mode of the Jackiw-Rebbi model in position b) Wigner-Weyl representation of the zero mode in phase space

The symbol of this Hamiltonian has the simple form

$$H = \begin{pmatrix} 0 & x - ik \\ x + ik & 0 \end{pmatrix} \tag{46}$$

and its energy spectrum reads $\pm\sqrt{k^2 + x^2}$ which indeed satisfies the gap condition when $|x|$ or $|k|$ is large. Moreover, the symbol of the flatten Hamiltonian can be computed easily as $H_F(x,k) = f(H(x,k)) = H(x,k)/\sqrt{H(x,k)^2}$. By using $H(x,k)^2 = (x^2 + k^2)\,\mathbb{1}$, we obtain

$$H_F(x,k) = \frac{1}{\sqrt{x^2 + k^2}} \begin{pmatrix} 0 & x - ik \\ x + ik & 0 \end{pmatrix} = \begin{pmatrix} 0 & e^{-i\phi} \\ e^{i\phi} & 0 \end{pmatrix} \tag{47}$$

which yields the expression for $U = e^{i\phi}$. One can then compute $U^\dagger dU = id\theta$ so that the winding number $W_\circlearrowleft = \frac{1}{2i\pi} \int_{S^1} \mathrm{Tr}^{\mathrm{int}}(U^\dagger dU)$ gives $W_\circlearrowleft = 1$ in agreement with the number of chiral zero-modes.

**Validity of the semi-classical limit**

In this example, we have $\partial_{x/k}H = \sigma_{x/y}$ and therefore $\|\partial_{x/k}H\| = 1$, which does not decrease when $(x,k)$ is large. However, because the gap of $H(x,k)$ varies as $\sqrt{x^2 + k^2}$, our definition of $1/d_{x/k} = \|\partial_{x/k}H(x,k)\|/\Delta(x,k)$ yields $1/d_{x/k} = O(1/\sqrt{x^2 + k^2})$ and hence $1/(d_x d_k) \to 0$. So, this is an example where, even though the variations of the symbol do not vanish at infinity, we still have an exact semi-classical limit because those variations become small compared to the gap.

## 3.4 Discrete approximations of continuous/unbounded topological models

In the previous section, we introduced the topological Hamiltonian (44) which acts on a continuous system that is unbounded both in position and wavenumber spaces. However, in practice, there are physical or numerical limitations which impose bounds on the validity of the model at high position/wavenumber. It is therefore instructive to study finite versions of such models with cut-offs in wavenumber and position, i.e. models defined on a lattice of length $L$ with a lattice spacing $a$. Such discretizations are not unique and may lead to different lattice models with different additional

zero-modes, and thus different topological characterizations, all of them captured with the mode-shell correspondence. We discuss this point for different discretizations of the Jackiw-Rebbi model (44) in this section, that may be skipped at first reading.

For example, the Hamiltonian (44) can be seen as a continuous limit of a discrete SSH Hamiltonian with varying coefficients. If one takes the symbol of the discrete SSH model (31) and replaces the constant coefficients $t$ and $t'$ by $t' = 1/a$ and $t = -1/a + \sin(2\pi x/L)L/(2\pi)$, one obtains a discrete Hamiltonian on a finite lattice of lattice spacing $a$ and length $L >> a$ with periodic boundary conditions, whose symbol reads

$$H_I(x,k) = \begin{pmatrix} 0 & \sin\left(\frac{2\pi x}{L}\right)\frac{L}{2\pi} + \frac{e^{-ika}-1}{a} \\ \sin\left(\frac{2\pi x}{L}\right)\frac{L}{2\pi} + \frac{e^{ika}-1}{a} & 0 \end{pmatrix} \tag{48}$$

and which, by construction, approximates the Jackiw-Rebbi model in the limit $a \to 0$ and $L \to +\infty$.

We now want to determine the points $(x,k)$ of phase space where band crossings occur at zero-energy (see figure 10). Indeed, if such singular points exist and are surrounded by sufficiently large gapped regions in phase space, their non-zero winding number would be associated with topologically protected chiral zero-modes at the operator level (see figure 11). Those points are solution of the equation

$$\sin(2\pi x/L)L/(2\pi) + (e^{ika} - 1)/a = 0 \iff \begin{cases} \sin(ak)/a = 0 \\ (\cos(ka) - 1)/a + \sin(2\pi x/L)L/(2\pi) = 0 \end{cases}. \tag{49}$$

This system has the expected solution $(x,k) = (0,0)$ of winding number $W_\circlearrowleft = +1$ consistently with the fact that this model is built in order to approximate the continuous Jackiw-Rebbi model whose symbol (46) also has this singular point. However, due to the discretisation process, we also get another singular point $(x,k) = (L/2,0) = (-L/2,0)$ (due to the $L$ periodicity in $x$), and whose winding number is found to be $W_\circlearrowleft = -1$ (see figure 10). The two winding numbers therefore sum up to zero as it is expected for finite lattices with equal number of sites of positive/negative chirality. This is due to the fact the total chiral number of zero modes of a finite Hilbert space is given by $\mathcal{I}_{\text{modes}}$ where the cut-off is the identity $\theta_\Gamma = \mathbb{1}$. Therefore its corresponding shell index (9) must vanish ($[\mathbb{1}, \hat{H}_F] = 0$). The existence of such a second singular point due to the discretisation process is therefore topologically constrained.

Note that those two singular points are the only existing ones when $4\pi/(aL) > 1$. In the case $4\pi/(aL) < 1$, two other points also appear at $(x,k) = (\arcsin(4\pi/(aL))/(2\pi)L, \pi/a)$ and $(x,k) = (L/2 - \arcsin(4\pi/(aL))/(2\pi)L, \pi/a)$ which are also characterized by a non-zero winding numbers that sum up to zero. For the sake of brevity and simplicity, we shall however only focus our discussion on the case $4\pi/(aL) > 1$ that yields only two singular points.

In that case, the two chiral zero-modes associated, at the operator level, to these two degeneracy points of opposite winding numbers, resemble the two edge states of the standard SSH model with open boundary conditions, in that they are well separated in position space, around $x = 0$ and $x = L/2$, the only difference being that the new system displays smoother interfaces. Therefore, one can also apply the usual bulk-edge correspondence, by relating the existence of a topological zero-mode with the difference of Brillouin zone winding numbers $W_\uparrow$ far to the left/right side of the mode in position space (vertical dashed lines in figure 10). The two results agree i.e. the value of the winding number $W_\circlearrowleft$, when the shell circles around a zero-energy degeneracy point, corresponds to the difference of the Brillouin zone winding numbers $W_\uparrow$ from each side of the interface (see figure 10).

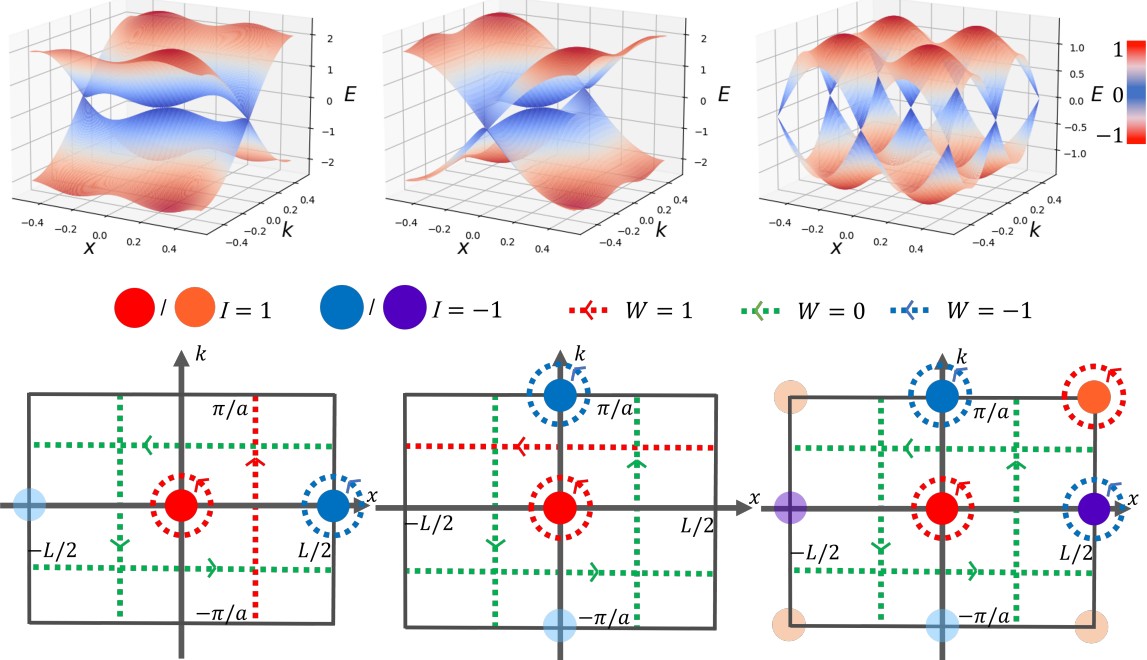

Figure 10: (top) Energies of the symbol Hamiltonians for (left) the model $H_I$ in the regime $4\pi/(aL) > 1$, (center) the model $H_{II}$ in the regime $La/\pi > 1$ and (right) the model $H_{III}$. (bottom) Position of the gap closing points of in phase space. Those positions are denoted by a red/blue dot depending on the chirality of the zero-mode associated to them at the operator level (light color is used for the equivalent periodic images). The different values of the winding number $W$ is enhanced by a red/blue/green color.

One should nevertheless notice that this equivalence is only well established here because there is no other singular mode in the vertical line ($x = 0, k \in [0, 2\pi/a]$) and so that the circle surrounding a degeneracy point can be smoothly deformed into two vertical lines along the Brillouin zone without crossing another band crossing. This is not always the case, and a good illustration is the following dual model of (48) where position and wavenumber play inverted roles

$$H_{II}(x,k) = \begin{pmatrix} 0 & -i\frac{\sin(ak)}{a} + i(e^{-i2\pi x/L} - 1)\frac{L}{2\pi} \\ i\frac{\sin(ak)}{a} - i(e^{i2\pi x/L} - 1)\frac{L}{2\pi} & 0 \end{pmatrix} . \tag{50}$$

The Jackiw-Rebbi model is again recovered in the limit $a \to 0$, $L \to +\infty$, but this second discretized model also exhibits two singular points (in the regime $La/\pi > 1$) $(x,k) = (0,0)$ and $(x,k) = (0,\pi/a)$ which are now separated in wavenumber space rather than in position space. The associated chiral zero-modes, at the operator level, are thus only separated in wavenumber space, unlike the previous discrete model[8]. As such, the difference of Brillouin zone winding numbers $W_\uparrow$ vanishes and is thus unable to detect the existence of chiral zero-modes. This is an example where

---

[8]As discussed at the end of section 3.2, this makes the zero-modes typically less protected in condensed matter applications as they can now be mixed due to scattering induced e.g. by impurities, which induces long range wavenumber couplings in phase space.

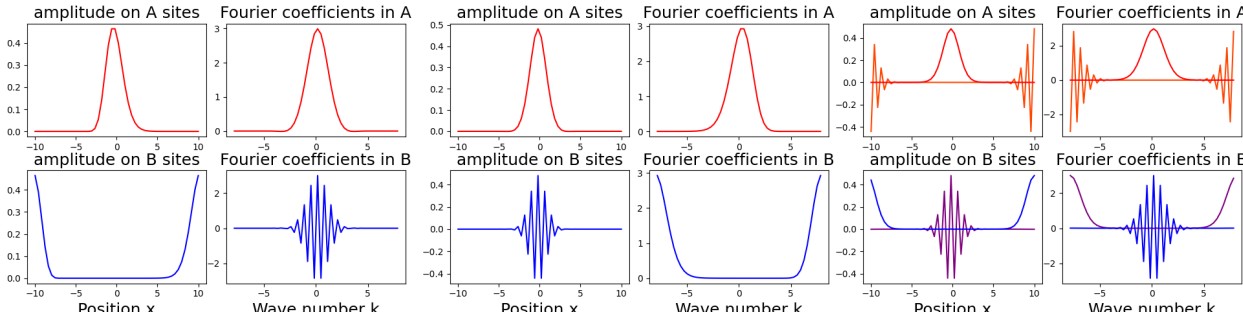

Figure 11: Plots of the different zero-modes of the operator associated to the symbol (left) $H_I$, (center) $H_{II}$, and (right) $H_{III}$ with $L = 20$ and $a = 0.4$. We plot in red/orange the modes of positive chirality and in blue/purple the modes of negative chirality in real space and Fourier space. All those modes are zero-modes in a very good approximation $\|H\,|\psi\rangle\| < 10^{-9}$.

the bulk-edge correspondence of a discretized version of a continuous model is not appropriate to identify chiral zero-modes, while the mixed $x - k$ correspondence, applied locally in phase space, still is. The difference of position winding numbers $W_\rightarrow$ along horizontal lines of positive/negative wavenumber – as in the bounded continuous case of section 3.2 – however coincides with the value of $W_\circlearrowleft$, since no low-wavenumber singular points is here to prevent the deformation of the circle contour into the horizontal one.

Finally, since the topological index only accounts for modes localised both in position and wavenumber spaces, pairs of spurious zero-modes of opposite chirality separated in either or both position/wavenumber directions could appear when one studies finite approximations of a continuous model. A last example where zero-modes appear on both directions is provided by the model

$$H_{III}(x,k) = \begin{pmatrix} 0 & -i\frac{\sin(ak)}{a} + \sin(2\pi x/L)\frac{L}{2\pi} \\ i\frac{\sin(ak)}{a} - \sin(2\pi x/L)\frac{L}{2\pi} & 0 \end{pmatrix} \tag{51}$$

which again converges to the Jackiw-Rebbi model in the limit $L \to +\infty$ and $a \to 0$, but displays now 4 singular points in phase space: 2 of winding number $W_\circlearrowleft = +1$ at $(x,k) = 0$ and $(x,k) = (L/2, \pi/a)$, and 2 of winding number $W_\circlearrowleft = -1$ at $(x,k) = (L/2, 0)$ and $(x,k) = (0, \pi/a)$. Therefore, the only winding numbers that can detect the presence of chiral zero-modes are the $W_\circlearrowleft$'s of the mixed $x - k$ mode-shell correspondence, that are evaluated on $(x - k)$-circle in phase space, since the position and Brillouin zone winding numbers $W_\rightarrow$ and $W_\uparrow$ both vanish.

Those three examples illustrate why the mode-shell correspondence is a natural and general formalism to describe in a unified fashion the existence of all the topologically protected chiral zero-modes. The bulk-edge correspondence and the low-high-wavenumber correspondence are just particular cases which, alone, are not always able to predict the existence of topologically protected chiral zero-modes.

# 4 Higher dimensional chiral mode-shell correspondences

## 4.1 Expression of the general chiral index

In the previous sections, we focused on the mode-shell correspondence in systems of dimension $D = 1$ since this is where the semi-classical invariant takes the simplest forms. However the mode-shell

correspondence generalizes when chiral zero-modes are embedded in a space of higher dimension $D > 1$. While this correspondence can still easily be shown to satisfy $\mathcal{I}_{\mathrm{modes}} = \mathcal{I}_{\mathrm{shell}}$, the main difficulty is to obtain a semi-classical expression of the invariant $\mathcal{I}_{\mathrm{shell}}$.

To understand why there is a difficulty in higher dimension, let us start in $D = 1$ dimension, but take into account the lattice polarisation terms $\mathrm{Tr}\,\hat{C}\hat{\theta}_\Gamma$ in the mode-shell correspondence (11). As we show in appendix D the naive semi-classical expansion of $\mathcal{I}_{\mathrm{shell}}$ becomes

$$\mathcal{I}_{\mathrm{shell}} = \sum_\alpha C_\alpha \Gamma + W^+ - W^- + O(1/\Gamma) \tag{52}$$

which contains the expected difference of winding numbers $W^+ - W^-$ but also a diverging term in $\Gamma$, proportional to $\sum_\alpha C_\alpha$, which is the chiral polarisation of the sites in the unit cell. We argue that, in fact, since $\mathcal{I}_{\mathrm{shell}}$ is finite under the gap condition, the term $\sum_\alpha C_\alpha \Gamma$ must vanish through the condition $\sum_\alpha C_\alpha = 0$.

Actually, this expression is reminiscent of what occurs in higher-dimensional spaces. Indeed, a naive semi-classical expansion of the shell index for $D \geqslant 1$ (i.e. with cut-off parameter $\Gamma \to +\infty$), would lead to an expansion of the form

$$\mathcal{I}_{\mathrm{shell}} = \sum_{k=1}^{D_\mathcal{I}} c_k \Gamma^{D_\mathcal{I}-k} + O(1/\Gamma) \tag{53}$$

where $D_\mathcal{I}$ is the number of infinite dimensions (in position and in wavenumber) of the problem. In the $1D$ case above $c_0 = \sum_\alpha C_\alpha$ and $c_1 = W^+ - W^-$. In general, because the index must converge toward an integer in the $\Gamma \to +\infty$ limit, some cancellations must occur so that $c_k = 0$ for $k < D_\mathcal{I}$ and only the term $c_{D_\mathcal{I}}$ remains, which turns out to be a (higher dimensional)-winding number. However it is not easy to prove that $c_k = 0$ for all $k < D_\mathcal{I}$, without demanding that the index much converge toward the integer. More importantly, because we would need to carry the naive semi-classical expansion of $\mathcal{I}_{\mathrm{shell}}$ to a higher order term, in order to capture the converging component $c_{D_\mathcal{I}}$, the number of terms in the expression of $c_{D_\mathcal{I}}$ should rise, which is difficult to manage and simplify.

In the appendix F, we develop a systematic method to make the cancellations apparent at the level of the operators. We are therefore able to obtain an operator expression of the shell index whose semi-classical limit gives directly the coefficient $c_{D_\mathcal{I}}$ as the leading term. This allows us to obtain a meaningful semi-classical expression of the shell index, as a generalized winding number $\mathcal{W}_{2D-1}$ in $2D - 1$ dimensions as

$$\mathcal{I}_{\mathrm{shell}} \underset{\mathrm{lim}}{=} \frac{-2(D!)}{(2D)!(2i\pi)^D} \int_{\mathrm{shell}} \mathrm{Tr}^{\mathrm{int}}(U^\dagger dU)^{2D-1} \equiv \mathcal{W}_{2D-1} \tag{54}$$

which is the expression anticipated in the introductory general outlines (14). This is one of the key results of this paper. We now provide some elements of the proof of this formula to give some intuition of the result while keeping the more computational intensive part in the appendix F.

Consider a $D$-dimensional system whose Hilbert space basis is labelled by $\mathbb{Z}^n \times \mathbb{R}^m \times [\![1, K]\!]$ with $n + m = D$, where $n$ is the number of discrete dimensions, $m$ is the number of continuous dimensions and $K$ is the number of internal degrees of freedom. Sub-systems of $\mathbb{Z}^n \times \mathbb{R}^m \times [\![1, K]\!]$ such as e.g. the discrete and continuous half-planes ($\mathbb{N} \times \mathbb{Z}$ and $\mathbb{R}^+ \times \mathbb{R}$ respectively) as well as finite lattices, could also be included as we often have a natural way to extend the sub-system Hamiltonian to a larger system by introducing trivial coefficients with no inter-site coupling elsewhere.

After assuming this structure, we assign to each continuous dimension $i$ a position operator $x_i$ and a wavenumber operator $\partial_{x_i}$ which satisfy $[\partial_{x_i}, x_i] = \mathbb{1}$. Similarly, to each discrete dimension $i$ is assigned a position operator $n_i$ and a translation operator $T_i$ which satisfy $T_i^\dagger n_i T_i - n_i = \mathbb{1}$. To treat the continuous and discrete cases in a unified way, we can define the operator $\hat{a}_i$ as $\hat{a}_{2j} = x_j$ and $\hat{a}_{2j+1} = i\partial_{x_j}$ in the continuous case, and as $\hat{a}_{2j} = T_j^\dagger n_j$ and $\hat{a}_{2j+1} = -iT_j$ in the discrete case, so that we have the single commutation relation $[\hat{a}_{2j+1}, \hat{a}_{2j}] = i\,\mathbb{1}$.

Since this commutation relation is proportional to the identity, it allows us to use an "integration by part trick". Indeed, similarly to functions, where $\int dx\, a(x) = -\int dx\, x\partial_x a(x)$, we also have the following relation for operators

$$\operatorname{Tr}\hat{A} = \operatorname{Tr}\Big([\partial_x, x]\hat{A}\Big) = -\operatorname{Tr}\Big(x[\partial_x, \hat{A}]\Big). \tag{55}$$

In the appendix F we use such an integration by part to make some cancellations apparent at the operator level, and thus obtain another expression of the shell index which reads

$$
\begin{aligned}
\mathcal{I} = (-i)^D \frac{D!}{(2D)!} \Bigg( & \operatorname{Tr}\Bigg( \sum_{j_1\ldots j_{2D}=1}^{2D} \varepsilon_{j_1,\ldots,j_{2D}} \hat{C}\hat{H}_F \prod_{l=1}^{2D-1} [\hat{a}_{j_l}, \hat{H}_F][\hat{a}_{2D}, \hat{\theta}_\Gamma] \Bigg) \\
& + \frac{1}{2} \operatorname{Tr}\Bigg( \sum_{j_1\ldots j_{2D}=1}^{2D} \varepsilon_{j_1,\ldots,j_{2D}} \hat{C}\hat{H}_F \prod_{l=1}^{2D} [\hat{a}_{j_l}, \hat{H}_F][\hat{\theta}_\Gamma, \hat{H}_F] \Bigg) \Bigg) + O(\Gamma^{-\infty})
\end{aligned}
\tag{56}
$$

where $\epsilon_{j_1,\cdots,j_{2D}}$ is the antisymetrised Levi-Civita tensor with orientation convention such that $(k_1, x_1, \ldots, k_D, x_D)$ is a direct base. $O(\Gamma^{-\infty})$ means that the equality is valid up to terms which decay faster than any polynomial in $\Gamma$.

The equality (56) can be obtained by assuming only that $\hat{H}$ is gapped deep inside the shell. But if moreover the symbol $H(x, k)$ admits a semi-classical limit when $\Gamma \to +\infty$, we can show that (56) reduces to the simplified expression

$$\mathcal{I}_{\text{shell}} \underset{\lim}{=} \frac{D!}{(2D)!(-2i\pi)^D} \int_{\text{shell}} \operatorname{Tr}^{\text{int}}(CH_F(dH_F)^{2D-1}) \tag{57}$$

where $dH_F$ is now the differential 1-form of $H_F$ in phase space which replaces the commutators $[\hat{a}_{j_l}, \hat{H}_F] \approx i\partial_{j_l}\hat{H}_F$ in the semi-classical limit, $dH_F^{2D-1}$ is the $2D-1$-wedge product of $dH_F$, and the shell is the $2D-1$ dimensional surface enclosing the zero-mode in phase space. The final result (54) is then obtained by substituting $H_F$ by

$$H_F = \begin{pmatrix} 0 & U^\dagger \\ U & 0 \end{pmatrix} \tag{58}$$

in (57). Note that, by homotopy, this formula can also be transformed into

$$\mathcal{W}_{2D-1} = \frac{-2(D!)}{(2D)!(2i\pi)^D} \int_{\text{shell}} \operatorname{Tr}^{\text{int}}(h^{-1}dh)^{2D-1} \tag{59}$$

where $h(x, k)$ is the lower off-diagonal block of the symbol $H(x, k)$ (see (2)). The homotopy invariance is obtained from the smooth deformation of $h$ into $U$ through the homotopic map $h_t = h(1 - t + t\sqrt{h^\dagger h})^{-1}$, with $t$ varying from 0 to 1.

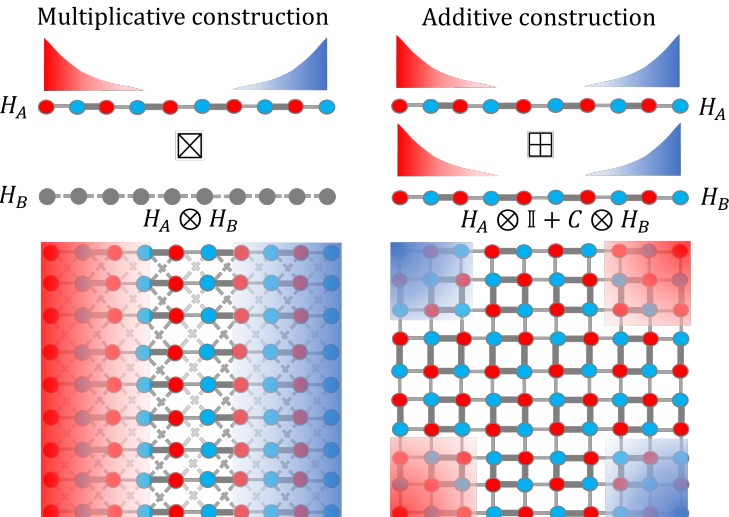

Figure 12: Sketched of $2D$ lattices built out of (a) the multiplicative tensor product construction and (b) the additive tensor product construction, from lower dimensional Hamiltonians $H_A$ and $H_B$. The density of the zero-modes is represented in red/blue depending on their positive/negative chirality; while grey sites denote non chirality. The thickness of the bounds represent the amplitude of the coupling.

In the next two sections, we present different examples of chiral topological systems in higher ($D > 1$) dimensions and analyse how the mode-shell correspondence stated above can be applied to those examples. In order to simplify the analysis, we focus on examples in $D = 2$ dimensions. We present two general methods that provide those simple-to-analyse higher-dimensional examples, combining lower-dimensional examples through tensor product structures. The first method, that we refer to as the *multiplicative* tensor product construction and denote with the symbol $\boxtimes$, yields examples of weak insulators, that exhibit a macroscopic number of boundary states, while the second method, that we refer to as the *additive* tensor product construction and denote with the symbol $\boxplus$, provides examples of higher-order insulators that exhibit e.g. corner states (see figure 12). The systems serving as building blocks for these construction can be discrete or continuous, and of any dimension. Also, those constructions can be combined or used multiple times to create examples in even higher dimension (see figure 13 for $D = 3$).

## 4.2 Chiral Weak-insulators and flat-band topology

One way to engineer topological states in higher dimension is to stack $1D$ topological systems, such like SSH chains. We would then have a number of gapless modes growing extensively with the transverse size (say $y$) of the sample as it would be equal to the number of copies $N_y$. The zero-modes would then gradually form a flat zero-energy band in this transverse direction, a phenomenon observed experimentally [99, 100, 103, 114].

Such stacked systems result is what is often called "weak topological insulators" in the literature [114–123]. Stacked versions of $2D$ quantum spin Hall [20, 21, 124] or quantum Hall [125] phases are other $3D$ examples beyond the chiral. The adjective "weak" was originally used since the edge states were first expected not to be topologically protected against disorder or inter-layer

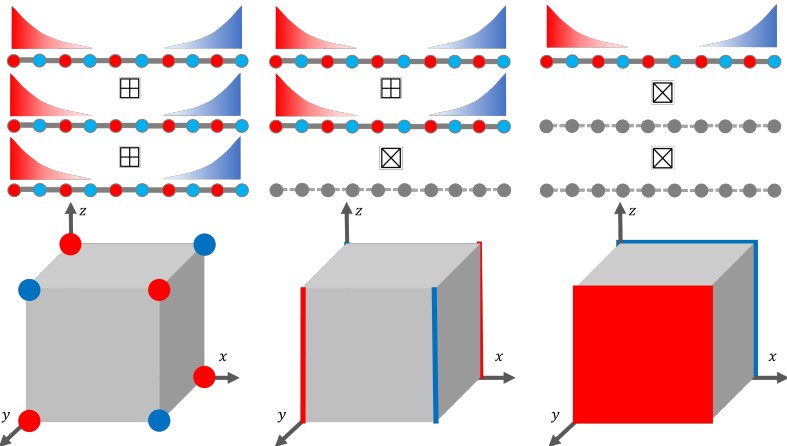

Figure 13: Sketches of possible of $3D$ topological systems obtained by applying (left) twice the additive tensor product construction yielding zero-modes corner states, (center) both the additive and the multiplicative tensor product construction, providing an extensive number of zero-modes states localised on hinges, and (right) twice the multiplicative tensor product construction leading to an extensive number of zero-modes states localised on surfaces.

couplings [115, 116], but it was later realized that they turned out to be robust to such kinds of perturbations [117–120] making the terminology nowadays a little bit outdated. Also, a weak topological insulator is usually characterized by a topological index associated to a reduced Brillouin zone (and thus dubbed *weak invariant*) in contrast with *strong* topological insulators whose (strong) invariants encompass the entire Brillouin zone. We recall that $1D$ strong chiral topological insulators are the only strong insulators that are captured by the chiral index defined in this paper. The mode-shell correspondence with higher-dimensional strong invariants will be exposed in a follow up paper.

In this section, we analyse chiral weak insulators through the mode-shell correspondence. To do so, we consider $2D$ systems, such as those depicted in figure 14 and 15, where the left and right edges host a macroscopic number of edge modes in the $y$ direction. To select the leftmost extended edge states, we then choose a cut-off operator which is uniform in the $y$ direction and localised near the left edge. Next, if the system is such that its bulk is gapped, and if its upper and lower edges are also gapped, then the invariant $\mathcal{I} = \mathrm{Tr}\left(\hat{C}(1 - \hat{H}_F^2)\hat{\theta}_\Gamma\right)$ can be shown to be quantised as in the $1D$ case. The only difference with the $1D$ case is that the index $\mathcal{I}$ is (macroscopically) much larger and depends of the transverse length $N_y$ of the lattice. The proof of the quantisation of the number of edge modes only requires chiral symmetry and is insensitive to the presence of disorder or inter-layer couplings, which shows the robustness of those modes.

Let us now compute the shell invariant in phase space using the Wigner-Weyl transform. One gets

$$\mathcal{I} = \frac{1}{2} \sum_{(x,y)\in\mathcal{L}} \int_0^{2\pi} \frac{dk_x}{2\pi} \int_0^{2\pi} \frac{dk_y}{2\pi} \, \mathrm{Tr}^{\mathrm{int}}(C \star H_F \star [\theta_\Gamma, H_F]_\star) \ . \tag{60}$$

The next step is to perform a semi-classical expansion in terms of $1/\Gamma$ and keep the dominant term. To be valid, this approximation requires the Hamiltonian to vary slowly in position $(x, y)$ (see section 2.5); this is valid in the major part of the shell which is in the bulk as we have translation invariance in both directions. However, it is invalid near the upper and lower edges since there the

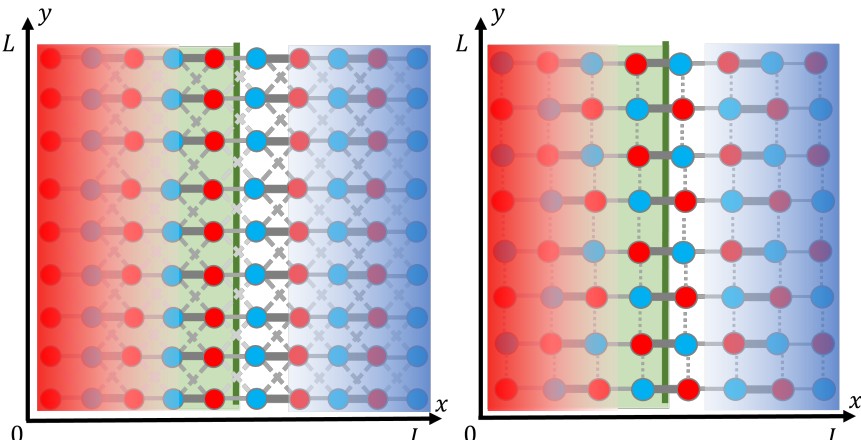

Figure 14: Two examples of lattices with a macroscopic number of chiral zero-modes on the left vertical edge. (Left) Stacking of topological chiral SSH chains in the vertical direction with inter-layer couplings that preserve chiral symmetry. (Right) Stacking of staggered trivial and topological SSH chains that preserve chiral symmetry. This example has the advantage of involving only nearest neighbour interactions without breaking chiral symmetry. To compute the macroscopic topological index associated to these lattices, one needs to chose a cut-off which is uniform in the vertical coordinate and decreases only in the horizontal coordinate.

Hamiltonian varies sharply in the $y$ direction. If we were ignoring the perturbations due to the edges, we would be allowed to perform a semi-classical expansion in both directions and we would get a bulk index $\mathcal{I}^b \sim \mathcal{I}$

$$
\begin{aligned}
\mathcal{I}^b &\underset{\lim}{=} \frac{1}{2} \int_0^{2\pi} \frac{dk_y}{2\pi} \int_0^{2\pi} \frac{dk_x}{2\pi} \operatorname{Tr}^{\mathrm{int}}(C H_F^b i \partial_{k_x} H_F^b \sum_{(x,y)\in\mathcal{L}} \delta_x \theta_\Gamma(x,y)) \\
&\underset{\lim}{=} N_y \int_0^{2\pi} \frac{dk_y}{2\pi} \int_0^{2\pi} \frac{dk_x}{4i\pi} \operatorname{Tr}^{\mathrm{int}}(C H_F^b \partial_{k_x} H_F^b)
\end{aligned}
\tag{61}
$$

where $N_y$ is the number of stacked chains and $H_F^b$ is the symbol of $\hat{H}_F$ in the bulk. In the right hand-side of the expression, the term

$$
\frac{1}{4i\pi} \int_{k_x\in[0,2\pi],k_y=k_{y0}} dk_x \operatorname{Tr}^{\mathrm{int}}(C H_F^b \partial_{k_x} H_F^b) \equiv \mathcal{I}_{\mathrm{weak}}(k_{y0})
\tag{62}
$$

is known to be a topological invariant which remains constant when deforming the symbol $H_F^b$ without closing the gap. As a result, it does not depend on the choice of $k_{y0}$, so the average $\int_0^{2\pi} dk_x$ can be replaced by the integration over any line of constant $k_y$ in Fourier space. Such an invariant is sometimes called weak invariant because the integration only runs over a one-dimensional path while the system is two-dimensional. The bulk invariant then reads

$$
\mathcal{I}^b = N_y \mathcal{I}_{\mathrm{weak}} .
\tag{63}
$$

By ignoring the effect of the edges, $\mathcal{I}^b$ is in principle an approximation of $\mathcal{I}$. To recover $\mathcal{I}$, one thus needs to add a correction term $\Delta_{\mathrm{edge}}$ coming from the fact that the actual Hamiltonian near the

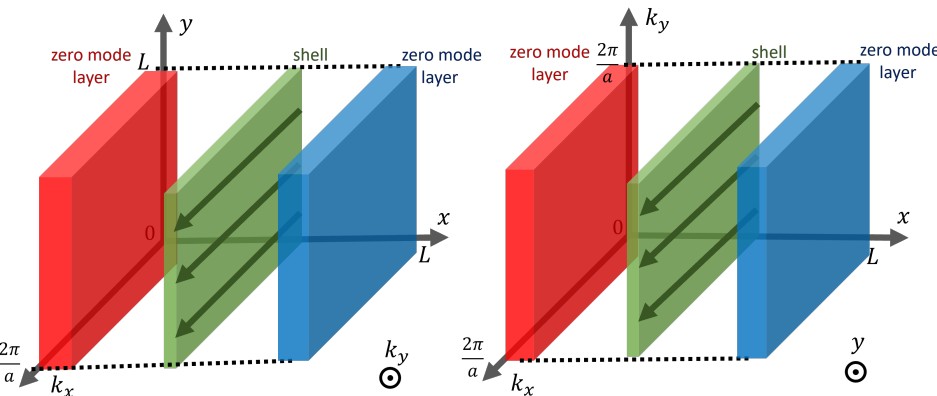

Figure 15: Phase space representation of the zero-modes (in red and blue) and the shell (in green) for weak insulators such as depicted in figure 14. In the bulk where semi-classical limit occurs, the shell invariant can be reduced to a weak invariant which is a winding number in the $k_x$ direction (green arrows) and a multiplicative constant depending on the $y$ and $k_y$ directions.

edges at $y = 0$ and $y = L_y$ differs from the bulk Hamiltonian, that is

$$\mathcal{I} = \mathcal{I}^b + \Delta_{\text{edge}}. \tag{64}$$

This correction can be computed numerically by evaluating $\mathcal{I}$ before the semi-classical expansion. Since $\mathcal{I}$, $N_y$ and $\mathcal{I}_{\text{weak}}$ are all integers, $\Delta_{\text{edge}}$ must also be an integer and its specific value may *a priori* depend on the boundary conditions. However, since the correction term only originates from the sites located close to a boundary, this term is bounded and thus cannot scale with $N_y$. As a result, even the strangest boundary condition cannot change the fact that there is a macroscopic number of zero-modes localised on the left edge of the 2D lattice. In fact even if one has a boundary condition that closes the gap on the upper/lower boundary, the computation above mostly remains the same and we still have the relation (64). Furthermore, since the chiral index also reads $\mathcal{I} = \text{Tr}\, \hat{C}(1 - \hat{H}_F^2)\hat{\theta}_\Gamma$, and since $(1 - \hat{H}_F^2)$ is only non-zero for modes of very small energy, it follows that there should be a macroscopic number of very small energy modes on the left edge. However, $\mathcal{I}$ and $\Delta_{\text{edge}}$ may in that case no longer be integers and deviate from quantisation. But the massive polarisation of the zero-modes remains. Those are still protected as long as $\mathcal{I}_{\text{weak}} \neq 0$ which is guaranteed since $\mathcal{I}_{\text{weak}}$ is a bulk topological invariant which cannot change as long as there is a bulk gap.

It should be noted that weak invariants are also used to prove the existence of edge Fermi arcs in semi-metals like graphene and Weyl semimetals [103, 125, 126]. If we decide to not add this example in order to not further lengthen the size of the article, it can be understood in a similar manner as the previous presentation, the only difference being that we have to introduce, in the definition of the index, a cut-off in wavenumber space of the direction tangent to the edge in order to select the part of the Fourier space where Fermi arcs exists (between the edge projection of two bulk Dirac/Weyl cones where the bulk gap closes).

### Multiplicative tensor product construction ⊠, for chiral weak insulators

In this section, we present a simple but general mathematical procedure to generate such staking of chiral topological systems. At the Hamiltonian level, it simply consists in defining a Hamiltonian

$\hat{H}$ as the tensor product of two lower dimensional gapped Hamiltonians $\hat{H}_A$ and $\hat{H}_B$ [127] as

$$\hat{H} = \hat{H}_A \otimes \hat{H}_B \tag{65}$$

where $\hat{H}_A$ is chiral symmetric, while $\hat{H}_B$ encodes the coupling between the stacked copies. Such a procedure was recently referred to as "multiplicative topology" in the literature [128–130]. If $\hat{H}_A$ and $\hat{H}_B$ are Hamiltonians on lattices or continuous spaces of dimension $D_A$ and $D_B$, then $\hat{H}$ is a Hamiltonian which acts on a $D = D_A + D_B$ dimensional space. Moreover, if we denote by $\hat{C}_A$, the chiral symmetry operator of $\hat{H}_A$, then $\hat{H}$ has the chiral symmetry $\hat{C} = \hat{C}_A \otimes \mathbb{1}$.

Importantly, the spectral properties of $\hat{H}$ are entirely determined by those of the sub-systems $\hat{H}_A$ and $\hat{H}_B$. Indeed, if $\left|\psi_n^A\right\rangle$ is an eigenbasis of $\hat{H}_A$ with energies $E_n^A$ and $\left|\psi_m^B\right\rangle$ is an eigenbasis of $\hat{H}_B$ with energies $E_m^B$, then $\left|\psi_n^A\right\rangle \otimes \left|\psi_m^B\right\rangle$ is an eigenbasis of $\hat{H}$ with energies $E_n^A E_m^B$. In particular the zero-modes of $\hat{H}$ are those which are of the form (or a linear combination of) $\left|\psi_n^A\right\rangle \otimes \left|\psi_m^B\right\rangle$ where either $\left|\psi_n^A\right\rangle$ or $\left|\psi_m^B\right\rangle$ a zero-mode. This means that if $\hat{H}_B$ acts on a finite space with $N$ sites, then for each zero-mode $\left|\psi_{n_0}^A\right\rangle$ of $\hat{H}_A$, one can associate $N$ zero-modes of $\hat{H}$ since no matter what is $\left|\psi_m^B\right\rangle$, $\left|\psi_{n_0}^A\right\rangle \otimes \left|\psi_m^B\right\rangle$ remains a zero-mode.

In particular, if $\hat{H}_A$ is just the identity operator on a finite lattice of $N_y$ sites, i.e. $\hat{H}_B = \sum_{j=1}^{N_y} |j\rangle \langle j|$, then $\hat{H} = \hat{H}_A \otimes \hat{H}_B$ is just the Hamiltonian of $N_y$ stacked copies of topological chiral systems described by $\hat{H}_A$ with no coupling between the different copies. If $\mathcal{I}_A$ is the non-trivial chiral topological index of $\hat{H}_1$ with respect to a cut-off operator $\hat{\theta}_\Gamma$, and if $\hat{H}_B$ is gapped, then, as we detail below, one can check that $\hat{H}$ has also a well defined topological index $\mathcal{I}$ associated to the cut-off operator $\hat{\theta}_\Gamma \otimes \mathbb{1}$ which is given by

$$\mathcal{I} = \mathcal{I}_A \times N_y \,. \tag{66}$$

We thus naturally end up with a number of chiral zero-modes that grows extensively with the stacking direction of the system. The zero-modes would then gradually form a zero energy flatband in this direction, a phenomenon observed experimentally [99, 100, 103, 114].

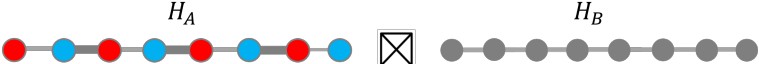

Figure 16: Tensor product structure which generates the model illustrated in figure 14 a) The left model, given by $H_A$ is a topological SSH models. The right model, given by $H_B$, is a 1D chain with some on-site and nearest neighbour couplings.

**Example 1:** One can use this multiplicative construction to generate some of the lattices in the figure 12 and 14. For example, the model described in the left parts of these figures can be generated from the tensor product of a topological SSH model, i.e. $\hat{H}_1 = \hat{H}_{\text{SSH}}$ given by (29), with a simple non-chiral model of the form $\hat{H}_2 = \sum_n |n\rangle \langle n| + t''(|n\rangle \langle n+1| + |n+1\rangle \langle n|)/2$ which, in the bulk, has the symbol $H_2(n, k) = 1 + t'' \cos(k_y)$ and is hence gapped when $|t''| < 1$. In the multiplicative construction, the $t''$ coefficient then creates vertical couplings between different SSH layers which preserve chiral symmetry and the existence of topological zero-modes. The symbol of the tensored

Hamiltonian $\hat{H} = \hat{H}_1 \otimes \hat{H}_2$ reads

$$H(n_x, n_y, k_x, k_y) = \begin{pmatrix} 0 & t + t'e^{ik_x} \\ t + t'e^{-ik_x} & 0 \end{pmatrix} (1 + t'' \cos(k_y)) \tag{67}$$

which is just the symbol of the SSH model multiplied by a scalar constant depending of $k_y$. There-fore, when computing the weak index $\mathcal{I}_{\text{weak}}(k_{y_0})$ as described by the formula (66), we obtain that, for $|t''| < 1$, $\mathcal{I}_{\text{weak}}(k_{y_0}) = \mathcal{W}_1$ with $\mathcal{W}_1$ the winding number of the $1D$ SSH model. The $1D$ mode-shell correspondence gives us $\mathcal{I}_1 = \mathcal{W}_1$ and the multiplicative structure implies $\mathcal{I} = \mathcal{I}_1 \times N_y$. One can therefore verify that at the leading order in $N_y$ we have

$$\mathcal{I} \underset{N_y}{\sim} \mathcal{I}_{\text{weak}}(k_{y_0}) N_y \tag{68}$$

which is indeed the result predicted by the mode-shell correspondence. In this system, this relation is in fact an equality, due to the multiplicative structure which forbids edge corrections of the formula to occur.

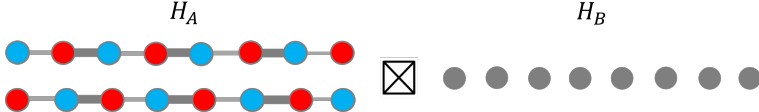

Figure 17: Tensor product structure which generates the model illustrated in figure 14 b) up to the addition, *a posteriori*, of inter-layer couplings. The left model, given by $H_A$ is the superposition of a topological and trivial SSH models. The right model, given by $H_B$, is a trivial chain with constant onsite coupling $\hat{H}_B = \sum_n |n\rangle \langle n| = \mathbb{1}$.

**Example 2:** One problem one may have with the above example is that nearest neighbour cou-plings are forbidden because they would break chiral symmetry. There are models which do not have this drawback. For example, one could first stack a topological chain with a trivial SSH chain as depicted in figure 17 and then use the tensor product construction to create a $2D$ stack of this two-layer quasi-$1D$ model. If one adds nearest neighbour interactions between the layers, one would then obtain the model depicted in the right part of the figure 14. This inter-layer coupling breaks the multiplicative structure but the existence of chiral zero-modes only relies on the chiral symmetry and on a gap on the shell, two assumptions which are not broken by those couplings. So, as long as these inter-layer couplings are not too strong to close the gap, the macroscopic number of chiral zero-modes on the left edge remains topologically protected.

## 4.3  Higher-order chiral insulators

In this section, we discuss how the modes-shell correspondence can be applied to describe higher-order chiral insulators which exhibit zero-modes localised in more than 1 dimensions. To generate higher-dimensional chiral topological examples which are simple to study, we follow a procedure that we call the *additive tensor product construction* and we refer to it by the symbol $\boxplus$. We use this method to generate two examples in $D = 2$: one lattice model and one continuous model on which we verify, illustrate and discuss the predictions of the mode-shell correspondence theory.

**Additive tensor product construction ⊞ for higher-order chiral insulators**

The additive tensor product construction is another procedure [131, 132] by which one can generate a higher-dimensional topological chiral Hamiltonian $\hat{H}$ from two lower dimensional Hamiltonians $\hat{H}_1$ and $\hat{H}_2$. It requires that both $\hat{H}_A$ and $\hat{H}_B$ are chiral symmetric with chiral operators $\hat{C}_A$ and $\hat{C}_B$ respectively. A chiral higher dimensional Hamiltonian can then defined as

$$\hat{H} = \hat{H}_A \otimes \mathbb{1} + \hat{C}_A \otimes \hat{H}_B \tag{69}$$

or equivalently by substituting $A$ by $B$. If this additive construction seems a little bit more involved than the multiplicative one, the spectral properties of $\hat{H}$ are still determined by those of $\hat{H}_1$ and $\hat{H}_2$ since

$$\hat{H}^2 = \hat{H}_A^2 \otimes \mathbb{1} + \mathbb{1} \otimes \hat{H}_B^2 + \{\hat{H}_1, \hat{C}_1\} \otimes \hat{H}_2 = \hat{H}_A^2 \otimes \mathbb{1} + \mathbb{1} \otimes \hat{H}_B^2 . \tag{70}$$

Therefore, if $\left|\psi_n^A\right\rangle$ is an eigenbasis of $\hat{H}_A$ with energies $E_n^A$ and $\left|\psi_m^B\right\rangle$ is an eigenbasis of $\hat{H}_B$ with energies $E_m^B$, then $\left|\psi_n^A\right\rangle \otimes \left|\psi_m^B\right\rangle$ is the eigenbasis of $\hat{H}^2$ with energies $(E_n^A)^2 + (E_m^B)^2$, so that the eigenvalues of $\hat{H}$ are $\pm\sqrt{(E_n^A)^2 + (E_m^B)^2}$. It follows that the zero-modes of $\hat{H}$ are of the form $\left|\psi_{n_0}^A\right\rangle \otimes \left|\psi_{m_0}^B\right\rangle$ where $\left|\psi_{n_0}^A\right\rangle$ *and* $\left|\psi_{m_0}^B\right\rangle$ are zero-modes respectively of $\hat{H}_A$ and $\hat{H}_B$. This is quite different from the additive construction, since we need here the two Hamiltonians $\hat{H}_A$ and $\hat{H}_B$ to have of zero-modes, and not only one of them. As a result, this procedure generates higher order chiral insulators with few zero-modes, in contrast with weak chiral insulators (see figure 12). Indeed, if $\hat{H}_A$ and $\hat{H}_B$ have each a well defined chiral topolgical index $\mathcal{I}_A$ and $\mathcal{I}_B$ (with respect to the cut-off operators $\hat{\theta}_{\Gamma,A}$ and $\hat{\theta}_{\Gamma,B}$), then one can check that $\hat{H}$ has also a well defined chiral index $\mathcal{I}$, associated to the cut-off operator $\hat{\theta}_{\Gamma,A} \otimes \hat{\theta}_{\Gamma,B}$. We then use the fact that the chiral polarisation of the zero-modes $\left|\psi_{n_0}^A\right\rangle \otimes \left|\psi_{m_0}^B\right\rangle$ is the product of the chiral polarisation of each individual mode, which leads to

$$\mathcal{I} = \mathcal{I}_A \times \mathcal{I}_B . \tag{71}$$

Of course, since the higher-dimensional Hamiltonian $\hat{H}$ is itself chiral symmetric, it can serves as a new block to apply the procedure again. By induction, we get the more general formula

$$\hat{H} = \hat{H}_1 \otimes \mathbb{1}^{\otimes(N-1)} + \hat{C}_1 \otimes \hat{H}_2 \otimes \mathbb{1}^{\otimes(N-2)} + \cdots + \hat{C}_1 \otimes \cdots \otimes \hat{C}_{N-1} \otimes \hat{H}_N \tag{72}$$

of a chiral Hamiltonian resulting from the additive tensor product construction with $N$ chiral symmetric Hamiltonians $\hat{H}_j$ of chiral symmetry operator $\hat{C}_j$ and chiral topological index $\mathcal{I}_j$ ($j = 1 \ldots N$). The chiral symmetry operator of $\hat{H}$ is then given by the tensor product $\hat{C}_1 \otimes \cdots \otimes \hat{C}_N$, and the zero-modes of $\hat{H}$ have a chiral index $\mathcal{I} = \mathcal{I}_1 \times \cdots \times \mathcal{I}_N$.

The next two paragraphs are dedicated to two simple illustrations of the additive tensor product construction in $D = 2$ and to the analysis of the resulting models through the higher-dimensional mode-shell correspondence.

**Example 1: $2D$ Jackiw-Rossi model with smooth potentials**

We discuss an example of the higher-dimensional mode-shell correspondence for a chiral zero-mode trapped at a domain wall where the Hamiltonian is smoothly varying. Such a situation has been studied in the literature in the context of defects modes [48, 133]. It allows for a full semi-classical

limit of the shell index leading to the Teo and Kane formula in the case of discrete lattice [48] and the Callias index formula in the continuous case [90]. As both discrete and continuous cases are relatively similar, we made the choice to focus our attention to the continuous case in this section.

For that purpose, we revisit the Jackiw-Rossi model [134] which follows from the same construction as the BBH model above: The two-dimensional Jackiw-Rossi Hamiltonian $\hat{H}$ is obtained by combining, in perpendicular directions $x$ and $y$, two one-dimensional Jackiw-Rebbi Hamiltonians $\hat{H}_{\mathrm{JR}}$ introduced in (44), by following the additive tensor product construction, that is

$$\hat{H} = \hat{H}_{\mathrm{JR}}(x, \partial_x) \otimes \mathbb{1} + \sigma_z \otimes \hat{H}_{\mathrm{JR}}(y, \partial_y) \tag{73}$$

where $\sigma_z$ is the chiral-symmetric operator of the two underlying Jackiw-Rebbi models, and which, in a more explicit form, reads

$$\hat{H} = \begin{pmatrix} 0 & y - \partial_y & x - \partial_x & 0 \\ y + \partial_y & 0 & 0 & x - \partial_x \\ x + \partial_x & 0 & 0 & -(y - \partial_y) \\ 0 & x + \partial_x & -(y + \partial_y) & 0 \end{pmatrix}. \tag{74}$$

Similarly to the previous example with open conditions, this Hamiltonian has a chiral symmetry with a chiral operator $\hat{C} = \sigma_z \otimes \sigma_z$. Writing $\hat{H}^2 = \hat{H}_{\mathrm{JR}}^2(x, \partial_x) \otimes \mathbb{1} + \mathbb{1} \otimes \hat{H}_{\mathrm{JR}}^2(y, \partial_y)$ implies that the chiral zero-modes of $\hat{H}$ must also be chiral zero-modes of $\hat{H}_{\mathrm{JR}}(x, \partial_x)$ and $\hat{H}_{\mathrm{JR}}(y, \partial_y)$ on each part of the tensor product. Those modes must thus be of the form $\psi_x \otimes \psi_y = (e^{-(x^2+y^2)/2}, 0, 0, 0)^t$ where $\psi(x) = (e^{-x^2/2}, 0)^t$ is the zero-mode of the Jackiw-Rebbi model. Therefore, $\hat{H}$ has one topological zero-mode of chirality $+1$ and thus $\mathcal{I} = 1$ for the cut-off $\hat{\theta}_\Gamma = e^{-(x^2+y^2-\partial_x^2-\partial_y^2)/\Gamma^2}$ which acts here both in position and wavenumber. The corresponding shell is a $3D$ sphere enclosing the chiral zero-mode in $4D$ phase space, as sketched in figure 18.

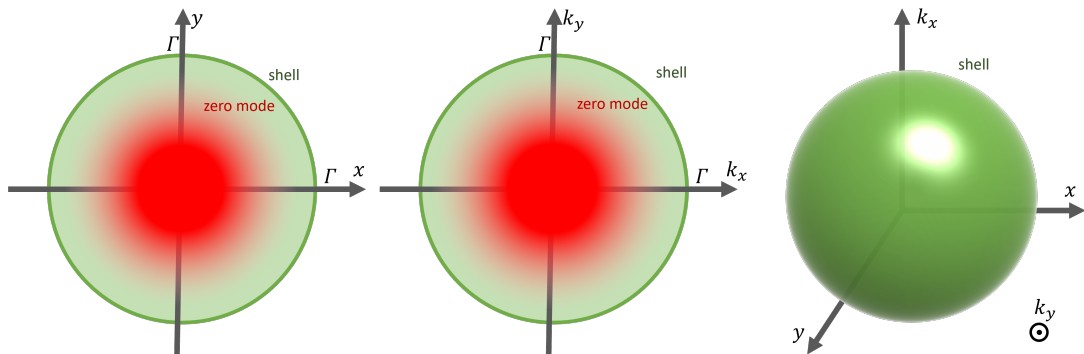

Figure 18: Amplitude of the zero-mode $\psi_x \otimes \psi_y = (e^{-(x^2+y^2)/2}, 0, 0, 0)^t$ of the Hamiltonian (73) in real space (left) and in Fourier space (center). The cut-off operator $\hat{\theta}_\Gamma = e^{-(x^2+y^2-\partial_x^2-\partial_y^2)/\Gamma^2}$ selects the zero-mode both in position and in wavenumber directions. It is represented in green and has the form of a $4D$-ball in phase space, with the shell being a $3D$ sphere (right).

The symbol $H$ of the Jackiw-Rossi Hamiltonian reads

$$H = \begin{pmatrix} 0 & x - ik_x \\ x + ik_x & 0 \end{pmatrix} \otimes \mathbb{1} + \sigma_z \otimes \begin{pmatrix} 0 & y - ik_y \\ y + ik_y & 0 \end{pmatrix} \tag{75}$$

from which we deduce

$$H_F = \frac{H}{\sqrt{x^2 + y^2 + k_x^2 + k_y^2}}$$

$$= \cos(\theta) \begin{pmatrix} 0 & e^{-i\phi_1} \\ e^{i\phi_1} & 0 \end{pmatrix} \otimes \mathbb{1} + \sigma_z \otimes \sin(\theta) \begin{pmatrix} 0 & e^{-i\phi_2} \\ e^{i\phi_2} & 0 \end{pmatrix} \tag{76}$$

where $(\theta, \phi_1, \phi_2) \in [0, \pi/2] \times [0, 2\pi]^2$ are the Hopf-coordinates of $S^3$. One can then compute analytically $\int_{S^3} \text{Tr}^{\text{int}}(CH_F dH_F^3) = -12(2\pi)^2$ which is exactly the normalisation needed to have $\mathcal{I} = 1$.

**Example 2: The Benalcazar-Bernevig-Hughes (BBH) model with open boundary conditions**

Higher-order topological insulators (HOTI) constitute a class of systems where topological zero-modes are embedded in a higher dimensional phase space. Those zero-modes could then be trapped at the corners of a material where the trapping potential varies sharply at the edges [23, 24, 132, 135–138]. The archetypal lattice model describing such a situation is the Benalcazar, Bernevig, Hughes (BBH) model [23], depicted in figure 19, which essentially consists in "crossing" arrays of SSH models along the $x$ and $y$ directions such that chiral symmetry is preserved. The resulting Hamiltonian follows the additive construction and reads

$$\hat{H} = \hat{H}_{\text{SSH},x} \otimes \mathbb{1} + \sigma_z \otimes \hat{H}_{\text{SSH},y} \tag{77}$$

where $\sigma_z$ is the chiral-symmetric operator of the two underlying SSH models, and which, in a more explicit form, becomes

$$\hat{H} = \begin{pmatrix} 0 & t + t'T_y^\dagger & t + t'T_x^\dagger & 0 \\ t + t'T_y & 0 & 0 & t + t'T_x^\dagger \\ t + t'T_x & 0 & 0 & -(t + t'T_y^\dagger) \\ 0 & t + t'T_x & -(t + t'T_y) & 0 \end{pmatrix} \tag{78}$$

where $T_x = \sum_{n_x, n_y} |n_x + 1, n_y\rangle \langle n_x, n_y|$ and $T_y = \sum_{n_x, n_y} |n_x, n_y + 1\rangle \langle n_x, n_y|$ are the translation operators of one lattice unit along $x$ and $y$ respectively. We also impose open boundary conditions as in figure 19.

The Hamiltonian $\hat{H}$ inherits chiral symmetry from the two underlying chiral symmetric lower dimensional systems, and its chiral operator reads $\hat{C} = \sigma_z \otimes \sigma_z$. The chiral zero-modes of $\hat{H}$ can then be easily found: using the additive chiral construction, we know that the zero-modes of $\hat{H}$ must also be zero-modes of $\hat{H}_{\text{SSH},x}$ and $\hat{H}_{\text{SSH},y}$ on each part of the tensor product. They must therefore be of the form $\psi_x \otimes \psi_y$ where $\psi_{x/y}$ is the zero-mode in the $x/y$ direction of the SSH model. It follows that for $|t'| > |t|$, where the SSH models are topological, the BBH model $\hat{H}$ has one topological zero-mode of chirality $+1$ in its bottom-left corner, and therefore, it has $\mathcal{I} = 1$ for the cut-off $\hat{\theta}_\Gamma = e^{-(x^2 + y^2)/\Gamma^2}$, which acts in both position directions.

The use of the shell invariant is, however, not as straightforward as in the previous cases. The reason is that the shell, that encircles the corner of interest (see figure 19), runs not only over the bulk but also over the sharp edges where the Hamiltonian does not vary smoothly. Therefore, taking the limit $\Gamma \to \infty$ in the shell invariant does not guarantee the semi-classical limit in every directions. As a consequence, we cannot derive an expression of the shell index which is as simple as (54).

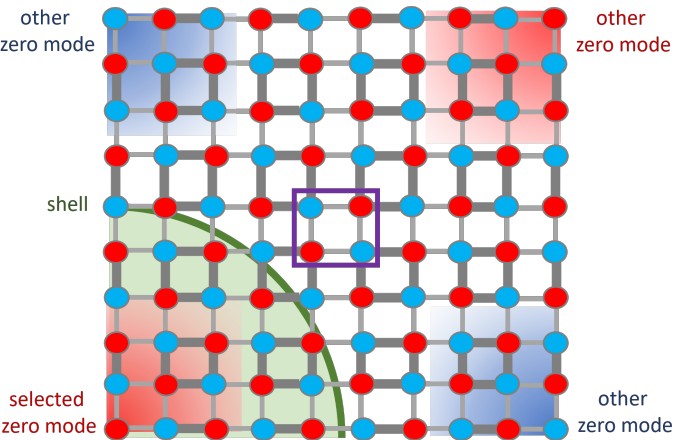

Figure 19: Lattice BBH model with chiral zero-modes of positive/negative chirality depicted in red/blue. The shell is shown in green, an example of unit cell is shown in purple. The thickness of the links represents the strength of the coupling. The system has open boundary conditions with edges given by the horizontal and vertical lines $x = 0$, $x = L$, $y = 0$, $y = L$.

We can still simplify partially the expression by using the translation invariance of $\hat{H}_F$ in the direction parallel to each edge. In fact, one can show (see appendix G) that the index can be written as the sum of two contributions $\mathcal{I}_{\text{shell}} = \mathcal{I}_{\text{edge},x} + \mathcal{I}_{\text{edge},y}$ where each contribution is localised at one of the two edges and where we can use the Fourier transform in the direction parallel to that edge. In particular, $\mathcal{I}_{\text{edge},x}$ can be written as

$$\mathcal{I}_{\text{edge},x} = \frac{-1}{24\pi} \int_0^{2\pi} dk_x \tilde{\text{Tr}} \left( \tilde{C} \tilde{H}_F [\tilde{d}, \tilde{H}_F]^3 \right) \tag{79}$$

where the notation $\sim$ means that the Wigner-Weyl transform is performed in the tangent direction only. The operator $\tilde{d}$ is for example $\tilde{d} = \partial_x dk_x + T_y^\dagger n_y dy - iT_y dk_y$. The expression of $\mathcal{I}_{\text{edge},x}$ is obtained by switching the $x$ and $y$ coordinates.

Since the semi-classical treatment is invalid in the perpendicular direction to the edge, $\mathcal{I}_{\text{edge},x}$ (or equivalently $\mathcal{I}_{\text{edge},y}$) remains cumbersome to manipulate by hand.So we prefer here to evaluate $\mathcal{I}_{\text{edge},x/y}$ numerically. This also gives us the opportunity to show explicit examples of numerical codes that compute numerical approximations of the indices. Those can be found in Appendix G. One of these codes computes the index $\mathcal{I}_{\text{modes}}$ directly from the initial formula (8) while the other one compute $\mathcal{I}_{\text{shell}}$ using the formulation with partial semi-classical limit (79). For lattices of length $L = 10$ sites, both codes give $\mathcal{I} = 1$ up to a deviation of less than 1%, which validates numerically the mode-shell correspondence for the BBH model.

We can mention that the recourse to additional symmetries is commonly used in the literature of higher-order topological insulators. Those symmetries can be interpreted as a way to reduce the complexity of the computation of the shell index by re-expressing it as a pure bulk quantity. For instance in [23], it is claimed that due to the $C_4$ rotational symmetry which is present in the BBH model, the quadruple moment is a topological invariant in the bulk related to the number of corner modes.

To summarize, we explained how the BBH model is an example of chiral higher-order insulator with one topologically protected zero mode that fits into the mode-shell correspondence paradigm.

This is a case where a full semi-classical limit cannot be obtained and where analytical expressions of the shell invariant are therefore difficult. However we used this last example to show how the mode and shell invariants can, in general, be computed numerically. This is a strength of the formalism as, even if the semi-classical limit is useful to obtain closed analytical expressions for the different pedagogical examples we discussed extensively throughout this paper, it is not mandatory to address the zero-modes.

## 5 Conclusion

In this paper, we have proposed a unifying mode-shell correspondence theory, a powerful tool in topological physics and in particular in the topology of wave operators. This correspondence relates a spectral property of gapless systems in localised regions of phase space – here the chiral number of zero-modes – to another topological invariant associated to a gapped operator on the shell surrounding this region in phase space. This correspondence is particularly useful since a semi-classical limit of the shell invariant can be derived in a lot of (but not all) situations. This limit simplifies the expression of the shell invariant into (higher-dimensional) winding numbers which are easier to calculate analytically.

We have shown that this correspondence unifies several results in wave topology, from the bulk-edge correspondence, higher-order topological phases, to the Callias index formula and the Atiyah-Singer index theory. We provided a wide variety of examples, for discrete and continuous systems, in one and higher dimensions, to illustrate the mode-shell correspondence in concrete models. In particular, we showed how the mode-shell correspondence describes not only zero-energy edge states, but also zero-modes that can be more generally localized in a region of phase space. We also discussed two systematic methods, dubbed "additive" and "multiplicative" tensor products constructions, to simply elaborate topological examples in higher $D > 1$ dimensions.

In this paper, we focused on the case where the mode index $\mathcal{I}_{\mathrm{modes}}$ is the chiral number of zero-dimensional zero-energy modes, since it already encompasses a rich variety of situations which deserved a complete study. Similar mode-shell correspondences can be derived in cases where the mode index is instead the $1D$-spectral flow invariant, as for edge states of the $2D$ quantum hall effect, or where the index gives the number of $2D$-Dirac or $3D$-Weyl points, which can either be localised in wavenumber or be surface states of respectively $3D$ of $4D$-insulators. Those examples are planned to be addressed in the part II of this work that will establish the mode-shell correspondence in higher dimensions for chiral and unitary classes.

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

## A    Smoothness/fast-decay Fourier duality

In this section, we recall some results about the links between the regularity of a function and the fast decay of its Fourier transform/series at infinity. These properties are fundamental for our purpose and are wildly used in the main text to make bridges between the dual behavior in position and wavenumber spaces.

Let $f$ be a function of one variable $x \in \mathbb{R}$, then one can define its Fourier transform $\tilde{f}$ as a function of the variable $k \in \mathbb{R}$ as

$$\tilde{f}(k) = \int dx f(x) e^{-ikx} \ . \tag{80}$$

If, $f$ is a periodic function with $x \in [0, 2\pi]$, one can associate to $f$ a Fourier series $\tilde{f}(k)$ with discrete parameter $k \in \mathbb{Z}$ such that

$$\tilde{f}(k) = \frac{1}{2\pi} \int_0^{2\pi} dx f(x) e^{-ikx} \ . \tag{81}$$

Importantly, for both Fourier transforms and series, the smoother $f$ is, the faster $\tilde{f}$ decays when $|k| \to \infty$. The first result of this kind is the Riemman-Lebesge lemma that states that if $f$ is a continuous integrable function, then its Fourier transform or its Fourier series goes zero at infinity:

$$\tilde{f}(k) \xrightarrow{|k| \to +\infty} 0 \tag{82}$$

Combined with the well-known Fourier transform/series property that $(\widetilde{\partial_x f})(k) = ik\tilde{f}(k)$ this lemma implies that if $f$ is a smooth function with $N$ derivatives which are continuous, then its

Fourier transform must decays faster than $k^{-N}$

$$\tilde{f}(k) = \frac{1}{(ik)^N}(\partial_x^{\tilde{N}} f)(k) = O(\frac{1}{k^N}) \ . \tag{83}$$

Moreover, since the Fourier transform is invertible, this property can be inverted and implies that if the Fourier transform of a function decays sufficiently fast at infinity, then the initial function must be smooth (with enough derivatives).

An example of how these considerations can be useful to study the properties of operators is the following: Let $f$ and $g$ be two periodic functions on $[0, 2\pi]$ and consider the simple Hamiltonian $\hat{H}$ such that

$$(\hat{H}f)(x) = g(x)f(x) \ . \tag{84}$$

Then, if one rewrites the operator in wavenumber space, we would have instead

$$(\hat{H}\tilde{f})(k) = \sum_{k'} \tilde{f}(k - k')\tilde{g}(k') \tag{85}$$

so we see that the multiplication by a function in the initial space translates as a hopping problem in the dual space, where a hopping of a distance $n$ has a strength $\hat{g}(k)$. The regularity/fast-decay is then quite useful since it implies that if $g$ is a smooth function (say $C^\infty$) then $\hat{g}(k)$ decays fastly, making the dual problem a short-range one (and vice-versa), which constrains a lot the properties of the dual problem.

These properties are important to relate the short-range behavior of the operator with the smoothness of the Wigner-Weyl transform, which is the goal of the next appendix.

# B  Wigner-Weyl transform

## B.1  The continuous version

In physics, there are a lot of situations where the relevant information of the system is encoded by continuous (vector valued) functions of $x \in \mathbb{R}^d$, and the properties of the systems are described by operators which act on this space of function. In those cases, the Wigner-Weyl transform is a powerful tool which allows ourselves to map these operators into the simpler objects which are the (matrix valued) functions on the phase space $(x, k) \in \mathbb{R}^d \times \mathbb{R}^d$.

This transformation can be described as follows. If $\hat{A}$ is an operator which acts on a function $f$, then we can associate to this operator $\hat{A}$ the function $A(x, k)$ which is called the "symbol of the operator" such that

$$A(x, k) = \int_{\mathbb{R}^d} dx'^d \left\langle x + x'/2 \right| \hat{A} \left| x - x'/2 \right\rangle e^{-ikx'} \ . \tag{86}$$

where $\langle x' | \hat{A} | x \rangle$ is in general called the "kernel" of $\hat{A}$. If one is not comfortable with the notations of quantum mechanics, the kernel can also be denoted $\hat{A}_{x,x'}$ and can be determined as the value in $x'$ of the function $\hat{A}\delta_x$ where $\delta_x$ is the Dirac function centered in $x$, $x' \to \delta(x' - x)$. This kernel may not always be a well defined function in $x$ and $x'$ (see for example $\hat{A} = \partial_x$) but is in general well defined as a distribution. In that case, one may compute the symbol by first regularising the operator with a parameter $\epsilon$ (for example $\hat{A}_\epsilon = \int_x (|x\rangle \langle x + \epsilon| - |x\rangle \langle x - \epsilon|)/(2\epsilon)$), compute the symbol for the regularised operator and then take the limit $\epsilon \to 0$. Using this, one can show that the symbol of $\partial_x$ is $ik$.

Since $A(x, k)$ is defined using a Fourier transform, this formula can be inverted and from the symbol one can determine the operator $\hat{A}$ using the formula

$$\langle x' | \hat{A} | x \rangle = \int_{\mathbb{R}^d} \frac{dk^d}{(2\pi)^d} \, A(\frac{x + x'}{2}, k) e^{ik(x' - x)} \ . \tag{87}$$

These definitions are extended straightforwardly to the case where functions are vectors with internal degrees of freedom $\alpha$. The symbol $A(x, k)$ is replaced by a matrix valued symbol $A(x, k) = (A_{\alpha,\alpha'}(x, k))_{\alpha,\alpha'}$ and $\langle x' | \hat{A} | x \rangle$ should now be interpreted as a matrix $(\langle x', \alpha | \hat{A} | x, \alpha' \rangle)_{\alpha,\alpha'}$.

A first property associated to this Wigner-Weyl transform is its behavior relative to the trace. Indeed one can show that

$$\operatorname{Tr} \hat{A} = \int_{\mathbb{R}^d} dx^d \, \operatorname{Tr}^{\text{int}} \langle x | \hat{A} | x \rangle = \int_{\mathbb{R}^d} dx^d \int_{\mathbb{R}^d} \frac{dk^d}{(2\pi)^d} \, \operatorname{Tr}^{\text{int}} A(x, k) \tag{88}$$

where $\operatorname{Tr}^{\text{int}}$ is the reduced trace only over the internal degrees of freedom $\alpha$.

An other important property of Weyl-Wigner calculus we need to introduce is that dealing with product of operators. If we define the *Moyal $\star$-product* of two symbols $A \star A'$ as the symbol of their operator $\hat{A} \circ \hat{A}'$, one can verify that such product can be written as

$$(A \star A')(x, k) = \int_{((\mathbb{R}^d)^4)} dx'^d dx''^d \frac{dk'^d}{\pi^d} \frac{dk''^d}{\pi^d} A(x + x', k + k') \, A'(x + x'', k + k'') e^{i2(x'k'' - x''k')}. \tag{89}$$

This formula can be proved by writing the formula of the symbol of $\hat{A} \circ \hat{A}'$ in function of the kernel $\langle x' | \hat{A} \circ \hat{A}' | x \rangle$. This kernel can then itself be expanded as $\langle x' | \hat{A} \circ \hat{A}' | x \rangle = \sum_{x''} \langle x' | \hat{A} | x'' \rangle \langle x'' | \hat{A}' | x \rangle$. Finally one uses the inversion formula for $\langle x' | \hat{A} | x'' \rangle$ and $\langle x'' | \hat{A}' | x \rangle$, to recover an equation which only includes symbols and which can be reduced to (89) up to some suitable changes of variables in $x'$ ,$x''$, $k'$, $k''$.

If the symbols $A(x + x', k + k')$ and $A'(x + x'', k + k'')$ are both smooth in $x$ and $k$ this expression can then be expanded in terms involving higher order derivatives. To do so, let us use the Taylor expansion

$$A(x + x', k + k') = \sum_{j,j'=0}^{j+j'=n} \partial_x^j \partial_k^{j'} A(x, k) \frac{x'^j}{j!} \frac{k'^{j'}}{j'!} + R(x, k) \tag{90}$$

where $R(x, k)$ decays faster than $\|(x, k)\|^n$ for $x'$ and $k'$ small. If we substitute this expansion into the Moyal product, we can write that $x'^k e^{i2x'k''} = \frac{1}{(2i)^j} \partial_{k''} e^{i2x'k''}$ or $k'^{j'} e^{-i2x''k'} = \frac{1}{(-2i)^{j'}} \partial_{x''} e^{-i2x''k'}$ and integrate by part. We then obtain an expansion of the Moyal product which is

$$(A \star A')_{\text{sym}}(x, k) = \sum_{j,j'=0}^{j+j'=n} \frac{(-1)^j}{j! j'! (2i)^{j+j'}} \partial_x^j \partial_k^{j'} A(x, k) \partial_x^{j'} \partial_k^j A'(x, k) + \text{Error} \tag{91}$$

where the error term can be explicitly bounded using the derivatives in $n + 1$ in $x$ and $n' + 1$ in $k$ of both symbols. From this expansion, one can notice that if both symbols are constant in $x$, or in $k$, then the Moyal product is simplified a lot since it reduces to the simple product of the symbols $(A \star A')(x, k) = A(x) A'(x)$ or $(A \star A')(x, k) = A(k) A'(k)$.

Importantly, for slow-varying symbols, we can perform a semi-classical expansion of the Moyal product. For example, if we re-scale the symbols in one of the variables like $A_\epsilon(x, k) = A(\epsilon x, k)$

or $A_\epsilon(x,k) = A(x,\epsilon k)$ so that the symbols look like almost constant in that variable in the limit $\epsilon \to 0$, then the expansion (91) becomes a perturbative expansion in $\epsilon$ in that limit and we have

$$(A_\epsilon \star A'_\epsilon)_{\frac{1}{\epsilon}}(x,k) = \sum_{j,j'=0}^{j+j'=n} \frac{(-1)^j \epsilon^{j+j'}}{j!j'!(2i)^{j+j'}} \partial_x^j \partial_k^{j'} A(x,k) \partial_x^{j'} \partial_k^j A'(x,k) + O(\epsilon^{n+1}) . \tag{92}$$

In particular, we obtain that for slow-varying fields, the product of operators can be replaced by the product of their symbol at lowest (zeroth) order in $\epsilon$

$$(A_\epsilon \star A'_\epsilon)_{\frac{1}{\epsilon}}(x,k) = A(x,k)A'(x,k) + O(\epsilon) . \tag{93}$$

Also, the symbol of a commutator of operators $[\hat{A}, \hat{A}']$ is given by the Moyal commutator $[A,B]_\star \equiv A \star A' - A' \star A$ of their symbols. A useful approximation is obtained in the same as above by keeping now the first order in $\epsilon$ in the expansion

$$[A_\epsilon, A'_\epsilon]_{\star,\frac{1}{\epsilon}}(x,k) = [A(x,k), A'(x,k)] - \frac{\epsilon}{2i} \left( [\partial_k A(x,k), \partial_x A'(x,k)]_+ - [\partial_x A(x,k), \partial_k A'(x,k)]_+ \right) + O(\epsilon^2)$$
$$\equiv [A(x,k), A'(x,k)] + \epsilon i \{A, A'\}(x,k) + O(\epsilon^2) \tag{94}$$

where $[B,C]_+ = BC + CB$ is just the usual symmetric sum (also called anti-commutator) and $\{A,B\}$ the Poisson bracket of the symbols. When $A(x,k)$ and $A'(x,k)$ commutes with each other (for example when $A(x,k)$ or $A'(x,k)$ is scalar, as it is often the case in the paper), one recovers the famous result that, at first order in $\epsilon$, the symbol of the commutator is the Poisson bracket of the symbols.

**Criteria for the semi-classical limit.** In order to estimate how good the semi-classical approximation $A \star A' \sim AA'$ is for two symbols $A$ and $A'$, one needs to check how small the first order correction $(\partial_x A \partial_k A' - \partial_x A \partial_k A')(x,k)$ is compared to the product $(AA')(x,k)$. If we introduce the quantities $1/d_{x/k,A}(x,k) \equiv \|\partial_{x/k} A(x,k)\| \|A^{-1}(x,k)\|$, we can define the parameter $\epsilon \equiv 1/(d_{x,A}d_{k,A'}) + 1/(d_{k,A}d_{x,A'})$ which measures how small is the first order correction compared to the standard product. When $\epsilon$ is small ($\epsilon \ll 1$), we can expect the semi-classical approximation to be good, as higher order corrections (like terms $\partial_x^2 A \partial_k^2 A'$ with higher derivative) should be even smaller (here of order $\epsilon^2$)[9]. Vice-versa when $\epsilon$ is not small, there is no reason to think that the semi-classical approximation is a good one.

In this paper, we deal with mainly two types of Moyal products: one involving the symbol of the Hamiltonian with the symbol of the cut-off operator $H \star \theta_\Gamma$, and one with only the symbol of the Hamiltonian $H \star H$ (in the computation of the flatten Hamiltonian $H_F$). $\theta_\Gamma$ can always be constructed to be slowly varying in the limit $\Gamma \to +\infty$ such that $1/d_{x/k,\theta_\Gamma} = O(1/\Gamma)$, so the semi-classical approxilation is always valid for the Moyal product $H \star \theta_\Gamma$ in that limit.

The problematic products that need to be verified are the $H \star H$ ones. There, the semi-classical criteria is $1/(d_{x,H}d_{k,H}) \ll 1$, which is the criteria given at the end of the section 2.5, using the fact that if $H(x,k)$ has a gap $\Delta(x,k)$, then $\|H^{-1}(x,k)\| = 1/\Delta(x,k)$.

---

[9]It is possible to fine-tune examples where in some points of phase space, the first order corrections are small but not the second order correction. However those examples are not really representative and our simple criteria will be enough to understand why the semi-classical approximation is valid or not in all situations we encounter in this paper

## B.2  The discrete version

For discrete systems, which are defined on a lattice $\mathbb{Z}^d$ rather than a continuous space $\mathbb{R}^d$, the usual Wigner-Weyl transform is not suitable and should instead be replaced by a discrete. If $\hat{A}$ is an operator acting on such a lattice, one can define its discrete symbol $A(n,k)$ as

$$A(x,k) = \sum_{n' \in \mathbb{Z}^d} \langle n + n' | \hat{A} | n \rangle e^{-ikn'} \tag{95}$$

where $n \in \mathbb{Z}^d$ is a discrete index, and $k \in [0, 2\pi]^d = \mathbb{T}^d$ is the equivalent of the Brillouin zone parameter.

When defining the discrete transform, one cannot choose a symmetric convention similar to the continuous one $\langle x + x'/n | \hat{A} | x - x'/2 \rangle$ mainly because if $n$ and $n'$ are sites of the lattice, their average $(n+n')/2$ is not guaranteed to be a site of the lattice. Therefore, every discrete conventions do not have all of the properties of the continuous ones. For example, having $\hat{A} = \hat{A}^\dagger$ does not imply $A^\dagger(x,k) = A(x,k)$.

However, the discrete transform still has good enough properties for the purpose of this article. The most important property is the fact that it is still a transform which maps, in a one to one correspondence, operators and symbols in phase space, with an inverse map given by

$$\langle n' | \hat{A} | n \rangle = \int_{\mathbb{T}^d} \frac{dk^d}{(2\pi)^d} A(n,k) e^{ik(n'-n)} \ . \tag{96}$$

We also have a trace property which is quite similar to that of the continuous case

$$\mathrm{Tr}\,\hat{A} = \sum_{n \in \mathbb{Z}^d} \mathrm{Tr}^{\mathrm{int}} \langle n | \hat{A} | n \rangle = \sum_{n \in \mathbb{Z}^d} \int_{\mathbb{T}^d} \frac{dk^d}{(2\pi)^d} \mathrm{Tr}^{\mathrm{int}} A(x,k) \ . \tag{97}$$

The Moyal product associated to this discrete Wigner-Weyl transform is a little bit different and can be expressed as

$$(A \star A')(n,k) = \sum_{n' \in \mathbb{Z}^d} \int_{(\mathbb{R}^d} \frac{dk'^d}{(2\pi)^d} A(n+n',k)\, A'(n,k+k') e^{in'k'} \tag{98}$$

from which we could also theoretically make a semi-classical expansion when the symbol is slowly varying in the $n$ direction. Since the expressions become quickly more cumbersome than in the continuous case, we will focus only on the zeroth and first orders of the expansion. For that purpose, we recall that if $\delta_n f(n) = f(n+1) - f(n)$, then for slowly varying function $f$, $f(n+n') \approx f(n) + \delta_n f(n) n' + \dots$. If we substitute that expression into the Moyal product we obtain

$$(A \star A')(n,k) \approx A(n,k)A'(n,k) + i\delta_n A(n,k)\partial_k A'(n,k) \ . \tag{99}$$

As a result, for slowly varying symbols, the leading coefficient of the symbol of a product of operators is just the product of symbols

$$AA'(n,k) \approx A(n,k)A'(n,k) \tag{100}$$

and the first order expansion of the symbol of a commutator is

$$\begin{aligned}
[A, A']_\star (n,k) &\approx [A(n,k), A'(n,k)] + i\left(\delta_n A(n,k)\partial_k A'(n,k) - \partial_k A(n,k)\delta_n A'(n,k)\right)\\
&\approx [A(n,k), A'(n,k)] + i\{A, A'\}(n,k)
\end{aligned} \tag{101}$$

which is again just a Poisson bracket when the symbols commute.

Since we only look at the leading coefficients of the semi-classical expansions, these properties are enough for the purpose of this paper. In particular, the semi-classical limit will be the same as in the continuous case, up to the fact that the continuous derivatives $\partial_x$ are replaced by discrete derivatives $\delta_n$.

## C  Short-range couplings in phase space

In multiple occasions, in the main text, we need the Hamiltonian to be local, meaning that we are allowed to neglect couplings between different separated regions which are far enough away from each other in phase space. For this property to hold, we need of course to assume that the initial Hamiltonian is local i.e. its coefficients in position representation $\hat{H}_{x,y} = \langle x | \hat{H} | y \rangle$ decay exponentially (or algebraically) with the distance $|x - y|$. The appendices A and B show us that this is linked to the assumption that the symbol $H(x, k)$ is smooth in the $k$ variable.

The same assumption can also be made in wavenumber, by assuming that $\hat{H}$ is local in wavenumber space, that is $\hat{H}_{k,k'} = \langle k | \hat{H} | k' \rangle$ also decays fast with the distance $|k - k'|$. Similarly, this is related to the assumption that the symbol $H(x, k)$ is smooth in the $x$ variable.

However, those assumption are not the end of the story because, in our theory, we also need to manipulate operators which are derived from the flatten Hamiltonian $\hat{H}_F = f(\hat{H})$. Actually the local assumption on $\hat{H}$ is transferred to $\hat{H}_F$ if $f$ is a smooth function of the energy [139].

In this section we present an idea of the proof of those results. We then explain how simple continuous systems (the non-relativistic ones) violate such an hypothesis. We finally argue how short-range behavior can be recovered in such systems if we localise in phase-space.

The idea to transfer short-range behavior from $\hat{H}$ to $f(\hat{H})$ is to first use the formula

$$f(\hat{H}) = \int dt \tilde{f}(t) e^{it\hat{H}} \tag{102}$$

where $\tilde{f}$ is the Fourier transform of $f$ and $e^{it\hat{H}}$ is the usual time propagator. Such formula can be proved using the usual definition of the Fourier transform in a diagonal basis of $\hat{H}$.

This formulation with time propagators allows us to use the well-known result called the Lieb-Robinson bound, which states that if $\hat{H}_{x,y}$ decays exponentially fast, then $e^{it\hat{H}}_{x,y}$ is also bounded by

$$|e^{it\hat{H}}_{x,y}| \leq e^{tv - |x-y|/d} \tag{103}$$

where $v$ and $d$ are constant which depends on how fast the couplings decays. We also have equivalent inequalities for algebraic decays. For example it is simple to prove that in general

$$\|[e^{it\hat{H}}, \hat{A}]\| \leq t \|[\hat{H}, \hat{A}]\| \tag{104}$$

which implies, when $\hat{A} = \hat{X}$ is the position operator, that $|e^{it\hat{H}}_{x,y}| \leq t\|[\hat{H}, \hat{X}]\|/|x - y|$.

Then, combining those results with the fact that, when $f$ is smooth, $\tilde{f}(t)$ decays quickly, and therefore, that in the integral $\int dt \tilde{f}(t) e^{it\hat{H}}$ only the evolution operators $e^{it\hat{H}}$ with relatively small $t$ contributes, it is possible to show that $f(\hat{H})$ is also local.

We do not aim at digging too far into mathematical considerations and prefer to remain voluntarily a bit vague here. However, rigorous results can be found for example in [98, 139]. The key result we use is that, when the coefficients of $\hat{H}$ decay faster than any polynomial in position

or/and in wavenumber, then the coefficients of $f(\hat{H})$ can also be shown to decay faster than any polynomial, and it follows that the couplings over long distances can always be neglected.

**Hamiltonians that do not satisfy the Lieb-Robinson bound:**  Regarding the short range properties in phase space, it is worth noticing that there are relevant physical Hamiltonians that do not satisfy any common Lieb-Robinson bound because they do not verify the necessary hypothesis that $\|[\hat{H}, \hat{X}]\|$ is bounded.

Examples of such Hamiltonians are any non-relativistic Hamiltonians of the form $\hat{H} = \partial_x^2 + V(x)$ which have no naive short range property in position space because $[\hat{H}, \hat{X}] = 2\partial_x$ is not a bounded operator. Moreover if the potential is the harmonic oscillator $V(x) = x^2$, it can be also shown to not be local in wavenumber for the same reason in the wavenumber basis.

To understand more intuitively why no usual locality propery can be found, one can look at the classical limit of such Hamiltonians $\hat{H}(x, i\partial_x) \rightarrow H(x, p)$ where the operator $i\partial_x$ is replaced by a classical momentum $p$ and where the Schrödinger equation is replaced by the classical Hamilton-Jacobi equations for the average position and momentum of the state as

$$\begin{aligned} \partial_t x &= \partial_p H(x, p) \\ \partial_t p &= -\partial_x H(x, p) \quad . \end{aligned} \tag{105}$$

For example, if we take the Hamiltonian of a particle in an harmonic potential $H(x, p) = p^2/(2m) + kx^2/2)$ we obtain

$$\begin{aligned} \partial_t x &= p/m \\ \partial_t p &= -kx \end{aligned} \tag{106}$$

We can see that there is *a priori* no bound for how fast the position or the momentum can change through time because if we start with an initial state of arbitrary high momentum $p$, $\partial_t x = p/m$ will be large and position can change arbitrary fast, and vice versa if we start with an initial state of arbitrary high position, because the force $F = -kx$ is proportional to $x$, position will also change arbitrary fast. The system is not relativistic so velocities are not bounded.

The quantum Hamiltonian contains the previous system as a semi-classical limit, therefore $e^{it\hat{H}}$ could not *a forciori* have short-range property in space or in wavenumber because it would imply a bound on how fast position or momentum can change in the classical limit, which does not exist.

However, in the classical system, it is well known that we can recover a bound on the rate of change of position/momentum by bounding the initial energy of the system $E = p^2/(2m) + kx^2/2 \leq E_0$. In the mode-shell correspondence, the cut-off $\hat{\theta}_\Gamma$ of symbols $\theta_\Gamma \approx e^{-(x^2+p^2)/\Gamma^2}$ plays a similar role by bounding position and wavenumber as $x^2 + p^2 \lesssim \Gamma^2$. Therefore, in analogy to the classical case, one may be able to obtain a bound (relative to the cut-off parameter) on how fast information can travel in phase space and therefore obtain again the short range property necessary in the proof of the mode-shell correspondence.

We do not want to go into such kinds of technical aspects in our proof of the mode-shell correspondence but want to signal that this is the kind of difficulties one will have to face if one wants to prove the mode-shell correspondence with the rigorous standard of the mathematical community.

## D   Beyond 1D lattices with balanced unit cells

In our proof of the bulk-edge correspondence, in section 3.1, we make the assumption of unit cells with a balanced number of degrees of freedom $A$ and $B$ of opposite chirality $\sum_\alpha C_\alpha = n_A - n_B = 0$.

We discuss here two situations where this assumption is broken.

**Unit cells with unbalanced chirality**

A first way to break the assumption of balanced unit cells, is to consider crystals where the number of degrees of freedom of opposite chirality differs in the unit cell, that is $\sum_\alpha C_\alpha = n_A - n_B \neq 0$, as depicted in figure 20. In fact, this situation cannot happen in our framework because it is incompatible with our assumption that the Hamiltonian is gapped far from the edge/interface.

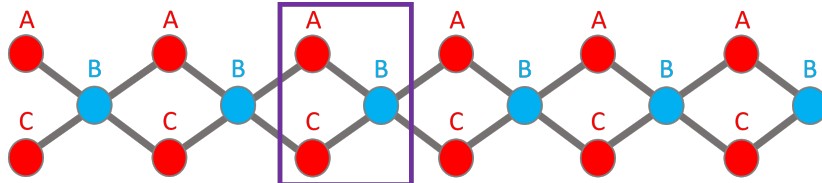

Figure 20: Example of a $1D$ chain with unbalanced unit-cells: the number of A sites (positive chirality, red) is greater than that of B sites (negative chirality, blue).

Indeed, under the bulk gap assumption the edge index $\mathcal{I}_{\mathrm{modes}}$ is finite and quantised [98] in the limit $\Gamma \to +\infty$. But, at the same time, the general relation (11) gives us $\mathcal{I} = \mathrm{Tr}\left(\hat{C}\hat{\theta}_\Gamma\right) + \frac{1}{2}\mathrm{Tr}\left(\hat{C}\hat{H}_F[\hat{\theta}_\Gamma, \hat{H}_F]\right)$, where, as we show in the main text, the term $\frac{1}{2}\mathrm{Tr}\left(\hat{C}\hat{H}_F[\hat{\theta}_\Gamma, \hat{H}_F]\right)$ converges to a difference of winding numbers and is therefore a finite integer. It follows that the last term, $\mathrm{Tr}\left(\hat{C}\hat{\theta}_\Gamma\right)$, must also be a finite integer. However, $\mathrm{Tr}\left(\hat{C}\hat{\theta}_\Gamma\right) = \sum_n \sum_\alpha C_\alpha e^{-n^2/\Gamma^2}$ roughly scales as $\sim (\sum_\alpha C_\alpha)\Gamma$ which tends to infinity in the limit $\Gamma \to +\infty$ if $\sum_\alpha C_\alpha \neq 0$. This contradiction implies that the unit cells must be balanced $\sum_\alpha C_\alpha = 0$, and that imbalanced unit cells are already implicitly incompatible with our initial hypothesis that the Hamiltonian is gapped far away in the bulk.

**Disordered chains without unit cells**

A second way to not satisfy the balanced unit cell assumption is simply to not having a unit cell. In physical systems, the translation invariant hypothesis of the bulk is not always satisfied as there is often disorder that hinder this regularity. One can then wonder what is the relevance of the semi-classical index in that case.

Let us first remember that the index $\mathcal{I}_{\mathrm{shell}}$ from which the semi-classical index is derived does not require any translation hypothesis to be quantised. Therefore, it remains quantised under any small/smooth perturbation, as long as the bulk gap is not closed[10]. One could thus first compute the semi-classical index in an idealised system, as it is easier to evaluate, and then extend the result by continuity.

When the disorder becomes too large, that is when the system cannot be in reasonably well approximated by a translation invariant system, one has no other choice than using the initial index (8). This can reasonably be done numerically in a finite size system, even if it is a little bit more time demanding. Typically, if the system has $N$ unit cells and $K$ internal degrees of freedom,

---

[10]Note that the addition of site of positive/negative chirality is always an abrupt perturbation of the system (there is an additional site or there is not). Therefore the chiral index may not and is not preserved under this kinds of change of the systems.

computing the index requires to diagonalise a matrix of size $N \times K$ which computationally require approximately $\approx O((N \times K)^3)$ numerical operations. This is to be compared with the semi-classical index which only require to diagonalise $M$ matrices but of reduced size $K$ (where $M$ is the number of points in the discretisation we make of the Brillouin zone or phase space). The semi-classical approximation thus allows for reducing the complexity to $O(M \times K^3)$ numerical operations, where importantly $M$ is not risen to the power 3.

This is why it is preferable to use the winding number when semi-classical analysis is possible in the bulk. Still, the computation of the original chiral invariant $\mathcal{I}_{\mathrm{modes}}$ remains viable numerically for $1D$ systems and is sometimes necessary for strongly disordered systems (see for example [98] for numerical computations of the index in a disorder system).

## E  Existence of topological modes on closed manifold of odd dimension: Less restrictive gap assumption than elliptic condition

In section 3.2, we mentioned that the mode-shell correspondence intersects Atiyah-Singer index theory in the case where modes are localised in wavenumber on bounded manifold. We then presented a 1D example where the manifold is a circle and where we obtained such modes. Actually, in the literature about the Atiyah-Singer theorem, it is stated that in odd dimension, the index of any differential operator should be zero. Therefore, it may be surprising that a model like (37), in $D = 1$, exists. This apparent contradiction can be explained by the fact that Atiyah-Singer theorem makes the assumption that $\hat{H}$ is *elliptic* [93, 95, 96] which requires that the polynomial expression of the symbol $H$ with highest degree in $k$, called the *principal symbol* $H_{\mathrm{pr}}$, is invertible (i.e. gapped in our vocabulary) when $k \neq 0$.

Indeed, a principal symbol $H_{\mathrm{pr}}$ of order $n$ in $k$ always have the symmetry $H_{\mathrm{pr}}(x, -k) = (-1)^n H_{\mathrm{pr}}(x, k)$ because $(-k)^n = (-1)^n k^n$. Therefore, when a principal symbol is gapped, we have the "asymptotic" symmetry $H_F(x, -k) \underset{\mathrm{lim}}{=} H_{\mathrm{pr},F}(x, -k) = (-1)^n H_{\mathrm{pr},F}(x, k) \underset{\mathrm{lim}}{=} (-1)^n H_F(x, k)$ when $|k| \to +\infty$ and thus $U(x, -k) = (-1)^n U(x, k)$. Substituting this relation in the formula (36) implies $\mathcal{I} = -\mathcal{I}$ so that the chiral index vanishes. In the more general case, given by (14), the shell can be a $D$-dimensional torus and we obtain in that case $\mathcal{I} = (-1)^D \mathcal{I}$. We thus recover the result that the index should vanish in odd dimension.

This conclusion however does not hold for our model (37), because our theory lies on gap assumption, which is less restrictive than the an elliptic assumption. In particular, the principal symbol of our model (37) reads $H_{\mathrm{pr}} = i\epsilon k \sin(x)\sigma_y$ (with $\sigma_y$ the standard Pauli matrix), which is not uniformly gapped, because the gap closes for $k \neq 0$ in $x = 0$ and $x = \pi$. The elliptic condition is thus broken. Instead, we have considered the full symbol $H(x, p)$ of $\hat{H}$ that also includes the component of order zero in $k$ and that satisfies our gap assumption (see (38)). The topological properties of such systems are thus not captured if we impose the elliptic condition. The gap condition of the full symbol, therefore, allows for more topological models to exist.

## F  Derivation of the semi-classical expression (14) of the shell index for chiral zero-modes

Here we provide a derivation of the semi-classical expression of the shell index as a generalized winding number (14) for a chiral symmetric Hamiltonian $[\hat{H}_F, \hat{C}]_+ = \hat{H}_F \hat{C} + \hat{C}\hat{H}_F = 0$ defined on

either a continuous or discrete space.

## Mathematical framework and strategy of the demonstration

Let us set to $K$ the number of internal degrees of freedom and let us consider that the space is continuous in $m$ dimensions, and discrete in $n$ dimensions, so that the total dimension of the system is $m + n = D$. To each continuous dimension $j$, we assign a position operator $x_j$ and a wavenumber operator $\partial_{x_j}$ which verify $[\partial_{x_j}, x_j] = \mathbb{1}$. Similarly, for each discrete dimension $j$, we introduce a translation operator $T_j$ and a position operator $n_j$ which verify $T_j^\dagger n_j T_j - n_j = \mathbb{1}$. To treat discrete and continuous spaces on the same footing, we introduce the operator $\hat{a}_j$ such that $\hat{a}_{2j} = x_j$ and $\hat{a}_{2j+1} = i\partial_{x_j}$ in the continuous case and $\hat{a}_{2j} = T_j^\dagger n_j$ and $\hat{a}_{2j+1} = -iT_j$ in the discrete case. This way, we obtain a single commutation relation $[\hat{a}_{2j}, \hat{a}_{2j+1}] = -i\,\mathbb{1}$ for both continuous and discrete spaces.

Next, we introduce a differential form formalism in phase space to automatically anti-symmetrize our expressions, which will simplify the calculations later. Phase space is spanned by $x^j$ and $k^j$ coordinates. Forms on phase space will be decomposed over $dx^j$ and $dk^j$, and their exterior $\wedge$-products that verify the anti-symmetric exterior algebra $dz^j \wedge dz^i = -dz^i \wedge dz^j$, where $z$ stands for both $x$ and $k$. We then introduce the operator-valued 1-form

$$\hat{\alpha} \equiv \sum_{j=1}^{D} \hat{a}_{2j} dx^j + \hat{a}_{2j+1} dk^j \tag{107}$$

whose square is related to the canonical symplectic form $\omega = \sum_{j=1}^{D} dk^j \wedge dx^j$ as

$$\hat{\alpha}^2 = i\,\mathbb{1}\,\omega \tag{108}$$

from which we deduce

$$\hat{\alpha}^{2D} = D!\,i^D \mathbb{1}\,\omega^D \tag{109}$$

where $\mathbb{1}$ is there to emphasize that $\hat{\alpha}^2$, and thus any even power of $\hat{\alpha}$, and in particular $\hat{\alpha}^{2D}$, are not operator-valued anymore; they commute with any operator and can even be inserted into operator-valued expressions for free, a trick we will use below. Instead, those forms are simply 2-form and $2D$-form respectively. In particular, $\hat{\alpha}^{2D}$ has the higher possible rank for a form in phase space, and has thus only one component proportional to the element basis $dk^1 \wedge dx^1 \wedge \cdots \wedge dk^D \wedge dx^D$. In order to re-express the chiral index $\mathcal{I}_{\text{shell}} = \mathcal{I}$, we shall thus interpret it as the coefficient of the operator-valued differential form $\iota = \mathcal{I}\omega^D$ that has also the same unique element basis, so that $\mathcal{I}$ is the unique component of this $2D$-form. Recalling

$$\mathcal{I} = \text{Tr}\left(\hat{C}\hat{\theta}_\Gamma\right) + \frac{1}{2}\,\text{Tr}\left(\hat{C}\hat{H}_F[\hat{\theta}_\Gamma, \hat{H}_F]\right) \tag{110}$$

one can insert (109) into it to get

$$\iota = \frac{(-i)^D}{D!}\left(\text{Tr}\left(\hat{C}\,\hat{\alpha}^{2D}\,\hat{\theta}_\Gamma\right) + \frac{1}{2}\,\text{Tr}\left(\hat{C}\,\hat{\alpha}^{2D}\,\hat{H}_F[\hat{\theta}_\Gamma, \hat{H}_F]\right)\right)\ . \tag{111}$$

Interestingly, the second term of this expression involves a commutator $[\hat{\theta}_\Gamma, \hat{H}_F]$ which takes non vanishing value only on the shell where, by definition, $\hat{\theta}$ is not constant. We can make apparent a

similar commutator on the first term also, by using the *odd trace cyclicity* property for $\hat{\alpha}$ with an operator $\hat{A}$

$$\mathrm{Tr}\left(\hat{\alpha}\,\hat{\alpha}^{2D-1}\,\hat{A}\right) = -\,\mathrm{Tr}\left(\hat{\alpha}^{2D-1}\,\hat{A}\,\hat{\alpha}\right) \tag{112}$$

where the minus sign is due to the fact that $\hat{\alpha}$ is a form. One obtains

$$\iota = \frac{(-i)^D}{D!\,2}\left(\mathrm{Tr}\left(\hat{C}\,\hat{\alpha}^{2D-1}[\hat{\alpha},\hat{\theta}_\Gamma]\right) + \mathrm{Tr}\left(\hat{C}\,\hat{\alpha}^{2D}\,\hat{H}_F[\hat{\theta}_\Gamma,\hat{H}_F]\right)\right) \ . \tag{113}$$

Now both terms of this expression involve a commutator of the form $[\hat{A},\hat{\theta}_\Gamma]$. Those commutators peak the value of each trace in regions of phase space where $H_F$ is gapped, because $\hat{\theta}_\Gamma \approx \mathbb{1}$ or $\hat{\theta}_\Gamma \approx 0$ where $H_F$ is gapless (which leads to a vanishing commutator). Moreover, in the gapped region of phase space, we have $\hat{H}_F^2 \approx \mathbb{1}$. Therefore, all the operators of the forms $(1-\hat{H}_F^2)\hat{B}[\hat{A},\hat{\theta}_\Gamma]$ or $[1-\hat{H}_F^2,\hat{\theta}_\Gamma]$ become quickly negligible when $\Gamma$ tends to infinity, typically faster than any power $1/\Gamma^n$ in $\Gamma$ (as long as $\hat{A}$ and $\hat{B}$ are local operators in phase space). In the rest of the proof, all the equalities we write should be understood as approximations up to these fast decaying corrections.

The strategy of the demonstration is to use such properties to show by induction that

$$\iota = \frac{(-1)^l l!^2 (-i)^D}{D!(2l)!}\left(\mathrm{Tr}\left(\hat{C}\,\hat{\alpha}^{2D-2l}\,\hat{H}_F[\hat{\alpha},\hat{H}_F]^{2l-1}[\hat{\alpha},\hat{\theta}_\Gamma]\right) + \frac{1}{2}\mathrm{Tr}\left(\hat{C}\,\hat{\alpha}^{2D-2l}\,\hat{H}_F[\hat{\alpha},\hat{H}_F]^{2l}[\hat{\theta}_\Gamma,\hat{H}_F]\right)\right) \tag{114}$$

for $l \leq D$ and then apply it for $l = D$. Then, we expand semi-classically this expression by taking the limit $\Gamma \to \infty$ to find the relation between the chiral index and the generalized winding numbers (14). In the following, we detail the derivation into three steps : (1) the proof of the initialisation for $l = 1$, (2) the proof of the induction by deriving the formula for $l+1$ from that for $l$, and finally (3) perform the semi-classical expansion for $l = D$.

**Initialisation of the induction, $l = 1$**

Initializing the induction amounts to show that

$$\iota = -\frac{(-i)^D}{D!\,2}\left(\mathrm{Tr}\left(\hat{C}\,\hat{\alpha}^{2D-2}\,\hat{H}_F[\hat{\alpha},\hat{H}_F][\hat{\alpha},\hat{\theta}_\Gamma]\right) + \frac{1}{2}\mathrm{Tr}\left(\hat{C}\,\hat{\alpha}^{2D-2}\,\hat{H}_F[\hat{\alpha},\hat{H}_F]^2[\hat{\theta}_\Gamma,\hat{H}_F]\right)\right) \ . \tag{115}$$

To do so, let us start by massaging the first trace in (113) to reveal the commutator $[\hat{\alpha},\hat{H}_F]$. One can insert $\hat{H}_F$ in that expression by using $\hat{H}_F^2 \approx \mathbb{1}$ on the shell, where the value of the trace is picked, and we can write

$$\mathrm{Tr}\left(\hat{C}\,\hat{\alpha}^{2D-1}[\hat{\alpha},\hat{\theta}_\Gamma]\right) = \mathrm{Tr}\left(\hat{C}\,\hat{\alpha}^{2D-2}\,\hat{H}_F^2\,\hat{\alpha}[\hat{\alpha},\hat{\theta}_\Gamma]\right) \tag{116}$$

in that region. Recalling that $\hat{H}_F$ anti-commutes with $\hat{C}$ and commutes with $\hat{\alpha}^2$, we can use its cyclicity in the trace to get

$$\mathrm{Tr}\left(\hat{C}\,\hat{\alpha}^{2D-1}[\hat{\alpha},\hat{\theta}_\Gamma]\right) = -\frac{1}{2}\mathrm{Tr}\left(\hat{C}\,\hat{\alpha}^{2D-2}\,\hat{H}_F[\hat{\alpha}[\hat{\alpha},\hat{\theta}],\hat{H}_F]\right) \ . \tag{117}$$

The commutator in this expression can be expanded by using twice the general identity $[AB, C] = [A, C]B + A[B, C]$ for operators $A$, $B$ and $C$, such that

$$[\hat{\alpha}[\hat{\alpha}, \hat{\theta}], \hat{H}_F] = [\hat{\alpha}, \hat{H}_F][\hat{\alpha}, \hat{\theta}_\Gamma] + \hat{\alpha}[[\hat{\alpha}, \hat{\theta}_\Gamma], \hat{H}_F] \tag{118}$$

$$= [\hat{\alpha}, \hat{H}_F][\hat{\alpha}, \hat{\theta}_\Gamma] + \hat{\alpha}[[\hat{\alpha}, \hat{H}_F], \hat{\theta}_\Gamma] + \hat{\alpha}[\hat{\alpha}, [\hat{\theta}_\Gamma, \hat{H}_F]] \,. \tag{119}$$

At this stage, the differential form expression of the index now reads

$$\iota = \frac{(-i)^D}{D!\,4} \left( -\operatorname{Tr}\!\left( \hat{C}\,\hat{\alpha}^{2D-2}\,\hat{H}_F([\hat{\alpha}, \hat{H}_F][\hat{\alpha}, \hat{\theta}_\Gamma] + \hat{\alpha}[[\hat{\alpha}, \hat{H}_F], \hat{\theta}_\Gamma] + \hat{\alpha}[\hat{\alpha}, [\hat{\theta}_\Gamma, \hat{H}_F]]) \right) \right.$$
$$\left. +2\operatorname{Tr}\!\left( \hat{C}\,\hat{\alpha}^{2D}\,\hat{H}_F[\hat{\theta}_\Gamma, \hat{H}_F] \right) \right) \tag{120}$$

where the desired term $\hat{H}_F[\hat{\alpha}, \hat{H}_F][\hat{\alpha}, \hat{\theta}_\Gamma]$ has appeared, together with two other terms that make the expression apparently more complicated, but that turn out to each other compensate after some additional algebraic manipulations. To see it, let us first focus on the last term of the first line, of the form $\hat{\alpha}[\hat{\alpha}, [\hat{\theta}_\Gamma, \hat{H}_F]]$, that we can rewrite by using the relation $B[A, C] = [A, B]_+ C - [A, BC]_+$ with $A = \hat{\alpha}$, $B = \hat{H}_F\,\hat{\alpha}$ and $C = [\hat{\theta}_\Gamma, \hat{H}_F]$ to get

$$\operatorname{Tr}\!\left( \hat{C}\,\hat{\alpha}^{2D-2}\,\hat{H}_F\,\hat{\alpha}[\hat{\alpha}, [\hat{\theta}_\Gamma, \hat{H}_F]] \right) = \operatorname{Tr}\!\left( \hat{C}\,\hat{\alpha}^{2D-2}[\hat{\alpha}, \hat{H}_F\,\hat{\alpha}]_+[\hat{\theta}_\Gamma, \hat{H}_F] \right)$$
$$- \operatorname{Tr}\!\left( \hat{C}\,\hat{\alpha}^{2D-2}[\hat{\alpha}, \hat{H}_F\,\hat{\alpha}[\hat{\theta}_\Gamma, \hat{H}_F]]_+ \right) \,. \tag{121}$$

The second term actually vanishes exactly. Indeed, by expanding its anti-commutator and using the commutation between $\alpha$ and the chiral operator $\hat{C}$, we have the two terms

$$\operatorname{Tr}\!\left( \hat{C}\,\hat{\alpha}^{2D-2}[\hat{\alpha}, \hat{H}_F\,\hat{\alpha}[\hat{\theta}_\Gamma, \hat{H}_F]]_+ \right) = \operatorname{Tr}\!\left( \hat{\alpha}\,\hat{C}\,\hat{\alpha}^{2D-2}\,\hat{H}_F\,\hat{\alpha}[\hat{\theta}_\Gamma, \hat{H}_F] \right) + \operatorname{Tr}\!\left( \hat{C}\,\hat{\alpha}^{2D-2}\,\hat{H}_F\,\hat{\alpha}[\hat{\theta}_\Gamma, \hat{H}_F]\,\hat{\alpha} \right)$$

that compensate each other because of the odd-cyclicity of the trace with $\hat{\alpha}$ (112). The anti-commutator in the remaining term of the right-hand side of (121) can in turn be expanded such as

$$\operatorname{Tr}\!\left( \hat{C}\,\hat{\alpha}^{2D-2}\,\hat{H}_F\,\hat{\alpha}[\hat{\alpha}, [\hat{\theta}_\Gamma, \hat{H}_F]] \right) = \operatorname{Tr}\!\left( \hat{C}\,\hat{\alpha}^{2D-2}(\hat{\alpha}\,\hat{H}_F\,\hat{\alpha} + \hat{H}\,\hat{\alpha}^2)[\hat{\theta}_\Gamma, \hat{H}_F] \right) \tag{122}$$

$$= \operatorname{Tr}\!\left( \hat{C}\,\hat{\alpha}^{2D-2}([\hat{\alpha}, \hat{H}_F]\,\hat{\alpha} + 2\hat{H}_F\,\hat{\alpha}^2)[\hat{\theta}_\Gamma, \hat{H}_F] \right) \tag{123}$$

$$= \operatorname{Tr}\!\left( \hat{C}\,\hat{\alpha}^{2D-2}[\hat{\alpha}, \hat{H}_F]\,\hat{\alpha}[\hat{\theta}_\Gamma, \hat{H}_F] \right) + 2\operatorname{Tr}\!\left( \hat{C}\,\hat{\alpha}^{2D}\,\hat{H}_F[\hat{\theta}_\Gamma, \hat{H}_F] \right) \tag{124}$$

where we have used the commutation of $\hat{H}_F$ with $\hat{\alpha}^2$. We thus find that the second term in the right-hand side of (124) exactly compensates the last term in (120), so that the form expression of the index becomes

$$\iota = -\frac{(-i)^D}{D!\,4} \operatorname{Tr}\!\left( \hat{C}\,\hat{\alpha}^{2D-2}(\hat{H}_F[\hat{\alpha}, \hat{H}_F][\hat{\alpha}, \hat{\theta}_\Gamma] + \hat{H}_F\,\hat{\alpha}[[\hat{\alpha}, \hat{H}_F], \hat{\theta}_\Gamma] + [\hat{\alpha}, \hat{H}_F]\,\hat{\alpha}[\hat{\theta}_\Gamma, \hat{H}_F]) \right) \,. \tag{125}$$

Let us now massage the second term in the sum. By using the relation $A[B, C] = [C, A]B + ABC - CAB$ with $A = \hat{H}_F\,\hat{\alpha}$, $B = [\hat{\alpha}, \hat{H}_F]$ and $C = \hat{\theta}_\Gamma$, one obtains

$$\operatorname{Tr}\!\left( \hat{C}\,\hat{\alpha}^{2D-2}\,\hat{H}_F\,\hat{\alpha}[[\hat{\alpha}, \hat{H}_F], \hat{\theta}_\Gamma] \right) = \operatorname{Tr}\!\left( \hat{C}\,\hat{\alpha}^{2D-2}[\hat{\theta}_\Gamma, \hat{H}_F\,\hat{\alpha}][\hat{\alpha}, \hat{H}_F] \right)$$
$$+ \operatorname{Tr}\!\left( \hat{C}\,\hat{\alpha}^{2D-2}\,\hat{H}_F\,\hat{\alpha}[\hat{\alpha}, \hat{H}_F]\,\hat{\theta}_\Gamma \right) - \operatorname{Tr}\!\left( \hat{C}\,\hat{\alpha}^{2D-2}\,\hat{\theta}_\Gamma\,\hat{H}_F\,\hat{\alpha}[\hat{\alpha}, \hat{H}_F] \right)$$
$$\tag{126}$$

where the two last terms compensate each other due to the commutation of an even power of $\hat{\alpha}$ with $\hat{\theta}_\Gamma$ and the cyclicity of the trace. The commutator $[C, A]B$ in the remaining first term of the (126) can then be written as

$$[C, A]B = (\hat{\theta}_\Gamma \, \hat{H}_F \, \hat{\alpha} - \hat{H}_F \, \hat{\alpha} \, \hat{\theta}_\Gamma - \hat{H}_F \, \hat{\theta}_\Gamma \, \hat{\alpha} + \hat{H}_F \, \hat{\theta}_\Gamma \, \hat{\alpha})B \tag{127}$$

$$= (-[\hat{H}_F, \hat{\theta}_\Gamma] \, \hat{\alpha} - \hat{H}_F[\hat{\alpha}, \hat{\theta}_\Gamma])[\hat{\alpha}, \hat{H}_F] \tag{128}$$

such that the form expression of the index reads

$$\iota = -\frac{(-i)^D}{D! \, 4} \operatorname{Tr} \hat{C} \, \hat{\alpha}^{2D-2} \left( \hat{H}_F[\hat{\alpha}, \hat{H}_F][\hat{\alpha}, \hat{\theta}_\Gamma] - \hat{H}_F[\hat{\alpha}, \hat{\theta}_\Gamma][\hat{\alpha}, \hat{H}_F] \right.$$
$$\left. -[\hat{H}_F, \hat{\theta}_\Gamma] \, \hat{\alpha}[\hat{\alpha}, \hat{H}_F] + [\hat{\alpha}, \hat{H}_F] \, \hat{\alpha}[\hat{\theta}_\Gamma, \hat{H}_F] \right) . \tag{129}$$

This expression can further be simplified since the two first terms, as well as the two last terms, turn out to be equal on the shell and thus add up. To see it, we use the fact that $\hat{H}_F$ anti-commutes with $[\hat{\alpha}, \hat{H}_F]$ on the shell, which follows from

$$[\hat{\alpha}, \hat{H}_F]\hat{H}_F = \hat{\alpha} \, \hat{H}_F^2 - \hat{H}_F \, \hat{\alpha} \, \hat{H}_F + \hat{H}_F^2 \, \hat{\alpha} - \hat{H}_F^2 \, \hat{\alpha} \tag{130}$$

$$= -\hat{H}_F[\hat{\alpha}, \hat{H}_F] + \underbrace{[\hat{\alpha}, \hat{H}_F^2]}_{\approx 0} \tag{131}$$

where the last term vanishes on the shell where $\hat{H}_F^2 \approx 1$. By cyclicty of the trace, we can now re arrange the second term in (129) as

$$- \operatorname{Tr}\left( \hat{C} \, \hat{\alpha}^{2D-2} \, \hat{H}_F[\hat{\alpha}, \hat{\theta}_\Gamma][\hat{\alpha}, \hat{H}_F] \right) = - \operatorname{Tr}\left( \hat{C} \, \hat{\alpha}^{2D-2}[\hat{\alpha}, \hat{H}_F]\hat{H}_F[\hat{\alpha}, \hat{\theta}_\Gamma] \right) \tag{132}$$

where we have used the odd-cyclicity of the trace with $\hat{\alpha}$ that provides a minus sign, the chiral symmetry of $\hat{H}_F$ that provides a second minus sign, and the commutation of $\hat{\alpha}^{2D-2}$ with any operator. The last step consists in using the anti-commutation of $\hat{H}_F$ with $[\hat{\alpha}, \hat{H}_F]$ as announced, that leads to

$$- \operatorname{Tr}\left( \hat{C} \, \hat{\alpha}^{2D-2} \, \hat{H}_F[\hat{\alpha}, \hat{\theta}_\Gamma][\hat{\alpha}, \hat{H}_F] \right) = + \operatorname{Tr}\left( \hat{C} \, \hat{\alpha}^{2D-2} \, \hat{H}_F[\hat{\alpha}, \hat{H}_F][\hat{\alpha}, \hat{\theta}_\Gamma] \right) \tag{133}$$

which coincides with the first term in (129). To now show that the third and fourth terms in (129) also add up, we now use the anti-commutation of $\hat{\alpha}$ with $[\hat{\alpha}, \hat{H}_F]$ which follows from

$$\hat{\alpha}[\hat{\alpha}, \hat{H}_F] = \hat{\alpha}^2 - \hat{\alpha} \, \hat{H}_F \, \hat{\alpha} + \hat{H}_F \, \hat{\alpha}^2 - \hat{H}_F \, \hat{\alpha}^2 \tag{134}$$

$$= -[\hat{\alpha}, \hat{H}_F] \, \hat{\alpha} + \underbrace{[\hat{\alpha}^2, \hat{H}_F]}_{=0} \tag{135}$$

where the last term vanishes because of (108). Combining this property with the chiral symmetry of $\hat{H}_F$ allows us to rewrite the third term in (129) as

$$- \operatorname{Tr}\left( \hat{C} \, \hat{\alpha}^{2D-2}[\hat{H}_F, \hat{\theta}_\Gamma] \, \hat{\alpha}[\hat{\alpha}, \hat{H}_F] \right) = \operatorname{Tr}\left( \hat{C} \, \hat{\alpha}^{2D-2}[\hat{\alpha}, \hat{H}_F] \, \hat{\alpha}[\hat{\theta}_\Gamma, \hat{H}_F] \right) \tag{136}$$

that corresponds to the last term in (129). The form expression of the index thus reads

$$\iota = -\frac{(-i)^D}{D! \, 2} \operatorname{Tr}\left( \hat{C} \, \hat{\alpha}^{2D-2}(\hat{H}_F[\hat{\alpha}, \hat{H}_F][\hat{\alpha}, \hat{\theta}_\Gamma] + [\hat{\alpha}, \hat{H}_F] \, \hat{\alpha}[\hat{\theta}_\Gamma, \hat{H}_F]) \right) . \tag{137}$$

The first term is the desired one. We thus still have to rearrange the second term to make apparent a second commutator $[\hat{\alpha}, \hat{H}_F]$, which requires to re-introduce another $\hat{H}_F$ in the expression. As previously, this can be done by using $\hat{H}_F \approx 1$ on the shell, to get

$$\text{Tr}\Big(\hat{C}\,\hat{\alpha}^{2D-2}[\hat{\alpha}, \hat{H}_F]\,\hat{\alpha}[\hat{\theta}_\Gamma, \hat{H}_F]\Big) = -\,\text{Tr}\Big(\hat{C}\,\hat{\alpha}^{2D-2}\,\hat{H}_F^2\,\hat{\alpha}[\hat{\alpha}, \hat{H}_F][\hat{\theta}_\Gamma, \hat{H}_F]\Big) \tag{138}$$

$$= \text{Tr}\Big(\hat{H}_F\hat{C}\,\hat{\alpha}^{2D-2}\,\hat{H}_F\,\hat{\alpha}[\hat{\alpha}, \hat{H}_F][\hat{\theta}_\Gamma, \hat{H}_F]\Big) \tag{139}$$

by chiral symmetry. After using the cyclicity of the trace for $\hat{H}_F$, one uses its anti-commutation with $[\hat{\alpha}, \hat{H}_F]$ together with its anti-commutation with $[\hat{H}_F, \hat{\theta}_\Gamma]$, that follows from

$$[\hat{\theta}_\Gamma, \hat{H}_F]\hat{H}_F = \hat{\theta}_\Gamma\,\hat{H}_F^2 - \hat{H}_F\,\hat{\theta}_\Gamma\,\hat{H}_F \tag{140}$$

$$= \hat{H}_F^2\,\hat{\theta}_\Gamma - \hat{H}_F\,\hat{\theta}_\Gamma\,\hat{H}_F \tag{141}$$

$$= -\hat{H}_F[\hat{\theta}_\Gamma, \hat{H}_F] \tag{142}$$

and the equality (139) becomes

$$\text{Tr}\Big(\hat{C}\,\hat{\alpha}^{2D-2}[\hat{\alpha}, \hat{H}_F]\,\hat{\alpha}[\hat{\theta}_\Gamma, \hat{H}_F]\Big) = \text{Tr}\Big(\hat{C}\,\hat{\alpha}^{2D-2}\,\hat{H}_F\,\hat{\alpha}\,\hat{H}_F[\hat{\alpha}, \hat{H}_F]\,\hat{\alpha}[\hat{\theta}_\Gamma, \hat{H}_F]\Big) . \tag{143}$$

Comparing (138) with (139) allows us to express the second term of (137) as

$$\text{Tr}\Big(\hat{C}\,\hat{\alpha}^{2D-2}[\hat{\alpha}, \hat{H}_F]\,\hat{\alpha}[\hat{\theta}_\Gamma, \hat{H}_F]\Big) = \frac{1}{2}\,\text{Tr}\Big(\hat{C}\,\hat{\alpha}^{2D-2}\,\hat{H}_F[\hat{\alpha}, \hat{H}_F]^2[\hat{\theta}_\Gamma, \hat{H}_F]\Big) \tag{144}$$

which, once inserted in (137), completes the derivation of the initialization of the induction (115).

**Heredity of the induction**

Let us now prove that, if for a given $l < n$, we have

$$\iota = \frac{(-1)^l(l!)^2(-i)^D}{D!(2l)!}\left(\text{Tr}\Big(\hat{C}\,\hat{\alpha}^{2D-2l}\,\hat{H}_F[\hat{\alpha}, \hat{H}_F]^{2l-1}[\hat{\alpha}, \hat{\theta}_\Gamma]\Big) + \frac{1}{2}\,\text{Tr}\Big(\hat{C}\,\hat{\alpha}^{2D-2l}\,\hat{H}_F[\hat{\alpha}, \hat{H}_F]^{2l}[\hat{\theta}_\Gamma, \hat{H}_F]\Big)\right) \tag{145}$$

– that we shall simply write as

$$\iota = f_D(l)(T_1 + T_2) \tag{146}$$

for short – where $T_1$ and $T_2$ refer to the two trace terms in (145) and $f_D(l)$ is the prefactor – then this expression is also true for $l \to l+1$. For that purpose, we take advantage of the same few technical tricks than those previously used: inserting freely $\alpha^2$ and $\hat{H}_F^2$ in the trace expressions since a commutator with $\hat{\theta}_\Gamma$ fixes the value of the invariant on the shell, performing trace cyclicities (remembering that those implying $\hat{\alpha}$ pick a minus sign), and using various anti-commutation relations, such as that of $\hat{H}_F$ with the chiral operator $\hat{C}$, and others implying $\hat{\alpha}$, $\hat{H}_F$ and $\hat{\theta}_\Gamma$, namely (131), (135) and (142). In particular, (131) and (135) can more generally (and more usefully for this section) be written as

$$[\alpha, \hat{H}_F]^p\hat{H}_F = (-1)^p\hat{H}_F[\alpha, \hat{H}_F]^p \tag{147}$$

$$[\alpha, \hat{H}_F]^p\,\hat{\alpha} = (-1)^p\,\hat{\alpha}[\alpha, \hat{H}_F]^p \tag{148}$$

$$[\hat{\theta}_\Gamma, \hat{H}_F]^p\hat{H}_F = (-1)^p\hat{H}_F[\hat{\theta}_\Gamma, \hat{H}_F]^p \tag{149}$$

for $p$ an integer and under the usual assumptions related to the shell.

Let us now modify the expression (145)-(146) by first rearranging the terms in the first trace, $T_1$. By extracting $\alpha$ from $\alpha^{2D-2l}$, we trivially get a term $\hat{\alpha}\,\hat{H}_F$. This is somehow half the commutator $[\hat{\alpha}, \hat{H}_F]$ which we would like to introduce. To get the second half, $\hat{H}_F\alpha$, we permute $\hat{\alpha}$ cyclically to the left. Since these two terms are equal under the trace, $T_1$ becomes

$$T_1 \equiv \mathrm{Tr}\Big(\hat{C}\,\hat{\alpha}^{2D-2l}\,\hat{H}_F[\hat{\alpha}, \hat{H}_F]^{2l-1}[\hat{\alpha}, \hat{\theta}_\Gamma]\Big)$$
$$= \frac{1}{2}\,\mathrm{Tr}\Big(\hat{C}\,\hat{\alpha}^{2D-2l-1}[\hat{\alpha}, \hat{H}_F]^{2l}[\hat{\alpha}, \hat{\theta}_\Gamma]\Big)\ . \tag{150}$$

To get the correct term $\hat{\alpha}^{2D-2l-2}\,\hat{H}_F$ and another commutator $[\hat{\alpha}, \hat{H}_F]$, we can extract again $\hat{\alpha}$ out of $\hat{\alpha}^{2D-2l-1}$, and insert $\hat{H}_F^2$, so that, from the previous equation,

$$T_1 = \frac{1}{2}\,\mathrm{Tr}\Big(\hat{C}\,\hat{\alpha}^{2D-2l-2}\,\hat{H}_F\hat{H}_F\,\hat{\alpha}[\hat{\alpha}, \hat{H}_F]^{2l}[\hat{\alpha}, \hat{\theta}_\Gamma]\Big) \tag{151}$$

where we permute cyclically one of the two $\hat{H}_F$'s to the left to get

$$T_1 = \frac{1}{4}\,\mathrm{Tr}\Big(\hat{C}\,\hat{\alpha}^{2D-2l-2}\,\hat{H}_F(\hat{H}_F\,\hat{\alpha}[\hat{\alpha}, \hat{H}_F]^{2l}[\hat{\alpha}, \hat{\theta}_\Gamma] - \hat{\alpha}[\hat{\alpha}, \hat{H}_F]^{2l}[\hat{\alpha}, \hat{\theta}_\Gamma]\hat{H}_F)\Big)$$
$$= -\frac{1}{4}\,\mathrm{Tr}\Big(\hat{C}\,\hat{\alpha}^{2D-2l-2}\,\hat{H}_F\,[\hat{\alpha}[\hat{\alpha}, \hat{H}_F]^{2l}[\hat{\alpha}, \hat{\theta}_\Gamma], \hat{H}_F]\Big) \tag{152}$$

where a commutator of the form $[A_1A_2A_3, B]$ has appeared. This commutator can be expanded as

$$[A_1A_2A_3, B] = [A_1, B]A_2A_3 + A_1[A_2, B]A_3 + A_1A_2[A_3, B] \tag{153}$$

which leads to

$$[\hat{\alpha}[\hat{\alpha}, \hat{H}_F]^{2l}[\hat{\alpha}, \hat{\theta}_\Gamma], \hat{H}_F] = [\hat{\alpha}, \hat{H}_F]^{2l+1}[\hat{\alpha}, \hat{\theta}_\Gamma] + \hat{\alpha}\underbrace{[[\hat{\alpha}, \hat{H}_F]^{2l}, \hat{H}_F]}_{=0}[\hat{\alpha}, \hat{\theta}_\Gamma] + \hat{\alpha}[\hat{\alpha}, \hat{H}_F]^{2l}[[\hat{\alpha}, \hat{\theta}_\Gamma], \hat{H}_F]$$
$$\tag{154}$$

where the term vanishes because of (147), and we get

$$T_1 = -\frac{1}{4}\,\mathrm{Tr}\Big(\hat{C}\,\hat{\alpha}^{2D-2l-2}\,\hat{H}_F[\hat{\alpha}, \hat{H}_F]^{2l+1}[\hat{\alpha}, \hat{\theta}_\Gamma]\Big) - \frac{1}{4}\,\mathrm{Tr}\Big(\hat{C}\,\hat{\alpha}^{2D-2l-2}\,\hat{H}_F\,\hat{\alpha}[\hat{\alpha}, \hat{H}_F]^{2l}[[\hat{\alpha}, \hat{\theta}_\Gamma], \hat{H}_F]\Big)$$
$$\equiv U_1 + U_2\ . \tag{155}$$

The first term of this expression, $U_1$, is the desired one for the heredity (up to prefactors). We thus need to massage the second one that we call $U_2$. Let us first cyclically permute $\hat{H}_F$ to the left in $U_1$. By using its anti-commutation with $\hat{C}$ and with $[[\hat{\alpha}, \hat{\theta}_\Gamma], \hat{H}_F]$ (which follows again from $\hat{H}_F^2 \approx 1$), this term reads

$$U_2 = -\frac{1}{4}\,\mathrm{Tr}\Big(\hat{C}\,\hat{\alpha}^{2D-2l-1}\,\hat{H}_F[\hat{\alpha}, \hat{H}_F]^{2l}[[\hat{\alpha}, \hat{\theta}_\Gamma], \hat{H}_F]\Big)\ . \tag{156}$$

Next, we use the identity $[[A, B], C] = [[A, C], B] + [A, [B, C]]$ with $A = \hat{\alpha}$, $B = \hat{\theta}_\Gamma$ and $C = \hat{H}_F$, as

$$[[\hat{\alpha}, \hat{\theta}_\Gamma], \hat{H}_F] = [[\hat{\alpha}, \hat{H}_F], \hat{\theta}_\Gamma] + [\hat{\alpha}, [\hat{\theta}_\Gamma, \hat{H}_F]] \tag{157}$$

leaving us with

$$U_2 = -\frac{1}{4}\operatorname{Tr}\left(\hat{C}\,\hat{\alpha}^{2D-2l-1}\,\hat{H}_F[\hat{\alpha},\hat{H}_F]^{2l}[[\hat{\alpha},\hat{H}_F],\hat{\theta}_\Gamma]\right) - \frac{1}{4}\operatorname{Tr}\left(\hat{C}\,\hat{\alpha}^{2D-2l-1}\,\hat{H}_F[\hat{\alpha},\hat{H}_F]^{2l}[\hat{\alpha},[\hat{\theta}_\Gamma,\hat{H}_F]]\right)$$
$$\equiv V_1 + V_2\ . \tag{158}$$

The second term, $V_2$, can be split into two terms: one of them compensating $T_2$ exactly, which simplifies the complete expression of $\iota$, the other one contributing to the heredity on $T_2$. Indeed,

$$V_2 = -\frac{1}{4}\operatorname{Tr}\left(\hat{C}\,\hat{\alpha}^{2D-2l-1}\,\hat{H}_F[\hat{\alpha},\hat{H}_F]^{2l}(\hat{\alpha}[\hat{\theta}_\Gamma,\hat{H}_F] - [\hat{\theta}_\Gamma,\hat{H}_F]\,\hat{\alpha})\right) \tag{159}$$

using cyclicity of the trace and the usual anti-commutation relation, one can get that

$$V_2 = -\frac{1}{4}\operatorname{Tr}\left(\hat{C}\,\hat{\alpha}^{2D-2l-2}(\hat{\alpha}\,\hat{H}_F\,\hat{\alpha} + \hat{\alpha}^2\,\hat{H}_F)[\hat{\alpha},\hat{H}_F]^{2l}[\hat{\theta}_\Gamma,\hat{H}_F]\right)$$
$$= -\frac{1}{4}\operatorname{Tr}\left(\hat{C}\,\hat{\alpha}^{2D-2l-2}(2\,\hat{\alpha}^2\,\hat{H}_F - \hat{\alpha}[\hat{\alpha},\hat{H}_F])[\hat{\alpha},\hat{H}_F]^{2l}[\hat{\theta}_\Gamma,\hat{H}_F]\right) \tag{160}$$
$$= \underbrace{-\frac{1}{2}\operatorname{Tr}\left(\hat{C}\,\hat{\alpha}^{2D-2l}\,\hat{H}_F[\hat{\alpha},\hat{H}_F]^{2l}[\hat{\theta}_\Gamma,\hat{H}_F]\right)}_{-T_2} + \frac{1}{4}\operatorname{Tr}\left(\hat{C}\,\hat{\alpha}^{2D-2l-1}[\hat{\alpha},\hat{H}_F]^{2l+1}[\hat{\theta}_\Gamma,\hat{H}_F]\right)\ .$$

Therefore, to sum up,

$$T_1 = -\frac{1}{4}\operatorname{Tr}\left(\hat{C}\,\hat{\alpha}^{2D-2l-2}\,\hat{H}_F[\hat{\alpha},\hat{H}_F]^{2l+1}[\hat{\alpha},\hat{\theta}_\Gamma]\right) + \frac{1}{4}\operatorname{Tr}\left(\hat{C}\,\hat{\alpha}^{2D-2l-1}[\hat{\alpha},\hat{H}_F]^{2l+1}[\hat{\theta}_\Gamma,\hat{H}_F]\right) + V_1 - T_2 \tag{161}$$

so that, at this stage, the form expression of the index reads

$$\iota = \frac{(-1)^l(l!)^2(-i)^D}{D!(2l)!}\left(-\frac{1}{4}\operatorname{Tr}\left(\hat{C}\,\hat{\alpha}^{2D-2l-2}\,\hat{H}_F[\hat{\alpha},\hat{H}_F]^{2l+1}[\hat{\alpha},\hat{\theta}_\Gamma]\right)\right.$$
$$\left. + \frac{1}{4}\operatorname{Tr}\left(\hat{C}\,\hat{\alpha}^{2D-2l-1}[\hat{\alpha},\hat{H}_F]^{2l+1}[\hat{\theta}_\Gamma,\hat{H}_F]\right) + V_1\right) \tag{162}$$

where the term $T_2$ has been cancelled out and where

$$V_1 = -\frac{1}{4}\operatorname{Tr}\left(\hat{C}\,\hat{\alpha}^{2D-2l-1}\,\hat{H}_F[\hat{\alpha},\hat{H}_F]^{2l}[[\hat{\alpha},\hat{H}_F],\hat{\theta}_\Gamma]\right) \tag{163}$$

must still be modified. We will see below that $V_1$ can be written in two terms proportional to those already present in (162). To do so, we use the generalization of the expansion (153)

$$[A^{2l+1},B] = \sum_{j=0}^{2l} A^j[A,B]A^{2l-j} \tag{164}$$

for $A = [\hat{\alpha},\hat{H}_F]$ and $B = \hat{\theta}_\Gamma$, to deduce, using the usual cyclicity and anti-commutation relation

$$\operatorname{Tr}\left(\hat{C}\,\hat{\alpha}^{2D-2l-1}\,\hat{H}_F[A^{2l+1},B]\right) = \operatorname{Tr}\left(\hat{C}\,\hat{\alpha}^{2D-2l-1}\,\hat{H}_F\sum_{j=0}^{2l} A^j[A,B]A^{2l-j}\right)$$
$$= \operatorname{Tr}\left(\hat{C}\,\hat{\alpha}^{2D-2l-1}\,\hat{H}_F\sum_{j=0}^{2l} A^{2l}[A,B]\right)$$
$$= (2l+1)\operatorname{Tr}\left(\hat{C}\,\hat{\alpha}^{2D-2l-1}\,\hat{H}_F A^{2l}[A,B]\right) \tag{165}$$

from which $V_1$ becomes

$$V_1 = -\frac{1}{4(2l+1)} \operatorname{Tr}\left(\hat{C}\,\hat{\alpha}^{2D-2l-1}\,\hat{H}_F[[\hat{\alpha},\hat{H}_F]^{2l+1},\hat{\theta}_\Gamma]\right) . \tag{166}$$

By permuting $\hat{\theta}_\Gamma$ to the right in the first term of this commutator, we can write

$$V_1 = -\frac{1}{4(2l+1)} \operatorname{Tr}\left(\hat{C}\,\hat{\alpha}^{2D-2l-2}(\theta\,\hat{\alpha}\,\hat{H}_F - \hat{\alpha}\,\hat{H}_F\,\hat{\theta}_\Gamma)[\hat{\alpha},\hat{H}_F]^{2l+1}\right) \tag{167}$$

such that the new commutator $[\hat{\alpha}\,\hat{H}_F,\hat{\theta}_\Gamma]$ can be decomposed following (153) such that

$$V_1 = -\frac{1}{4(2l+1)} \operatorname{Tr}\left(\hat{C}\,\hat{\alpha}^{2D-2l-2}(-[\hat{\alpha},\hat{\theta}_\Gamma]\hat{H}_F - \hat{\alpha}[\hat{H}_F,\hat{\theta}_\Gamma])[\hat{\alpha},\hat{H}_F]^{2l+1}\right) . \tag{168}$$

Finally, by permuting cyclically $[\hat{\alpha},\hat{\theta}_\Gamma]$ and $\hat{\alpha}[\hat{H}_F,\hat{\theta}_\Gamma]$ to the left, one finds

$$V_1 = -\frac{1}{4(2l+1)} \left(\operatorname{Tr}\left(\hat{C}\,\hat{\alpha}^{2D-2l-2}\,\hat{H}_F[\hat{\alpha},\hat{H}_F]^{2l+1}[\hat{\alpha},\hat{\theta}_\Gamma]\right) - \operatorname{Tr}\left(\hat{C}\,\hat{\alpha}^{2D-2l-1}[\hat{\alpha},\hat{H}_F]^{2l+1}[\hat{\theta}_\Gamma,\hat{H}_F]\right)\right) \tag{169}$$

which is indeed composed of the first two terms of (162) up to a prefactor, and we get

$$\iota = -\frac{2l+2}{4(2l+1)} f(l) \left(\operatorname{Tr}\left(\hat{C}\,\hat{\alpha}^{2D-2l-2}\,\hat{H}_F[\hat{\alpha},\hat{H}_F]^{2l+1}[\hat{\alpha},\hat{\theta}_\Gamma]\right)\right.$$
$$\left. - \operatorname{Tr}\left(\hat{C}\,\hat{\alpha}^{2D-2l-1}[\hat{\alpha},\hat{H}_F]^{2l+1}[\hat{\theta}_\Gamma,\hat{H}_F]\right)\right) \tag{170}$$

where the prefactor becomes

$$-\frac{2l+2}{4(2l+1)}\frac{(-1)^l(l!)^2(-i)^D}{D!(2l)!} = \frac{(-1)^{l+1}((l+1)!)^2(-i)^D}{D!(2(l+1))!} = f_D(l+1) \tag{171}$$

which is the desired one for the heredity. Thus at this stage, (170) shows the correct prefactor and the correct expression of first trace for the heredity. We still need to modify the second trace to complete the proof, which we do by adding $H_F^2 \approx 1$, and then circulating cyclically $\hat{H}_F$ to the left such that

$$\operatorname{Tr}\left(\hat{C}\,\hat{\alpha}^{2D-2l-2}\,\hat{H}_F^2\,\hat{\alpha}[\hat{\alpha},\hat{H}_F]^{2l+1}[\hat{\theta}_\Gamma,\hat{H}_F]\right) = -\operatorname{Tr}\left(\hat{C}\,\hat{\alpha}^{2D-2l-2}\,\hat{H}_F\,\hat{\alpha}\,\hat{H}_F[\hat{\alpha},\hat{H}_F]^{2l+1}[\hat{\theta}_\Gamma,\hat{H}_F]\right) \tag{172}$$

which allows us to deduce

$$\operatorname{Tr}\left(\hat{C}\,\hat{\alpha}^{2D-2l-1}[\hat{\alpha},\hat{H}_F]^{2l+1}[\hat{\theta}_\Gamma,\hat{H}_F]\right) = -\frac{1}{2}\operatorname{Tr}\left(\hat{C}\,\hat{\alpha}^{2D-2l-2}\,\hat{H}_F[\hat{\alpha},\hat{H}_F]^{2l+2}[\hat{\theta}_\Gamma,\hat{H}_F]\right) \tag{173}$$

that we substitute into (170)

$$\iota = \frac{(-1)^{l+1}((l+1)!)^2(-i)^D}{D!(2(l+1))!} \left(\operatorname{Tr}\left(\hat{C}\,\hat{\alpha}^{2D-2l-2}\,\hat{H}_F[\hat{\alpha},\hat{H}_F]^{2l+1}[\hat{\alpha},\hat{\theta}_\Gamma]\right)\right.$$
$$\left. +\frac{1}{2}\operatorname{Tr}\left(\hat{C}\,\hat{\alpha}^{2D-2l-2}\,\hat{H}_F[\hat{\alpha},\hat{H}_F]^{2l+2}[\hat{\theta}_\Gamma,\hat{H}_F]\right)\right) \tag{174}$$

to complete the proof of the heredity.

Now that we have demonstrated the formula (114), we can substitute $l = D$ to get the final expression

$$\iota = i^D \frac{D!}{(2D)!} \left( \text{Tr}\left(\hat{C}\hat{H}_F[\hat{\alpha}, \hat{H}_F]^{2D-1}[\hat{\alpha}, \hat{\theta}_\Gamma]\right) + \frac{1}{2} \text{Tr}\left(\hat{C}\hat{H}_F[\hat{\alpha}, \hat{H}_F]^{2D}[\hat{\theta}_\Gamma, \hat{H}_F]\right) \right) . \tag{175}$$

This is the form expression $\iota$ of the shell index $\mathcal{I}$, that we defined as $\iota = \mathcal{I}\omega^D$ with $\omega$ the symplectic form. We would like to extract $\mathcal{I}$ from (175) which does not seem to be straightforward since the volume element $\omega^D$ is not apparent in this expression. Still, we can use the anti-symmetric decomposition of the product of the operator-valued form $\hat{\alpha}$ with usual operators $\hat{A}_l$ with $l = \{1, \ldots, 2D\}$

$$\prod_{l=1}^{2D}(\hat{\alpha}\,\hat{A}_l) = \sum_{j_1 \ldots j_{2D}=1}^{2D} \varepsilon_{j_1, \ldots, j_{2D}} \prod_{l=1}^{2D} \hat{a}_{j_l} \hat{A}_l (-\omega)^D \tag{176}$$

with $\varepsilon_{j_1, \ldots, j_{2D}}$ the antisymmetric Levi-Civita tensor, and $\hat{a}_j$ the components of $\hat{\alpha}$ as defined in (107), which yields

$$\mathcal{I} = (-i)^D \frac{D!}{(2D)!} \left( \text{Tr}\left( \sum_{j_1 \ldots j_{2D}=1}^{2D} \varepsilon_{j_1, \ldots, j_{2D}} \hat{C}\hat{H}_F \prod_{l=1}^{2D-1} [\hat{a}_{j_l}, \hat{H}_F][\hat{a}_{2D}, \hat{\theta}_\Gamma] \right) \right.$$
$$\left. + \frac{1}{2} \text{Tr}\left( \sum_{j_1 \ldots j_{2D}=1}^{2D} \varepsilon_{j_1, \ldots, j_{2D}} \hat{C}\hat{H}_F \prod_{l=1}^{2D} [\hat{a}_{j_l}, \hat{H}_F][\hat{\theta}_\Gamma, \hat{H}_F] \right) \right) . \tag{177}$$

### Semi-classical expression of the shell index

We now derive the semi-classical expression of the shell index, by first applying the trace formula (97) that converts a trace of product of operators into an integral over phase space of their Wigner-Weyl symbols, and then taking the limit $\Gamma \to \infty$. The semi-classical limit consists in keeping the higher order terms in $1/\Gamma$. In that limit, commutators of operators are replaced by Poisson brackets of their symbols as $[\hat{a}_{j_l}, \hat{H}_F] \to i\{a_{j_l}, H_F\}$, with $\hat{x}_j \to x_j$, $\partial_{x_j} \to ik_j$, $\hat{T}_j \to e^{-ik_j}$ and $\hat{T}_j^\dagger n_j \to e^{ik_j} n_j + e^{ik_j}/2$, and the second trace in (177) can be neglected, which yields

$$\mathcal{I} = \frac{D!}{(2D)!} \left(\frac{i}{2\pi}\right)^D \sum_{j_1 \ldots j_{2D}=1}^{2D} \varepsilon_{j_1, \ldots, j_{2D}} \text{Tr}^{\text{int}}\left(\int \left(\prod_{l=1}^{2D-1} dk_l dx_l\right) dk_{2D} dx_{2D} C H_F \{a_{j_l}, H_F\}\{a_{2D}, \theta_\Gamma\}\right) + O(1/\Gamma) \tag{178}$$

where the integrals run over $\mathbb{R}$ as we have assumed continuous coordinates, but should be understood as $\sum_{n_l \in \mathbb{Z}} \int_0^{2\pi} dk_l$ for lattices. This expression can equivalently be written in a more compact way with $\alpha$, the symbol of $\hat{\alpha}$, which is the differential one-form

$$\alpha = \sum_{j}^{D} a_{2j} dx^j + a_{2j+1} dk^j \tag{179}$$

such that

$$\mathcal{I} = \frac{D!}{(2D)!}\left(\frac{i}{2\pi}\right)^D \int \mathrm{Tr}^{\mathrm{int}}(CH_F\{\alpha, H_F\}^{2D-1}\{\alpha, \theta_\Gamma\}) + O(1/\Gamma) \ . \tag{180}$$

The Poisson brackets implying the components of $\alpha$ are given by

$$\{a_{2j}, A\} = \{x_j, A\} = \partial_{k_j} A \tag{181}$$

$$\{a_{2j+1}, A\} = \{i^2 k_j, A\} = \partial_{x_j} A \tag{182}$$

in the continuous case, and by

$$\{a_{2j}, A\} = \{e^{ik_j} n_j + e^{ik_j}/2, A\} = e^{ik_j}\partial_{k_j} A - i(n_j + 1/2)e^{ik_j} i\delta_{n_j} A \tag{183}$$

$$\{a_{2j+1}, A\} = \{-ie^{-ik_j}, A\} = e^{-ik_j}\delta_{n_j} A \tag{184}$$

in the discrete one, where the expressions look more involved. However we have an anti-symmetrisation of the coefficients in $\{a_j, \}$, therefore since we already have a derivative in $\delta_{n_j}$ due to the $\{T_j, A\}$ terms, the coefficient in $\delta_{n_j}$ in the Poisson bracket $\{T_j^\dagger n_j, A\}$ are cancelled (note that $\theta_\Gamma$ is just a scalar and thus commutes with everything) and we only keep the coefficient in $\partial_{k_j} A$. Therefore, we end up with the simplified equation

$$\mathcal{I} = \frac{(D)!}{(2D)!(-2i\pi)^D}\int \mathrm{Tr}^{\mathrm{int}}(CH_F(dH_F)^{2D-1}(d\theta_\Gamma)) + O(1/\Gamma) \tag{185}$$

where $dA = \sum_j \partial_{x_j}A dx_i + \partial_{k_j}A dk_j$ ($\partial_{x_j}$ should be replaced by the discrete derivative in the lattice case) is the one-form differential of the symbol $A$ in phase space.

Actually, now that we work with symbols in phase space, we can show that the precise shape of the cut-off symbol $\theta_\Gamma$ does not matter as long as it decays far away from the gapless region where the zero mode seats. Indeed, for any phase space cut-off $\theta_\Gamma'$ such that $\theta_\Gamma' - \theta_\Gamma$ is non-zero only in the gapped region of phase space, we have

$$\begin{aligned}
\int \mathrm{Tr}^{\mathrm{int}}(CH_F(dH_F)^{2D-1}(d\theta_\Gamma - d\theta_\Gamma')) &= -\int \mathrm{Tr}^{\mathrm{int}}(C(dH_F)^{2D}(\theta_\Gamma - \theta_\Gamma')) \\
&= -\int \mathrm{Tr}^{\mathrm{int}}(CH_F^2(dH_F)^{2D}(\theta_\Gamma - \theta_\Gamma')) \\
&= \frac{1}{2}\int \mathrm{Tr}^{\mathrm{int}}(CH_F[(dH_F)^{2D}(\theta_\Gamma - \theta_\Gamma'), H_F]) \\
&= 0
\end{aligned} \tag{186}$$

where we have used at the end that $H_F$ anticommutes with $dH_F$ and commutes with the scalars $\theta_\Gamma - \theta_\Gamma'$. As a consequence, one can replace $\theta_\Gamma$ by the step-function $\theta_\Gamma = \mathbb{1}_B$ where $B$ is any volume which contains the gapless region, so that the shell index in the semi-classical limit becomes

$$\mathcal{I}_{\mathrm{shell}} = \frac{D!}{(2D)!(-2i\pi)^D}\int_{\mathrm{shell}}\mathrm{Tr}^{\mathrm{int}}(CH_F(dH_F)^{2D-1}) \tag{187}$$

where the shell is the boundary $\partial B$ of $B$ which encloses the gapless region. Finally, this expression can be expressed in terms of $U$, the off-diagonal component of $H_F = \begin{pmatrix} 0 & U^\dagger \\ U & 0 \end{pmatrix}$, as

$$\mathcal{I}_{\mathrm{shell}} = \frac{-2(D)!}{(2D)!(2i\pi)^D}\int_{\mathrm{shell}}\mathrm{Tr}^{\mathrm{int}}(U^\dagger dU)^{2D-1} \tag{188}$$

which is the formula (14) announced in the main text.

# G Higher order insulators with hard boundary: Partial semi-classical limit and numerical programs

In this appendix we want to explain how we can partially simplify the computation of shell index (56) by doing a partial semi-classical expansion of the invariant and obtain the expression (79). First let us recall the general expression (56) we obtained for the shell index in the $D$-dimensional case without semi-classical hypothesis. This expression

$$\mathcal{I}_{\text{shell}} \underset{\text{lim}}{=} \frac{-1}{12} \left( \text{Tr}\left(\hat{C}\hat{H}_F[\hat{\alpha},\hat{H}_F]^3[\hat{\alpha},\hat{\theta}_\Gamma]\right) + \frac{1}{2}\text{Tr}\left(\hat{C}\hat{H}_F[\hat{\alpha},\hat{H}_F]^4[\hat{\theta}_\Gamma,\hat{H}_F]\right) \right) \tag{189}$$

where $\hat{\alpha} = T_x^\dagger n_x dx + -iT_x dk_x + T_y^\dagger n_y dy + -iT_y dk_y$ is a one form, and the whole expression is therefore an anti-symmetrised sum of all types of commutators.

Let's now try to simplify it assuming that the system is invariant by translation in both direction in the bulk and invariant in the tangent direction near the edges far away from the corners. In the above expression, one can first notice that the term $\text{Tr}\left(\hat{C}\hat{H}_F[d,\hat{H}_F]^4[\hat{\theta}_\Gamma,\hat{H}_F]\right)$ is always a product of three commutators $[T_x,\hat{H}_F]$ (non zero only near a vertical edge), $[T_y,\hat{H}_F]$ (non zero only near an horizontal edge) and $[\hat{\theta}_F,\hat{H}_F]$ (non zero only near the shell). This three terms are non zero in incompatible regions and the couplings are local, therefore this term decats to zero in the limit $\Gamma \to +\infty$ and we have that

$$\mathcal{I}_{\text{shell}} \underset{\text{lim}}{=} \frac{-1}{12}\text{Tr}\left(\hat{C}\hat{H}_F[\hat{\alpha},\hat{H}_F]^3[\hat{\alpha},\hat{\theta}_\Gamma]\right) \tag{190}$$

We can then simplify again the expression by noting that a commutator of the form $[T_x,\hat{H}_F]$ or $[T_y,\hat{H}_F]$ must always be present in the expression of $\mathcal{I}_{\text{shell}}$ which imply that the contribution to the expression are localised in the regions which are on the shell and near an edge (see figure 19). We can then simplifies partially the expression by using the invariance of translation of $\hat{H}_F$ in the direction tangent to each edge. Therefore the index can be written as the sum of two contributions $\mathcal{I}_{\text{shell}} = \mathcal{I}_{\text{edge},x} + \mathcal{I}_{\text{edge},y}$ with each contribution coming from a different edge where we are able use the Fourier transform in the tangent direction. For example we have that

$$\mathcal{I}_{\text{edge},x} = \frac{-1}{24\pi}\int_0^{2\pi} dk_x \tilde{\text{Tr}}\left(\tilde{C}\tilde{H}_F[\tilde{\alpha},\tilde{H}_F]^3\right) \tag{191}$$

where the $\sim$ notation denotes here the fact that we do the Wigner-Weyl transform only in one direction. $\tilde{d}$ can for example be written as $\tilde{\alpha} = \partial_x dk_x + T_y^\dagger n_y dy + T_y dk_y$ now. This is indeed the expression (79) claimed in the main text.

We then present the numerical programs which compute the invariant. The first one using the expression of the mode index (8), the second one using the semi-classical limit of the shell index.

**Computation of the index of the model (77) using the formula (8) of the mode index**

```
from scipy import linalg
import numpy as np
#Function computing the operator func(H) for a given operator H
def funm_herm(H, func):
    w, v = linalg.eigh(H)
    w = func(w)
```

```
    return (v * w).dot(v.T.conj())

#function which is 0 for negative number and 1 for positive number
def step(x):
    return (np.sign(x)+1)/2

#the tanh serve as the smooth transition function in energy
def smoothstep(x):
    k=20
    return np.tanh(k*x)

#Lenght in site of the lattice
L = 13
#Coupling constants, topological when |t1|>|t|
t=0.6
t1 = 1

#C is the chiral operator, theta the cut-off operator,
#Id the identity and H the Hamiltonian
Id = np.eye(L*L*4)
H = np.zeros((L*L*4,L*L*4))
C = np.zeros((L*L*4,L*L*4))
theta = np.zeros((L*L*4,L*L*4))

#The degree of freedom which is situated on the i-th site in the x direction,
#the j-th site in the y direction
#and which correspond to the k-th degrees of freedom of tha site
#is encoded by the single number i+j*L+k*L**2
for i in range(L):
    for j in range(L):
        H[i+j*L+0*L**2,i+j*L+1*L**2]=t
        H[i+j*L+0*L**2,i+j*L+2*L**2]=t
        H[i+j*L+1*L**2,i+j*L+0*L**2]=t
        H[i+j*L+1*L**2,i+j*L+3*L**2]=t

        H[i+j*L+2*L**2,i+j*L+0*L**2]=t
        H[i+j*L+2*L**2,i+j*L+3*L**2]=-t
        H[i+j*L+3*L**2,i+j*L+1*L**2]=t
        H[i+j*L+3*L**2,i+j*L+2*L**2]=-t

        C[i+j*L+0*L**2,i+j*L+0*L**2]=1
        C[i+j*L+1*L**2,i+j*L+1*L**2]=-1
        C[i+j*L+2*L**2,i+j*L+2*L**2]=-1
        C[i+j*L+3*L**2,i+j*L+3*L**2]=1

        for k in range(4):
            theta[i+j*L+k*L**2,i+j*L+k*L**2] = step(i-L//2)* step(j-L//2)
        if i < L-1:
            H[(i+1)+j*L+0*L**2,i+j*L+2*L**2]=t1
            H[(i+1)+j*L+1*L**2,i+j*L+3*L**2]=t1
            H[i+j*L+2*L**2,(i+1)+j*L+0*L**2]=t1
            H[i+j*L+3*L**2,(i+1)+j*L+1*L**2]=t1
        if j < L-1:
            H[i+(j+1)*L+0*L**2,i+j*L+1*L**2]=t1
            H[i+j*L+1*L**2,i+(j+1)*L+0*L**2]=t1
            H[i+(j+1)*L+2*L**2,i+j*L+3*L**2]=-t1
```

```
            H[i+j*L+3*L**2,i+(j+1)*L+2*L**2]=-t1

#computation of the flatten Hamiltonian H_F
HF = funm_herm(H, smoothstep)

#computation of the index
I = np.trace(C.dot(Id-HF.dot(HF)).dot(theta)).real
```

**Computation of the index of the model (77) using the partial semi-classical limit (79) of the shell index**

```
from scipy import linalg
import numpy as np
from math import pi
#Function computing the operator func(H) for a given operator H
def funm_herm(H, func):
    w, v = linalg.eigh(H)
    w = func(w)
    return (v * w).dot(v.T.conj())

#function which is 0 for negative number and 1 for positive number
def step(x):
    return (np.sign(x)+1)/2

#the tanh serve as the smooth transition function in energy
def smoothstep(x):
    k=20
    return np.tanh(k*x)

#Commutator of two matrices AB-BA
def com(A,B):
    return A.dot(B)-B.dot(A)

#Sum of all antisimetrised combination of A,B,C
#which are ABC+BCA+CAB-ACB-CBA-BAC
def tricom(A,B,C):
    return A.dot(com(B,C))+B.dot(com(C,A))+C.dot(com(A,B))

#computation of the flatten Hamiltonian for a given transverse momentum k
#The degree of freedom which is situated on the i-th site in the normal direction,
#and which correspond to the k-th degrees of freedom of tha site
#is encoded by the single number i+k*L
def HF(k):
    H = np.zeros((L*4,L*4), dtype= np.complex128)
    for i in range(L):
        H[i+0*L,i+1*L]=t+t1*np.exp(-1j*k)
        H[i+0*L,i+2*L]=t
        H[i+1*L,i+0*L]=t+t1*np.exp(1j*k)
        H[i+1*L,i+3*L]=t

        H[i+2*L,i+0*L]=t
        H[i+2*L,i+3*L]=-(t+t1*np.exp(-1j*k))
        H[i+3*L,i+1*L]=t
        H[i+3*L,i+2*L]=-(t+t1*np.exp(1j*k))
```

```python
        if i < L-1:
            H[(i+1)+0*L,i+2*L]=t1
            H[(i+1)+1*L,i+3*L]=t1
            H[i+2*L,(i+1)+0*L]=t1
            H[i+3*L,(i+1)+1*L]=t1
    return funm_herm(H, smoothstep)

#Lenght in site of the lattice
L = 13
#Coupling constants, topological when |t1|>|t|
t=0.6
t1 = 1

#C is the chiral operator, theta the cut-off operator,
#Id the identity, H the Hamiltonian
#ni is the diagonal position operator equal to i on the i-th site
#T is the tranlation operator of one unit cell in the normal direction
H = np.zeros((L*4,L*4))
C = np.zeros((L*4,L*4))
theta = np.zeros((L*4,L*4))
T = np.zeros((L*4,L*4))
ni = np.zeros((L*4,L*4))

for i in range(L):

        C[i+0*L,i+0*L]=1
        C[i+1*L,i+1*L]=-1
        C[i+2*L,i+2*L]=-1
        C[i+3*L,i+3*L]=1

        for k in range(4):
            theta[i+k*L,i+k*L] = step(L//2-i)
            ni[i+k*L,i+k*L] = i
        if i < L-1:
            for k in range(4):
                T[i+1+k*L,i+k*L] = 1
#operator T^\dagger*n_i
T1 = T.T.dot(ni)

#number of point in the discretisation of the tranverse wavenumber
n=50

listek = np.linspace(0,2*pi,n+1)
ListeI = np.zeros((n), dtype= np.complex128)

HF1 = HF(listek[0])
for i in range(n):
    HF0 = HF1
    HF1 = HF(listek[i+1])
    #discretisation of the discrete derivative in k
    dHF1 = (HF1-HF0)*n
    dHF2 = com(T1,HF0)
    dHF3 = com(T,HF0)
    #Computation of the density to integrate
    Z = C.dot(HF0).dot(tricom(dHF1,dHF2,dHF3)).dot(theta)
```

```
    #Integration in the transverse wavenumber k
    ListeI[i] = np.trace(Z*1j).real/n
I = -np.sum(ListeI)/(2*pi)/6
```