# Peer review of "Mode-Shell correspondence, a unifying theory in topological physics -- Part I: Chiral number of zero-modes"

_SciPost Physics_

## Round 1 · Referee Report · Anonymous (Referee 1) · 2023-12-3

Strengths

  • novel expression for the topological index of a chiral system in terms of the mode-shell correspondence
  • flexible expression treating momentum and position on equal footing
  • results applicable both for analytical and numerical work

Weaknesses

  • clarity of the paper, in particular Sec. 4 can be improved
  • appendix E is cluttered and not digestible
  • it feels that exceptions, subtleties, relations to prior mathematical work is given too much space when compared to the generic case that is of interest to the generic physicist

Report

In their manuscript titled 'Mode-Shell correspondence, a unifying theory in topological physics', the authors present a novel way, dubbed 'mode-shell correspondence', to calculate the index of a chiral Hamiltonian. The index of a Hamiltonian is of importance in physics, as it is related to the number of modes that are pinned at zero energy due to topology. It is also referred to as a topological quantum number. The authors provide a method to determine the index locally in phase space, by integrating over the boundary (shell) surrounding this point. In this respect, the results are similar to the Gauss's law. The manuscript, in large parts, relies on a semiclassical description of the problem treating position and momentum on an equal footing, making all of phase-space accessible. The present manuscript concentrates on chiral symmetric model. In a future work, the extension to other symmetries is planned.

In general, I do believe that the results are valid and interesting. They help to provide a unified picture of many different effects, weak-topology, higher-order topology, states trapped in defects, ..., that relate to topological obstruction. Even though it is clear that the authors have tried to make their results accessible, the later chapters, especially chapter 3.2 and chapter 4, are hard to digest. Even though the main ideas presented in chapter 2 are rather simple, in concrete applications, subtleties appear quite often. I would like the authors to consider simplifying the message even more by concentrating on problems where no subtleties appear. Even though special cases are interesting, they often cloud the message. A particular, but not the only case is example 1 in Sec. 4.3.

Before I can potentially give a positive recommendation, I would like the authors to improve the presentation of their results.

Requested changes

1) While I appreciate the general result (11), for analytical calculations a semiclassical approximation has to be used. To this end, the limit $\Gamma\to\infty$ is employed. Am I correct that the notion "shell" is not suited in this limit as $I_\text{shell}$ then extends over all of phase-space. Why do you not use a "shell" where $\theta$ drops from 1 to 0 at some specific surface in phase-space with width $\Gamma$? I am thinking more like Fermi function that specifies the Fermi surface as the shell. The present choice $\exp(-x^2/\Gamma^2)$ is centered around 0 and does not really encode a shell.

2) The Sec. 2.1, there is the statement that the results have applications in many fields outside of quantum-mechanics. The operator $\hat H$ should then be understood as the wave-operator. However, it seems that for the results it is crucial that the Hamiltonian is Hermitian. Is this correct? How then to treat general systems, in particular damping and potential driving? It would be good to explain a bit more in details for what systems and in which formalism the results are applicable outside of QM. (there are explicit statement about pressure fields and velocity fields that would correspond to hydrodynamics without giving any details). Can H be the dynamical matrix of a classical system?

3) Generically, in a finite system, the zero mode splits away from zero due to overlap with other zero modes. The formula in Eq. (11) only counts the number of exact zero modes. I would expect such modes only to exist of the system is either fine tuned or infinitely large. Related to this is the fact that the index of any finite matrix vanishes. There is a subtle order of limits involved because of this. First the system size has to go to $\infty$ before the index can be calculated. An alternative would be to count all states in an energy window around zero with the window shrinking with system size. Could the authors comment on this point. This seems to be in particular relevant, as finite sized systems are shown in some of the figures.

3) In what way is the Wigner representation important for the examples? (I do understand that it is crucial for showing Eq. (18) and other exact results.). There are other ways of ordering, e.g., normal ordering. In general, I could simply just replace $\hat x \mapsto x$ and $\hat k \mapsto k$. It seems that in the limit $\Gamma \to \infty$ that could be enough. This would remove artifacts such as $\epsilon c'$ in Eq. (37) that muddy the discussion.

4) I do suggest to streamline and clarify the discussion in Chapters 3 and 4. I do like Chapter 3.1, 3.3 and the examples therein.

  • I do not understand what "a low-high wavenumber correspondence" is and why it deserves an own chapter. The subsection "gap conditions is less restrictive" seems to break the flow of the text as it refers to the relation to some other literature.

  • The example after Eq. (36) is quite confusing. It starts very general though uninspired Hamiltonian (with arbitrary function $c(x)$ and $V(x)$) before specifying that these function are in fact only a name for sin and cos. I guess that could be introduced from the beginning.

  • Section 3.4 is interesting but from my point of view it should be skipped at first reading; maybe some advice to the reader would be in order. I would hope to find an argument why the winding numbers sum up to zero (it is simply stated at the moment). In the discussion after (48), it would be maybe useful to add why in the condensed matter context zero modes that are separated in momentum space are often not of interested due to the presence of on-site disorder mixing them.

5) Chapter 4 is hard to digest as the current write-up is unfocused: it combines aspects very different aspects such as the construction of systems via the tensor-product and direct-sum and the calculation of topological indices. In my point of view, it should be more clearly separated. First the construction should be introduced, potentially with an example, only then the invariant should be calculated. The titles could also be more consistent, e.g., "4.2: tensor product construction: weak-topological insulator" and "4.3: additive construction: higher-order topological insulators". The "examples" in Sec. 4.2 seem to be only a single example. I do prefer example 2 over example 1 in Sec. 4.3. Example 1 seems to make a case against the conventional $C_4$ classification of HOTI which I do share. However, it is not quite pedagogical to introduce this as an example is "the use of the shell invariant is, however, not as straightforward as in the previous cases".

6) The notation in Eq. (14) is unclear. What is meant by $(U^\dagger dU)^{2D-1}$? It would be also good to state in Eq. (12) that in general the Hamiltonian symbol has internal degrees of freedom such that $H(x,k)$ is a matrix which would explain the trace. Similar, for Eq. (15).

7) In Eq. (54), the notation is not explained. What is $[\hat d,\hat H_F]^{2D-1}$ what is $\hat C'$?

7) In what relation, if at all, Eq. (22) stands when compared to Eq. (14). Both seem to encode the winding number.

8) Figure 5: the notions "gapless mode index", "gapped shell index", "gapless region in phase space", "gapless property of $\hat H$" are very much unclear. What is meant by "gap"? In which sense this property is a function of phase space? ...

9) Above Eq. (7): it would be useful to state that $[\hat H^2, \hat C] =0$ follows from ${\hat H ,\ hat C}=$.

The authors should read the manuscript once more carefully. I have found quite some typos. In particular, the following could be typos:

  • Figure 2: "Sp" should be replaced by "Tr" in figure for consistency
  • Figure 5: not clear what is the caption and where the text is? Is Eq. (14) part of the caption?
  • At some point there is a space before the "." or before the ":"
  • footnote 2: move 2 after "." on page 9
  • footnote 10: this is not a sentence, also the subscript 10 can be read as an exponent in the text: I would propose to use $E_1^{(n)}$ and drop the footnote
  • Is there a Ref. showing the result of footnote 4?
  • In footnote 4: "simplifying hypothesis" -> "simplifying hypotheses"
  • Eq. (17): it should read $(A \star B)(n,k)$
  • last sentence of first paragraph on page 15: $\theta_\gamma$ -> $\theta_\Gamma$
  • "Schockley" -> "Shockley" on page 16
  • "not valid" -> "invalid" on page 23
  • footnote 8: "largely below" -> "far below"
  • before Eq. (40): "gradient of $d\theta$" -> "gradient of $\theta$"
  • page 27 first paragraph: "around $x=0$ and $x=L$" -> "around $x=0$ and $x=L/2$"
  • page 29: "to make appear ..." -> "to make ... apparent"
  • reference [116] and [117] are double
  • appendix A is quite standard: I would propose to drop it, state the result and cite a reference: e.g., Stone, Goldbart, "Mathematics for Physics"
  • appendix B: what is $\sigma(A_\epsilon)$?
  • there are brackets missing in Eq. (107)
  • the notation in Eq. (132) is unclear: what is meant by $\hat d$, what is $[\hat d, \hat H_F]^{2D-1}$?
  • maybe the calculation after (110) can be compactified? At the moment, it is long without any clear understanding why and what is done.
  • page 29: "tric" -> "trick"

  • validity: top
  • significance: high
  • originality: top
  • clarity: good
  • formatting: good
  • grammar: excellent

Author:  Lucien Jezequel  on 2024-04-18  [id 4430]

(in reply to Report 1 on 2023-12-03)

1)The referee is right, the Gaussian function is not the more intuitive function to have in mind in the limit $\Gamma \xrightarrow{} +\infty$ to illustrate the shell. Even if this choice actually formally works to simplify our expressions, we indeed have more specifically in mind a Fermi-like distribution function with a finite temperature. We used a Gaussian expression because it is simple and also because it is used in index theory with Dirac operators to make the heat kernel apparent (when the shell is in k-space, in our language). That was thus an attempt to unify the formalism with the bulk-boundary correspondence. But actually, the specific expression of the theta function is not needed in the derivation of the semi-classical invariant, only are its properties. We have changed the text accordingly.

2)The referee is right, our approach only focuses on operators $H$ being Hermitian. But those already encompass a large variety of classical wave systems, from mechanical or photonic metamaterials to various fluid media, whose linearized wave dynamics mimics a Schrodinger-like equation which, in the absence of damping or potential driving, or other dissipation sources, is ruled by an Hermitian wave operator.

There, chiral symmetry can be implemented in different ways. If dealing with a lattice described with a tight-binding model, then there is no much difference between (spinless single-particle) quantum condensed matter systems and classical wave metamaterials; in that case, chiral symmetry can follow from bi-partition of the lattice. If dealing with continuous fluids, then it turns out that the chiral symmetry of the wave operator in phase space follows from time-reversal symmetry of the original set of primitive differential equations. This is because primitive equations -- basically momentum conservation and mass conservation -- are first order differential equations in time. As such, they yield a relation between fields that are odd with respect to time inversion, such as velocity fields, and fields that are even with respect to time reversal, for instance pressure fields. This automatically creates a kind of bipartition between the physical fields depending on their parity in time, that one can eventually translate into a chiral symmetry of the wave operator. Such a structure appears in geo- and astrophysical fluid dynamics and plasmas models, in the absence of Coriolis force and magnetic field [1-4]. We have clarified this point in the updated version of our manuscript.

Other mechanical systems, such as certain mass-spring lattices, although usually more naturally described by second order differential equations of motion (Newton laws expressed for position or displacements variables) have also been expressed as an eigenvalue problem for a Hermitian chiral symmetric operator, where indeed $H$ corresponds to the dynamical matrix of this classical system (see for instance [5]).

Finally, as the referee mentions, classical systems are sometimes also described more generally by a non-Hermitian wave operator (e.g. in the presence of dissipation, or as a linearized perturbation of a nonlinear problem). In that case, we expect one could extend the mode-shell correspondence to the non-Hermitian realm in the same way it was done for the bulk-boundary correspondence. In particular, it is well-known that non-Hermitian point-gap Hamiltonians $H$ can be mapped onto Hermitian chiral Hamiltonians $H'$ as $H'= \begin{pmatrix} 0&H^\dagger \ H&0 \end{pmatrix}$ which is our starting point in the present manuscript. Also, line-gap non-Hermitian Hamiltonians can be continuously deformed to gapped Hermitian Hamiltonians with real spectrum where the usual topological tools apply. We previously discussed such an example of the mode-shell correspondence in a non-Hermitian model in a different symmetry class [6].

3A) We thank the referee for this comment and agree with the point that is raised. Indeed, capturing zero-modes in finite systems is a bit tricky and requires some care. In particular, as pointed out by the referee, finite size effects always split the zero-modes in energy, so the formula (5) cannot be used, since strictly zero energy modes generically do not exist. This is one of the reasons why we introduce the smooth energy filter $f(E)$ in section 2.2, that we use to define our version of the flatten Hamiltonian in equations (6) and (8). This energy filter solves this issue by selecting the mid gap states with an energy resolution (given by the smoothness of $f$) such that the "almost-zero-energy" modes are selected as if they had zero energy. In that way, one can get a topological number with a very good approximation. Still, to obtain this good approximation, there is indeed a particular order of limits. Typically, the smooth filtering can be done using the function $f(E) = \tanh(E/\delta)$ where $\delta$ characterises the size of the energy window. In order to flatten the bands of energy $|E| > \Delta$, we need to have $\delta \ll \Delta$, but at the same time, we cannot take directly the limit $\delta = 0$ due to these finite size effects. In a previous paper [7], we proved rigorously that in the case of the 1D bulk-edge correspondence and in the limit where the window in energy decreases as $\delta \sim \frac{1}{\sqrt{L}}$ with $L$ the length of the system (scaling where $\delta$ decreases even quicker are also possible), the mode index and the shell index both converge exponentially fast toward an integer in the large size limit $L\xrightarrow{}\infty$, implying the existence of modes whose energy tends toward zero in this limit. We expect these results to extend in a similar way to the general mode-shell correspondence.

3B) First, the Wigner transform is used to represent the zero-modes in phase space in the figures for each example. Thanks to this representation, we can introduce the appropriate shell that surrounds it in phase space.

Second, in the example of the 1D Dirac equation with both a varying potential and velocity (section 3.2 the referee refers to), the existence of the term $\epsilon c'$ does not really come from a special ordering but rather from the semi-classical expansion of the Moyal product ($\epsilon c'$ is an higher order term in this expansion). Actually, in any ordering, $c(\hat{x}) \hat{k}$ and $\hat{k}c(\hat{x}) $ cannot both be mapped to $c(x)k$. We agree with the referee that this term makes the discussion a bit more involved. But on the other hand, we are aware of situations where keeping the $c'$ term is relevant [2-8]. Simply ignoring it may lead to the misleading and unwanted message that we can indeed always make use of the substitutions $\hat{x} \xrightarrow{} x$ and $\hat{k} \xrightarrow{} k$. We cannot neither just take a constant $c$ otherwise we would not get zero-modes.

In the new version, we discuss this issue and explain why we need a small parameter $\epsilon$ to reach the semi-classical limit in this specific model, and then we use the simplest symbol in the semi-classical limit to continue the discussion, as the referee suggests.

4) We denote by a "low-high wavenumber correspondence" in section 3.2 the situation where the zero-modes are trapped in phase space around $k\sim 0$, and the shell the shell that enclose them is defined at large $k$. We agree that the meaning of this appellation is not obvious, in particular from the title. We have changed the title of the section to a "dual bulk-boundary correspondence in wavenumber space", since the idea is really to swap the roles of $x$ and $k$, compared to a model where the bulk-boundary correspondence applies. Since the mode-shell correspondence allows us to address this original problem on the same footing as the usual bulk-boundary correspondence, it makes sense to dedicate it a specific section.

We have reorganized the section 3.2. We keep the discussion general with $V(x)$ and $c(x)$ to derive the symbol and discuss the validity of the semi-classical approximation in full generality (see point 3B) above)). Then we replace them by their expression at the end for numerical purposes. We shifted the discussion about the elliptic condition to an appendix, E, entitled "Existence of topological modes on closed manifold of odd dimension: Less restrictive gap assumption than elliptic condition"

Concerning the section 3.4, we agree with the referee. We warn the reader in the new version that this section can be skipped at first reading.

We can understand why the total chiral number of zero-modes in a finite Hilbert space must sum up to zero in the following way: The total number of zero-modes in a system is given by $I_\text{mode}= Tr(C(1-H_F^2)$ where the cut-off $\theta_\Gamma$ is the identity everywhere (we take the whole system). Then the corresponding shell index in the mode-shell correspondence is $\mathcal{I}_\text{shell}=Tr(\hat{C}+1/2Tr(\hat{C}\hat{H}_F[1,\hat{H_F}])$. Since we consider systems with a balanced chirality of degrees of freedom, the first term vanishes $Tr(\hat{C})=0$, and the second term also vanishes because $[1,\hat{H_F}]=0$. Thus, since the isolated chiral zero-modes are given by the winding number evaluated over a shell that surrounds them, it follows that the sum of the winding numbers is zero. We now explain this point after (48).

Concerning the remark about the fragility of modes only separated in wavenumber, this point was already already made above in the article, but we agree that it makes sense to recall it in that section. We have thus added a footnote (number 8) accordingly. Also, just to make the point clear, we do not say that zero-modes separated in $k$-space are disregarded in condensed matter physics. Dirac points and Weyl nodes would be strong counter examples. We just say that they can couple due to short-range scattering processes in the bulk, while zero-energy edge states are insensible to such processes (at least provided the gap assumption does not break).

5)We thank the referee for this remark. We originally tried to separate the tensor product construction from the computation of the topological indices in the new version, however it resulted in too many repetitions or confusion in our opinion. Also, we prefer to present the general theory of weak/higher order topology before the presentation of the tensor product construction as the later is a shortcut useful in particular examples but is not necessary for the theory in general.

In the new version, we have modified the titles to be more specific as recommended by the referee. The examples in Sec. 4.2 are two separated examples. We have added one figure and separated them more clearly to avoid any confusion. In sec 3 we agree that the example 1 is less straightforward than the second one as one does not have an analytical expression of the shell invariant. We still want to present this example with open boundary due to its relevance in the physical literature and because it allows us to highlight numerical methods that we believe to be as useful as analytical results. We have added a discussion on that during the example. We have also swapped the order of the two examples in the new version so that the simpler one is presented first.

6) The notation $dU$ denotes the differential in phase space $dU = \sum_i (\partial_{x^i} U) dx^i+(\partial_{k^i} U) dk^i$. $(U^\dagger dU)^{2D-1}$ is then a $2D-1$-form which is an anti-symetrised sum of all possible order of derivative of the product of $U^\dagger$ with $dU$. In the revised version, we have added Eq.15 and the paragraph around to detail this technical point.

The paragraph above Eq.15 in the revised version of our manuscript should clarify that the symbol is still a matrice, acting on the internal degrees of freedom.

7A) $\hat{C}'$ is a typo. It should simply be the chiral operator $\hat{C}$.

$[\hat{d},\hat{H}_F]^{2D-1} is a commutator to the power $2D-1$ of $\hat{d}$ with $\hat{H}F$. $\hat{d}$ being a form, the commutator is defined in a similar way than the term $(U^\dagger dU)^{2D-1}$ above. It is an antisymetric sum where differentials $\partial U are replaced by commutators $[\hat{d}_{j},\hat{H}] with $\hat{d}{2j} = x_j$ and $\hat{d} in the continuous case and } = \partial_{x_j$\hat{d}_{2j} = T_j^\dagger n_j and $\hat{d}_{2j+1} = T_j in the discrete case. The two coincide in the semi-classical limit where the commutators are replaced by derivatives in phase space. In the new version, the short-hand $\hat{d}$ has been replaced by the notation $\hat{\alpha}$, to avoid any confusion with the differential $d$.

7B) Eq. (23) is one particular case of Eq. (14) where the shell is made of two $1D$ Brillouin zones, one being picked at large positive $x$, the other one at large negative $x$. The goal of this section is to show how to recover the particular case where the chiral zero-modes at an interface is given by a difference of bulk indices, from a relatively simple computation. This is also true for sections 3.2 and 3.3 which aim at deriving the mode-shell correspondence in different specific 1D situations. We have added a comment after Eq. (26) in the revised manuscript.

8) Thanks to the Wigner-Weyl transform, operators, and in particular the Hamiltonian, can be represented in phase space. This is especially true in the semi-classical limit. The gap refers to the spectral gap in phase space, that is a spectral gap of the symbol Hamiltonian. This is a local definition. A gap of $\hat{H}$ is defined for a point $(x,k)$ of phase space if its symbol $H(x,k)$ at this point is gapped. We can also say that a point of phase space $(x,k)$ has a gap $\Delta$ if the Wigner-Weyl transform of the modes of energy inside the gap $E \in [-\Delta, \Delta]$ are all negligible in $(x,k)$. We assume that the shell is a region of phase space which is gapped in this sense, so we qualify the index defined on it as "gapped". We use the appellation "gapless mode index" to denote the index that count the chiral number of zero-modes as they live in the gap.

9)We thank the referee for this remark and have modified the text accordingly.

*$Sp$ is actually a notation for ''spectrum'. It is now explicit in the caption of the revised manuscript. *The Eq (14) is not part of the caption, it is a coincidence that it follows directly the caption. *ok *ok *In the new manuscript, we have modified the notations in a way which avoid the problem. *After some verification, we had in mind the similar but slightly different result for the formulation of the shell index $I_{shell}=Tr(CH(dH)^n)$ which vanishes when $n$ is even (as $H$ anticommutes with $C$ and $dH$). When $n$ is odd (the case we are in the article) such formulation is proportional to the formulation in terms of unitary. However this relation cannot be done in odd dimension, therefore the two results are not equivalents. We thank the referee for their vigilance and deleted the corresponding footnote. *ok *ok *ok *ok *ok *ok *ok *ok *Despite the close names and release date, those references mention different papers *We agree that this is fairly standard for some researchers and have added the mentionned reference. We preferred to keep the appendix in the manuscript for those who are less familiar with it. The presentation is kept concise and presents the key results needed elsewhere in the article. *This is an old notation for the symbol. It has been removed *ok *As explain above this is a compact form to write an antisymetrised sum of commutator $[d_{j},H_F]$ which is written completely in equation (131) with in the continuous case and $d_{2j} = T_j^\dagger n_j$ and $d_{2j+1} = T_j$ in the discrete case. *We agree that the computations in Appendix E are quite long and difficult to follow. We have totally revised this part to make the proof detailed and accessible to most readers. Rewriting this part, we changed the orientation convention of phase space which changed some signs of expressions in the main text. *ok

[1] Manolis Perrot, Pierre Delplace, and Antoine Venaille. Topological transition in stratified fluids. Nature Physics, 15(8):781–784, 2019. [2] Armand Leclerc, Guillaume Laibe, Pierre Delplace, Antoine Venaille, and Nicolas Perez. Topo- logical modes in stellar oscillations. The Astrophysical Journal, 940(1):84, nov 2022. [3] Jeffrey B. Parker. Topological phase in plasma physics. Journal of Plasma Physics, 87(2):835870202, 2021. [4] Hong Qin and Yichen Fu. Topological langmuir-cyclotron wave. Science Advances, 9(13):eadd8041, 2023. [5] Florian Allein, Adamantios Anastasiadis, Rajesh Chaunsali, Ian Frankel, Nicholas Boechler, Fotios K. Diakonos, and Georgios Theocharis. Strain topological metamaterials and revealing hidden topology in higher-order coordinates. Nature Communications, 14(1), October 2023. [6] Lucien Jezequel and Pierre Delplace. Non-hermitian spectral flows and berry-chern monopoles. Phys. Rev. Lett., 130:066601, Feb 2023. [7] Lucien Jezequel, Clément Tauber, and Pierre Delplace. Estimating bulk and edge topological indices in finite open chiral chains. Journal of Mathematical Physics, 63(12), December 2022. [8] A. Venaille and P. Delplace. Wave topology brought to the coast. Phys. Rev. Research, 3:043002, Oct 2021.

---

## Round 1 · Referee Report · Anonymous (Referee 2) · 2024-1-11

Report

In the manuscript the authors formulate mode-shell correspondence which is meant to generalize bulk-boundary correspondence of topological matter. I have two concerns with the present manuscript: the presentation of the results and their relevance to condensed matter systems.

Relevance:
The introduction tries to depict how bulk-boundary correspondence had to evolve and to be reformulated each time a new topological phase was found. This is where the authors find motivation for their unifying formulation that covers bulk-boundary correspondence and even goes beyond it.

Unfortunately, I find such motivation rather problematic. The bulk-boundary correspondence had always been a one-to-one relation between the bulk and the boundary topological invariants. In that sense, mode-shell correspondence is no exception. From the examples presented, I simply do not see an added value of mode-shell correspondence for a reader who is interested in bulk-boundary correspondence of different topological phases.

Furthermore, I doubt the authors claims that their mode-shell correspondence applies equally well to higher-order topological insulators (HOTIs). The reason is that bulk-boundary correspondence for HOTIs, unlike mode-shell correspondence, is not one-to-one: consider a stack of a first- and a second-order topological phase (i.e., direct sum of their Hamiltonians), the bulk invariant of the resulting phase corresponds to the "sum" of the corresponding bulk invariants, whereas on the boundary, the second-order signature is invisible (corner states cannot coexist with edge modes). Hence, one finds a correspondence between bulk and boundary only after "modding out" higher-order bulk invariants.

On a positive side, the shell-mode correspondence seems to generalize bulk-boundary correspondence to the situations where one has localization in the momentum space instead at the boundary. Do I understand correctly that the latter situation does not bare any relevance to condensed matter systems? If it does, it would be nice to illustrate some physical consequence of such correspondence.

Lastly, the expression for I_shell (say Eq. 9) looks very similar to the expressions for topological invariants from non-commutative geometry. It would be good to cite relevant works by H. Schulz-Baldes, J. Bellissard, and E. Prodan and explain the connection.

Presentation:
The main story is constantly being interrupted by (rather irrelevant) side remarks which sometimes even take form of whole subsections (for example 2.2, remarks in 3.2 in form of subsections "Remarks on...", "Validity of..."). Then in Sec. 4, the story line is totally lost. Why is multiplicative tensor product construction discussed there? Why is additive tensor product construction discussed there? I see no connection to shell-mode correspondence.

If authors want to introduce shell-mode correspondence as a unifying framework, then they need to provide its formulation and to apply it to various topological phases of interest. Focusing on 1d chiral systems and bunch of unrelated topics does not do the job in my opinion. Why not working out an example for a (strong) 3d chiral topological insulator?

In conclusion, I think the work deserves publication but only after a major overhaul of the manuscript.
  • validity: -
  • significance: -
  • originality: -
  • clarity: -
  • formatting: -
  • grammar: -

Author:  Lucien Jezequel  on 2024-04-18  [id 4431]

(in reply to Report 2 on 2024-01-11)

The referee writes:

In the manuscript the authors formulate mode-shell correspondence which is meant to generalize bulk-boundary correspondence of topological matter. I have two concerns with the present manuscript: the presentation of the results and their relevance to condensed matter systems.**

Relevance:
The introduction tries to depict how bulk-boundary correspondence had to evolve and to be reformulated each time a new topological phase was found. This is where the authors find motivation for their unifying formulation that covers bulk-boundary correspondence and even goes beyond it.

Unfortunately, I find such motivation rather problematic. The bulk-boundary correspondence had always been a one-to-one relation between the bulk and the boundary topological invariants. In that sense, mode-shell correspondence is no exception.
From the examples presented, I simply do not see an added value of mode-shell correspondence for a reader who is interested in bulk-boundary correspondence of different topological phases.

Our response:

The mode-shell correspondence establishes indeed a one-to-one relation between two indices. In certain cases, those indices can physically be interpreted as bulk and boundary invariants. In that sense, it includes the bulk-boundary correspondence, but is not restricted to it.

The purpose of our introduction is twofold. One is to recall that, even though the bulk-boundary correspondence can be formally summarized as the equality between a bulk and a boundary topological invariants, as the referee says, the expression of those indices may indeed change from phase to phase, and is not, in general, obvious neither to guess nor to justify. In other words, the identification of the relevant "bulk" invariant in a given model is not, in general, a simple task, and various such invariants have been identified over the years depending on the community interests. The identification of a Z2 index in time-reversal 2D systems by Kane and Mele, and the construction of higher-winding numbers in driven Floquet systems by Rudner, Lindner, Berg, and Levin are two such remarkable examples that we recall in the introduction. The mode-shell correspondence provides a systematic method to identify and derive invariants that are demonstrated to correspond to the number of topological modes (and this article is dedicated to the chiral symmery class).

The second goal of the introduction is to recall that there are many results in the literature, in physics but also in mathematics, that resemble the bulk-edge correspondence, by sharing the same idea which is, roughly, that the number of some special gapless states is given by a topological number. There is not, in this general statement, neither a clear notion of "bulk" and "boundary", nor any clear hypothesis for the validity of such a unifying statement. A theoretical framework is thus needed to clarify and justify this idea. This is the goal of the mode-shell correspondence, and we don't understand why the fact that it is rooted by an equality between two indices is "problematic"

Therefore, the unambiguous interest of the mode-shell correspondence is that it captures, on the same footing, distinct physical situations, whose some of them coincide with the bulk-boundary correspondence. Those distinct physical situations can all belong to the same symmetry class and dimension in the tenfold way classification, and result from the fact that the mode-shell correspondence deals with phase space and not reciprocal space only. It provides a systematic way to identify the relevant topological index in all those situations through the definition of the shell, and justifies, in a rather rigorous way, the use of those invariants to count the topological zero-modes. Several of those situations are explained and illustrated all along the text for the chiral symmetry class in dimension $D=1$, and we regret that the referee was not able to see any "added value" of such a unified and explicit approach. We hope this strong negative criticism without nuance is based on a misleading reading of our work. By rephrasing here several distinct situations to which our theory applies, including situations that cannot be qualified as examples of the bulk-boundary correspondence, we hope we shall clarify the "added values" of our work to the referee.

First, the meaning of "boundary" and "bulk" in "bulk-boundary correspondence" may lead to ambiguities and subtleties. The original sense is, of course, clear: the boundary of a solid is the location (edge, surface) where the solid cease to exist. Then the bulk refers to a description of the system where boundaries are ignored, e.g. by considering the Bloch Hamiltonian after assuming translation invariance. The bulk topological invariant is then naturally given by an integral over k-space. This is the typical situation of what is meant when referring to the "bulk"-"boundary" correspondence in the literature, and the original demonstrations [1-3], as well as textbooks [4], share this picture.
Now, let us consider an apparently similar situation that consists in an interface made of a domain wall. In that case, the domain wall is not, strictly speaking, a boundary, since the material still exists on both sides. We could have in mind a very smooth transition, for instance, a slowly varying Semenoff mass term in Boron Nitride, or on 2D Dirac fermions at the surface of a 3D Z2-topological insulator. Usually, it is claimed that such a situation can be understood from the previous one, by considering that the interface state can be seen as a boundary state living in between the two topologically distinct regions. In a lattice system, however, the difference of the "bulk" invariants (i.e. given by an integral in k-space) may nonetheless count additional modes than those of interest, i.e. localized at the domain wall/interface, as explained in our manuscript in 1D. In particular, the mode-shell correspondence is tailored to count modes per valley, unlike the bulk invariant that can vanish.

A second related issue concerns continuous models, which are also intensively used in condensed matter. In that case, "bulk" invariants are not systematically well-defined anymore, owing to the lack of compactness of the base space manifold, while topologically robust interface states may exist. This issue appears intrinsically in continuous media such as e.g. fluids and plasmas, where critical differences exist between domain walls and hard walls (i.e. boundaries) geometries. The mode-shell correspondence circumvents those issues (interface/valley/non-compactness of k-space), by providing a way to select, in real space and in valley, the zero-modes of interest, and to characterize their topology in phase space. In particular, as shown in our manuscript, such zero-modes, in chiral 1D-systems, are characterized by a winding number given by an integral in (x-k) phase-space, and not anymore in k-space only, in contrast with what was above defined as a "bulk" invariant. In that sense, the "bulk-boundary" correspondence is thus extended to a "phase space-interface" correspondence: there is no actual boundary in the system and the bulk invariant in k-space is replaced by another invariant in phase space.
For this reason, we thus believe that our approach is of interest to readers concerned with the bulk-boundary correspondence.

A third natural extension is the case of defect-modes in solids, as addressed in Teo and Kane's work [5]. In that work, the authors "introduce a generalization of the bulk-boundary correspondence that relates the topological classes to defect Hamiltonians to the presence of protected gapless modes at the defect", to quote them. It is neither a boundary nor an interface problem, as that discussed above, and the notion of codimension of the defect now enters the game. This is thus a different physical framework. Still, the semiclassical expression of our shell-invariant, Eq(14), includes Teo-Kane formula for point defects, where the surface of integration is the tensor product of a sphere in position space and of the Brillouin zone in k-space. Within the mode-shell correspondence approach, zero-modes localized in real space, such as defect modes, and zero-modes localized at given valleys in k-space (as discussed above), are treated on the same footing. We believe that this is an important "added-value" of our formalism.

Furthermore, and this is an important point, we provide an explicit demonstration of our general formula Eq(14), which was not the case in Teo and Kane paper. We think that this single demonstration is in itself a significant "added-value" to the field as Teo and Kane's paper has attracted significant interests in condensed matter physics. In the new version of our manuscript, the steps of this derivation have been made more detailed and explicit (Appendix F). We hope those changes will help convince the referee of the scope of our work.

At this stage, we have re-explained how the mode-shell correspondence captures "bulk-boundary", "phase space- interface" and "defect gapless modes - topology" correspondences, by identifying explicitly and in a systematic way the expression of the topological invariants that counts the zero-modes of interest in those physically different situations. It seems to us that this unified picture constitutes an "added-value" of our approach. But this is not all, so let us continue to quickly enumerate the situations that are captured by the mode-shell correspondence, beyond the bulk-boundary correspondence, as extensively discussed in the manuscript.

Another interesting and non standard situation is found when the system is unbounded and periodic. In that case, there is precisely no boundary, so that the notion of "bulk-boundary" correspondence is meaningless. Still, the mode-shell correspondence applies, and relates the existence of zero-modes in such systems with an appropriate topological invariant. This invariant, in its semiclassical formulation, is now given by an integration over position space only, in contrast with both the usual "bulk" invariants expressed in k-space, and also with the mixed (x-k) phase-space invariant encountered e.g. in defects Hamiltonian systems and in domain-wall problems. It seems to us that this single example of an invariant in real space alone brings an "added-value" of our theory to topological physics. Besides, it also relates with independent famous mathematical results, derived decades ago (Atiayh-Singer theorem and Callias index formula), that we aim at bringing closer to the physics community. This original situation, together with the bulk-boundary and the "phase space-interface" correspondences, are three direct applications of the mode-shell correspondence, when applied to 1D chiral systems only.

Since our formalism, and in particular our semiclassical formula Eq(14), also applies to higher dimension, we then investigated higher dimensional models to illustrate the efficiency of our approach. However, our starting point, in this paper, being the mode-invariant that captures only zero-dimensional modes, we restrict ourselves here to the gapless invariant usually associated with gapped $D=1$ chiral system in the tenfold way classification. As announced in the introduction and in the conclusion, the higher dimensional zero-mode invariants will be discussed in the Part II of this work (including both chiral and unitary symmetry classes, and thus in particular strong 3D chiral topological insulators). Still, this does not imply that the zero-dimensional modes we describe in this manuscript are not relevant in higher dimension; they are actually the building blocks of weak- and higher-order chiral insulators. To illustrate this point, we discuss several models in both situations and show that their zero-modes are indeed captured by our shell invariant, which can take the form of an integral of a higher dimensional surface, such as a 3-sphere, as explicitly demonstrated in arbitrary dimension. So again, within the same mode-shell picture, we are able to embrace also different higher dimensional topological phenomenologies. The only difference between all those cases is the geometry and dimension of the shell. We believe that this higher-dimensional discussion also constitutes an "added-value" to topological physics and to readers interested in the bulk-edge correspondence.

Finally, as a spoiler of the Part II of this work, and as briefly announced in the conclusion of the present manuscript, bulk 2D Dirac points and 3D Weyl points localised in reciprocal space in semimetals, can also be cast as higher-dimensional zero-modes within the mode-shell correspondence. Their topological characterization is then found to be the Berry winding number and the Chern number respectively, as expected. Here again, this result does not enter the bulk-boundary correspondence, but is nonetheless naturally contained in the mode-shell correspondence.

For all these multiple reasons, we thus strongly think that our work is "relevant" and brings several "added values" to topological physics in general and to the bulk-boundary correspondence in particular. We hope that this summary, together with all the changes made to improve the clarity of our manuscript, including the introduction and the derivation of our main formula, could change the referee's opinion about the interest of our work.

The referee writes:

Furthermore, I doubt the authors claims that their mode-shell correspondence applies equally well to higher-order topological insulators (HOTIs). The reason is that bulk-boundary correspondence for HOTIs, unlike mode-shell correspondence, is not one-to-one: consider a stack of a first- and a second-order topological phase (i.e., direct sum of their Hamiltonians), the bulk invariant of the resulting phase corresponds to the "sum" of the corresponding bulk invariants, whereas on the boundary, the second-order signature is invisible (corner states cannot coexist with edge modes). Hence, one finds a correspondence between bulk and boundary only after "modding out" higher-order bulk invariants.

Our response:

The configuration proposed by the referee is not a higher-order TI, but rather an hybrid construction between two kinds of topological phases, in the spirit of the recent work [6]. Also, the system proposed by the referee is gapless on its edges which seems to contradict HOTIs basic principles (see the founding papers [7-8]). So we are skeptical about the conclusion of the referee, from this example, that our mode-shell theory does not apply to HOTIs.

Then, if the question is, "can we also analyse such a model with the mode-shell correspondence?" the quickest answer is no. The main reason is that the mode-shell analysis requires the Hamiltonian to be gapped on the shell. This is not satisfied in the current situation, since the shell that encloses the corner modes has to intersect the boundary, which is gapless here by hypothesis.

So this is certainly an interesting situation, but it goes beyond the scope of the present manuscript, and more information should be provided to investigate it (e.g. is chiral symmetry preserved? If not, it does not enter the present framework. If yes, the referee should precise how to get edge states in a 2D chiral symmetric first-order TI...?).

In contrast, we (1) show that our shell invariant correctly gives the zero-energy modes in two cornerstones models for HOTIs, one being discrete, the other one being continuous, and (2) provide an explicit proof of the correspondence between this invariant and corner modes/higher order interface modes. So it seems fair to us to claim that the mode-shell approach has some relevance for HOTIs.

The referee writes:

On a positive side, the shell-mode correspondence seems to generalize bulk-boundary correspondence to the situations where one has localization in the momentum space instead at the boundary. Do I understand correctly that the latter situation does not bare any relevance to condensed matter systems? If it does, it would be nice to illustrate some physical consequence of such correspondence.

Our response:

We thank the referee for his/her warm enthusiasm about this original situation that we unveil. Our goal here was to explain and illustrate, with models, the formal equivalent role of position and wavenumber/momentum spaces in the mode-shell correspondence, not to propose a material realization, which would go beyond the scope of this work.

The fact that this model exhibits zero-modes localized in wavenumber space, rather than in real space, does not prevent its relevance in condensed matter physics. Topological modes taking place in a confined region of momentum space are quite usual in 2D (e.g. Dirac points, valley Hall effect) and 3D (e.g. chiral anomaly, Weyl nodes). Our model is a kind of Dirac equation in 1D, as encountered in graphene strips and Carbon nanotubes at low energy. However, a prerequisite of our model is the periodicity/unbounded condition of the system, which is experimentally met for instance in rolled up graphene strips into cylinders, and ring-Carbon nanotubes. Moreover, the application of the topological tools we discuss is not restricted to condensed matter systems, and ranges from many classical and quantum metamaterial and devices to field theories. Photonics, for instance, also provides interesting experimental perspectives, with the realization of rolled-up honeycomb lattice crystals [9-10], which might also constitute a good starting point to implement the effect we discuss.

Finally, the fact that this original situation, in contrast with other configurations we describe, has not been identified experimentally yet is, in our opinion, more a strength of our theory than a weakness, as it can stimulate future experimental investigations.

The referee writes:

Lastly, the expression for $I_\text{shell}$ (say Eq. 9) looks very similar to the expressions for topological invariants from non-commutative geometry. It would be good to cite relevant works by H. Schulz-Baldes, J. Bellissard, and E. Prodan and explain the connection.

Our response:

The referee is right to mention the proximity with those kinds of invariants, which are indeed similar when our theta cutoff is only a function of space. This situation describes the bulk-edge correspondence. The strength of the mode-shell correspondence is that it more generally involves a theta cutoff that is function of phase space, and thus allows to tackle other situations.

The referee writes:

Presentation:
The main story is constantly being interrupted by (rather irrelevant) side remarks which sometimes even take form of whole subsections (for example 2.2, remarks in 3.2 in form of subsections "Remarks on...", "Validity of...").

Our response:

We are a bit confused with the fact that the referee complains about systematic interruptions of the main story, while at the same time, we are asked to add a remark about the relation between our shell invariant and topological invariants from non-commutative geometry. We believe that what are relevant or "irrelevant side-remarks" may depend on the reader background and interests, and it is not clear to us why would non-commutative geometry oriented remarks be more "relevant" than e.g. a comment on the validity of our semiclassical approximation which, in our opinion, is crucial to establish our theory.

This said, we thank the referee for his specific comment about the subsections remarks that may indeed break the flow. We have revised the manuscript such that the remark about the elliptic condition has been moved to an Appendix, while the discussion about the validity of the semiclassical limit, which is crucial in our theory, has been incorporated into the discussion of the model, to make the flow less interrupted. We are just left with one paragraph about the protection of zero-modes separated in wavenumber space, first because it is precisely the case of the example discussed just above, and second because it is also a discussion asked by the referee 1.

The referee writes:

Then in Sec. 4, the story line is totally lost. Why is multiplicative tensor product construction discussed there? Why is additive tensor product construction discussed there? I see no connection to shell-mode correspondence.

Our response:

Multiplicative and additive tensor constructions are simple and powerful methods to generate high-dimensional models whose topological properties are encoded into lower-dimensional sub-block models. We use those constructions to generate higher-order and weak chiral symmetric models, whose topological properties (the number of chiral zero-modes) can then simply be inferred from the lower-dimensional sub-blocks discussed in the previous sections. So the first interest is that the evaluation of the number of zero-modes of such models becomes straightforward. Without those tools, we would have been very likely constrained to numerical studies to determine the number of zero-modes in higher-dimensional models. The second interest is that it serves to check our results when we apply the mode-shell correspondence directly to the high-dimensional systems.

The referee writes:

If authors want to introduce shell-mode correspondence as a unifying framework, then they need to provide its formulation and to apply it to various topological phases of interest.

Our response:

The referee is right, and this is exactly what we do.

The referee writes:

Focusing on 1d chiral systems and bunch of unrelated topics does not do the job in my opinion.

Our response:

We do not understand what the referee means by a "bunch of unrelated topics". We believe the case of 1D chiral symmetric systems is a good starting point to illustrate our theory with chiral symmetry, and is rich enough. In the present manuscript, our purpose is to show that the very basic zero-dimensional zero-energy chiral modes equally appear, through the mode-shell correspondence, as boundary modes in the bulk-boundary correspondence (section 3.1), interface modes in the "phase-space - interface" correspondence (section 3.3), k-localized modes in what we now call the "dual bulk-boundary correspondence in wavenumber space" (section 3.2), localized modes in both position and wavenumber (section 2.4), edges modes of weak topological insulators (section 4.2), corner modes of higher-order topological insulators (section 4.3). All those situations are captured by the very same mode-invariant that turns into different semiclassical shell invariants, which justifies to treat them consistently in the same paper.

The referee writes:

Why not working out an example for a (strong) 3d chiral topological insulator?

Our response:

This is a very good suggestion, that was already announced in the end of our introduction :"Other higher dimensional chiral symmetric topological systems are expected from the tenfold way classification. Those are not discussed in the present paper, but will be treated in a follow up paper". We believe the length of this manuscript is long enough, and that the introduction of other mode-invariants (either non chiral or higher-dimensional) should be discussed in the part II of this work.

The referee writes:

In conclusion, I think the work deserves publication but only after a major overhaul of the manuscript.

Our response:

We hope our replies to the referee, together with the numerous modifications done to our manuscript, have clarified our results.

[1] Yasuhiro Hatsugai. Chern number and edge states in the integer quantum hall effect. Phys. Rev. Lett., 71:3697–3700, Nov 1993.
[2] Yasuhiro Hatsugai. Edge states in the integer quantum hall effect and the riemann surface of the bloch function. Phys. Rev. B, 48:11851–11862, Oct 1993.
[3] Gian Michele Graf and Jacob Shapiro. The Bulk-Edge Correspondence for Disordered Chiral Chains. Communications in Mathematical Physics, 363(3):829–846, November 2018.
[4] B Andrei Bernevig. Topological insulators and topological superconductors. Princeton university press, 2013.
[5] Jeffrey C. Y. Teo and C. L. Kane. Topological defects and gapless modes in insulators and superconductors. Phys. Rev. B, 82:115120, Sep 2010.
[6] Md Shafayat Hossain, Frank Schindler, Rajibul Islam, Zahir Muhammad, Yu-Xiao Jiang, Zi-Jia Cheng, Qi Zhang, Tao Hou, Hongyu Chen, Maksim Litskevich, Brian Casas, Jia-Xin Yin, Tyler A. Cochran, Mohammad Yahyavi, Xian P. Yang, Luis Balicas, Guoqing Chang, Weisheng Zhao, Titus Neupert, and M. Zahid Hasan. Discovery of a hybrid topological quantum state in an elemental solid, 2024.
[7] Wladimir A. Benalcazar, B. Andrei Bernevig, and Taylor L. Hughes. Quantized electric multi- pole insulators. Science, 357(6346):61–66, jul 2017. 7
[8] Frank Schindler, Ashley M. Cook, Maia G. Vergniory, Zhijun Wang, Stuart S. P. Parkin, B. Andrei Bernevig, and Titus Neupert. Higher-order topological insulators. Science Advances, 4(6), jun 2018.
[9] Rémi Briche, Aziz Benamrouche, Pierre Cremillieu, Philippe Regreny, Jean-Louis Leclercq, Xavier Letartre, Alexandre Danescu, and Ségolène Callard. Tubular optical microcavities based on rolled-up photonic crystals. APL Photonics, 5(10), 2020.
[10] A Danescu, Ph Regreny, P Cremillieu, and JL Leclercq. Fabrication of self-rolling geodesic objects and photonic crystal tubes. Nanotechnology, 29(28):285301, 2018. 8

---

## Editorial Decision

resubmitted